# Spatio-temporal dynamics of Hendra virus in Australia reveal stable maintenance of diverse viral clades among *Pteropus* bats

Claude Kwe Yinda [1,16], John-Sebastian Eden [2,3,16], Erica T. Prates [4,16], Anna Vlot [4], Sarah van Tol [1], Sarah L. Anzick [5], Jianning Wang [6], Kim Halpin [6], Benny Borremans [7], Tamika J. Lunn [8,9], Kent Barbian [5], Brown Bulloch [1], Benjamin Greene [1], Kimberly Meade-White [1], Trenton Bushmaker [1], Caylee A. Falvo [10], Daniel E. Crowley [10], Devin N. Jones-Slobodian [11], Manesh Shah [12], Mirko Pavicic [4], William Carr [13], Craig Martens [5], Daniel Jacobson [4], Raina K. Plowright [10], Alison J. Peel [3,14,15,17] & Vincent J. Munster [1,17] ✉

Hendra virus (HeV) was discovered in 1994 in Australia. Limited genomic data have hindered comprehensive understanding of HeV's evolutionary dynamics. Here we recovered 48 HeV genomes from bats and 9 from horses from Australia between 2016 and 2020, revealing four distinct clades. Each clade was distributed over a large spatial area with multiple clades co-circulating within a single bat roost on the same day and over consecutive years. The diversity and temporal stability of co-circulating clades suggest that viral dynamics are driven by episodic shedding of existing lineages maintained at the population level, rather than immune-driven strain-replacement dynamics. HeV isolates of different clades displayed variation in phenotypic properties but minimal antigenic differences. We provide an overview of evolutionary dynamics, phenotypic properties and assessment of countermeasures for HeV, and provide insights into the processes that maintain virus diversity in bats and influence the potential for viral emergence.

Bat populations host diverse communities of viruses[1,2], yet we have limited understanding of how viral diversity is structured spatiotemporally[3]. Understanding these ecological and evolutionary patterns could provide key insights into how viruses are maintained in bat populations and which factors drive their emergence in new hosts[4]. Hendra virus (HeV, genus *Henipavirus*) provides an ideal system for addressing this gap[5]. HeV is a non-segmented, negative-strand RNA virus with an 18-kb genome encoding six proteins: nucleocapsid, phosphoprotein, matrix protein, fusion glycoprotein, attachment glycoprotein and large polymerase. The virus circulates endemically in Australian flying foxes (*Pteropus* spp.), with high case fatality rates after spillover into horses and humans (75% and 57%, respectively)[6–11]. In the past 31 years[12,13], 68 spillover events into horses have been documented[14,15], including two spillovers of a novel genotype (HeV-g2) discovered by sentinel surveillance in 2021[16–18].

Australian flying foxes are the natural reservoir hosts of HeV, with virus detected in all four flying fox species: *P. alecto*, *P. conspicillatus*, *P. scapulatus* and *P. poliocephalus*. However, *P. alecto* is the primary reservoir in southeast Queensland and northeast New South Wales (NSW), where most spillovers have occurred[18–21]. These bats are nomadic and nocturnal foragers, moving large distances between roost sites[22–24] that comprise hundreds to thousands of individuals[15,25].

The factors contributing to HeV shedding in flying foxes and spillover dynamics into horses have been extensively studied[14,15,26–28], but the evolutionary dynamics of HeV remain underexplored.

So far, HeV genomic data remain sparse (10 genomes from horses, 2 from humans and 4 from *Pteropus*), with the majority consisting of incomplete sequences[20,21,29]. Understanding HeV's spatiotemporal genetic variability is essential for elucidating its evolutionary dynamics and the emergence of strains. Here we present a genome-scale phylodynamic analysis of HeV-g1 in Australia. By analysing 73 whole genomes spanning 29 years, we delineate circulating clades in fruit bats and reconstruct their temporal and spatial evolution, providing key insights into HeV evolutionary ecology.

## Results

### HeV-g1 whole genome sequencing

Of 9,869 *Pteropus* urine samples screened by PCR from 24 roosts (Supplementary Table 1 and Extended Data Fig. 1), 703 contained HeV-g1 with cycle threshold ($C_t$) values ≤ 40, and 253 of these had $C_t$ values ≤ 32, suitable for sequencing. Isolation success from PCR-positive bat samples was low (3/177; 1.7%), with full genome sequences obtained from the three isolates ($C_t$ values of 24, 35 and 37). From these three isolates, we were able to generate useful infectious stocks from the first two. In total, we generated 48 HeV-g1 full genome sequences (68.0% success) with a mean coverage depth of 415× (range, 8.0× to 3,700×) from bat urine samples with $C_t$ values between 21 and 32 ($7.4 \times 10^6$ to $8.3 \times 10^3$ genome copies per ml) (Supplementary Data 1). This included 4 genomes from *P. alecto* individuals and 44 genomes from under-roost sampling sheets, sampled from roosts spanning 595 km along the coastline of eastern Australia (~12,000 km²). Host genomic sequences detected within the sequenced libraries confirmed *P. alecto* as the source of these samples (Supplementary Data 2). We also generated a further 9 HeV genomes from spillover events in horses.

Within the combined dataset of 73 HeV-g1 sequences, genetic variation across the genome was relatively low, with dips in identity mostly restricted to the intergenic regions: the mean nucleotide identity was 96.9% (range, 47–100%) with the lowest values observed at the 3′ end (Extended Data Fig. 2). Significant differences in HeV genome percent identity ($P < 2.2 \times 10^{-16}$) existed between hosts. Notably, bat-derived genomes exhibited significantly lower identity values compared with both horse and human isolates ($P < 0.0001$), while horse and human genomes did not differ significantly ($P = 0.882$). Base composition was mostly stable across the genome (GC content of 39.8%), except for AT-rich regions at genome ends and intergenic regions.

### Phylogenetics and circulating diversity of HeV-g1 in Australian *Pteropus* spp

The phylogenetic analysis of all 73 HeV-g1 genome sequences from humans, horses and fruit bats revealed four well-sampled monophyletic clades of bat-derived sequences tentatively named clades A, B, C and D (Fig. 1). All four clades included HeV-g1 sequences detected in horses (Fig. 1a, Extended Data Fig. 3 and Supplementary Fig. 1). A few cryptic sequences not belonging to clades A to D were detected in horses and humans between 1994 and 2008, during a period when bat sampling was limited (Fig. 1b and Extended Data Fig. 4). Clade- and subclade-defining amino acid (aa) substitutions rarely occurred in the G glycoprotein (Fig. 1a and Extended Data Table 1), suggesting strong conservation of host entry receptors, ephrin B2 and ephrin B3, across the three hosts[30].

In addition to generating complete genomes, we sequenced 13 additional G glycoprotein sequences and retrieved 6 G glycoprotein sequences from GenBank (Supplementary Data 1). To verify the consistency of the phylogenetic topology, we used both the coding sequences and a more comprehensively sampled G glycoprotein gene dataset to construct the HeV-g1 phylogeny (Supplementary Fig. 2). The trees constructed using the G glycoprotein sequences showed the same topology and clustering patterns as the complete genome tree, and no significant amino acid substitutions were observed in G across these expanded datasets.

Sequences belonging to the four main clades co-circulated in time and space in the bat population. Each clade comprised sequences from flying fox roosts in both NSW and Queensland (Extended Data Fig. 3 and Supplementary Fig. 1), and at least three of the four clades co-circulated in each year of our intensive field sampling (2017–2020) (Fig. 1b). The largest number of genomes were recovered in 2017 and 2018 between June and September, corresponding to the peak of HeV shedding within the natural reservoir during the Australian winter months (Fig. 1c,d and Extended Data Fig. 4)[28]. Clade A comprises eight bat strains and two horse strains from the 2009 and 2014 Queensland spillovers. The 2014 horse strain forms a monophyletic cluster with the bat sequences, whereas the 2009 horse strain is more divergent, sharing 99.3–99.8% nucleotide identity with the clade. Clade B contains two subclades: one represented by a single 2020 NSW bat strain (99.5–99.7% identity to the rest), and a main subclade (99.8–100% identity) consisting of ten bat strains and one 2023 horse spillover strain from NSW. Clade C includes nine bat strains and one horse strain from a 2013 NSW spillover, all showing 99.6–99.7% identity. Clade D, the largest, contains early bat sequences from 2009[20] and later bat genomes sampled from 2017 onwards, as well as four horse spillover strains: two from Queensland (2011, 2017) and two from NSW (2011, 2015). We did not identify any examples where we could directly link a spillover strain to the sampled bat diversity (both temporally and by sequence identity); however, only 20 of 89 known horse spillovers had sequences available for inclusion in the phylogeny.

Overall, the HeV-g1 phylogeny showed a spatially diffuse structure with a high diversity of strains maintained in the bat population. To formally assess the extent of physical movement of HeV among geographic regions, we applied a Bayesian stochastic search variable selection (BSSVS) framework to the genome (codon) dataset. This analysis tests whether directional transitions between regions are statistically supported, thereby identifying potential migration pathways of the virus across bat populations. A small number of transitions were statistically supported as originating in southeast Queensland (Supplementary Tables 2 and 3), yet these involved very limited data ($n = 2$ genomes) and probably reflect artefacts of sparse sampling. Beyond this, no additional migration routes were strongly supported, consistent with the spatially diffuse phylogeographic structure. Strong mixing between roosts was further evidenced by the co-circulation of multiple strains, both within and across clades, within the same roosts, observed concurrently and across consecutive years (Figs. 1c,d and 2). Within each clade, detections spanned 500 km, 650 km, 172 km and 1,535 km for clades A to D, respectively (or 380 km and 172 km for clades A and B, respectively, if only bat sequences are considered). We detected an average of 2.2 clades per roost.

### Dating the emergence of HeV-g1 clades

HeV-g1 sequences demonstrated a strong temporal signal, indicative of clock-like evolution (Supplementary Fig. 3). Linear regression revealed a strong correlation between root-to-tip genetic distance against time of sampling for complete genome and coding sequences (correlation coefficients: 0.78 and 0.74, respectively). While G glycoprotein showed a weaker temporal signal (correlation coefficient: 0.53), this may be due to reduced phylogenetic signal in this region.

We next used a Bayesian Markov chain Monte Carlo (MCMC) approach to more rigorously assess the evolutionary dynamics of HeV. Based on the best-fit model (Supplementary Table 4), the coding sequences showed a refined estimate for the mean rate of evolution of $1.43 \times 10^{-4}$ (95% highest posterior density (HPD), $1.07 \times 10^{-4}$ to $1.80 \times 10^{-4}$) substitutions per site per year. The time to the most recent common ancestor (TMRCA) estimate was 1975 (95% HPD, 1954.8 to 1988.4)

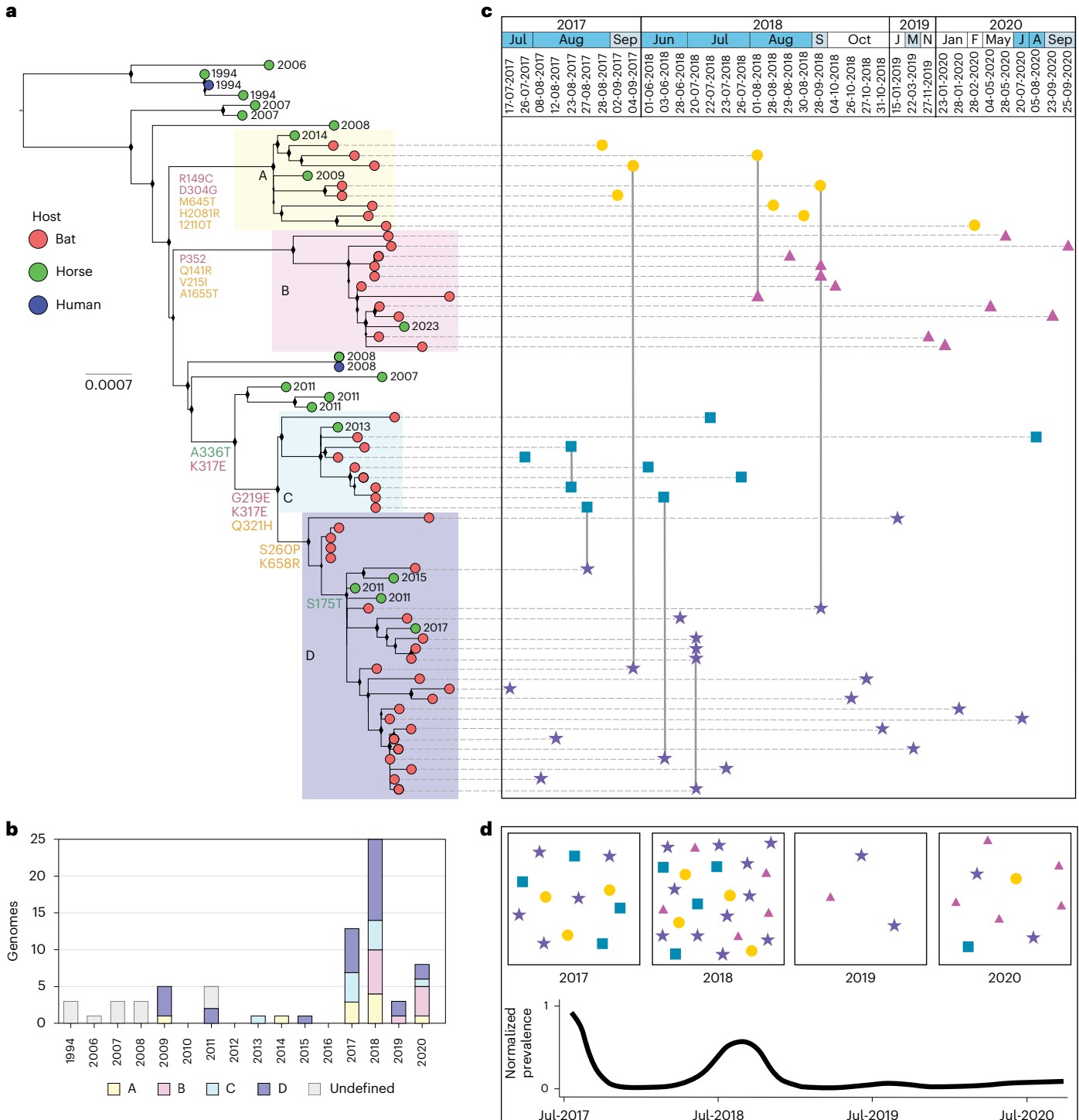

**Fig. 1 | HeV-g1 phylogeny and clades circulating in Australia between 1994 and 2023. a**, Phylogenetic analysis of HeV full genome sequences from bats, horses and humans. Of the 73 genomes used in the phylogeny, 57 were generated in this study. The phylogenetic tree was inferred using the maximum-likelihood method implemented in IQ-TREE v.2.2.0. The best-fit substitution model was selected automatically by ModelFinder (TN + F + I), and branch support values were calculated using 1,000 ultrafast bootstrap replicates. The scale bar is proportional to the number of nucleotide substitutions per site. Tip colours indicate the host species from which each sequence was obtained, branch annotations show clade-defining sets of substitutions (coloured according to the affected viral protein), and background shading denotes the distinct circulating clades. Year of spillover event is indicated at the end of the tip. **b**, Number of HeV genomes from each clade detected in Australia in each year from 1994 to 2023 noting uneven sampling effort over time. **c**, Timing of HeV genome detections in

flying foxes from 2017–2020, coloured by clade (A, yellow circle; B, pink triangle; C, blue square; D, purple star). Sampling years, months and dates are presented at the top of the panel, with months of peak HeV shedding shaded (dark blue, winter months (June, July, August); light blue, adjacent months (May and September)). Vertical alignment of symbols represents their position within the HeV phylogeny, and horizontal alignment represents timing of sampling. Genomes that are joined with a vertical grey line were detected within the same sampling session. **d**, Boxes show the diversity and frequency of genomes detected within each shedding pulse (following ref. 5), with the black line showing smoothed normalized HeV prevalence within the broader sample set across the 4 years, using the $C_t$ threshold we used for sequencing, $C_t ≤ 32$ (following ref. 28). The bat and horse derived HeV genomes generated here were combined with 16 previously published sequences from NCBI GenBank (Supplementary Data 1).

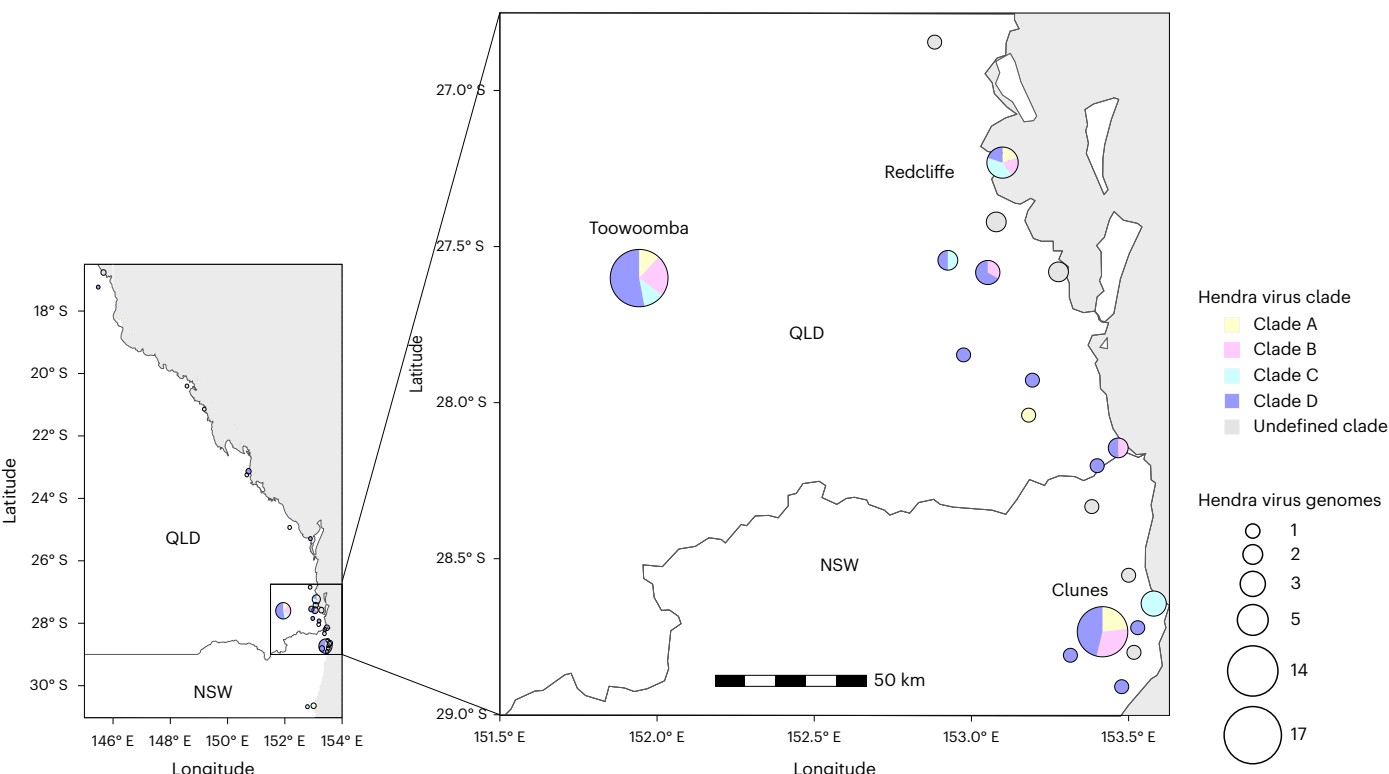

**Fig. 2 | Spatial distribution of HeV-g1 clades in subtropical Australia.** The cumulative total distribution of the identified clades at specific roost. The size of the pie is proportional to the number of full genome sequences recovered from each roost site. Maps were created in R (v.4.3.0)[72]. QLD, Queensland; NSW, New South Wales.

across the whole tree, which predates the first known HeV outbreak in humans by several years, highlighting a clear lag between viral evolution in bats and recognition as a cause of spillover (Fig. 3). Within this broader timescale, the four major clades (A to D) each coalesce to a common ancestor around a decade before the most recent detections (mean 13.2 years, range 11.9–13.9), giving a consistent measure of clade depth. However, the inferred divergence of clades A (~1994) and B (~1996) is >15 years before the first available sequences, highlighting clear undersampling. By contrast, clades C and D diverged more recently (~2005) and have been more continuously sampled since their divergence (Fig. 3).

Several spillover-only lineages fall outside the four major clades. The dating analysis places their divergence more than two decades before the spillover detections, further highlighting the long periods of unsampled diversity and evolutionary histories for HeV in Australian bats. Similarly, some spillover viruses are found within clade C and D (temporally 2002–2005), yet no related bat-derived viruses have been linked to them so far. Before this study, clade B-related viruses had not been detected. We performed a permutation analysis to determine whether this absence, in the face of existing sampling limitations, could be explained by chance alone. The probability of failing to observe clade B before 2018, assuming it was indeed present throughout the sampling periods, was 0.022 (details in Supplementary Results and Extended Data Fig. 5). This supports the notion that clade B probably represents a recent lineage rather than a long-established clade missed due to undersampling.

### Structure prediction and potential effect of clade-defining amino acid substitutions

We evaluated the functional relevance of all recurrent substitutions in HeV proteins, emphasizing the 16 lineage-defining (LD) substitutions (Fig. 1a). Structural context was inferred using experimental and predicted models and literature describing functional regions. Detailed functional profiles are described in the Supplementary Discussion.

LD substitutions are denoted as [Clade]:[Residue][Site][Residue]. Table 1 summarizes the likely functional substitutions.

Both LD substitutions in the G glycoprotein, D:S175T and C/D:A336T, are distant from ephrin B2/B3 binding and oligomerization interfaces (Fig. 4a), although S175T overlaps a mapped T-cell epitope[31]. The LD substitutions in P/V/W (A:R149C, C:G219E, B:P352S, A:D304G, C:K317E) lie within their common intrinsically disordered region (IDR; Fig. 4b, Supplementary Fig. 4 and Extended Data Fig. 6), which mediates interactions with host immune effectors[32] and drives phase separation[33]. Most LD substitutions occur in the L protein, but only three appear non-silent (C:Q321H, A:H2081R, A:I2110T) (Fig. 4c). Remarkably, 10 of the 16 LD substitutions encode the residues present in HeV-g2 and/or NiV (Supplementary Table 5), suggesting conserved constraints or unknown regulatory motifs.

### In vitro HeV phenotypic characterization

To compare the phenotypic characteristics of HeV from the different clades, we assessed the replication fitness, innate responses, IFN-I sensitivity and neutralization capacity of the two isolated HeV strains, HeV/*Pteropus* spp./Australia/RML-071/2018 (Hendra 71, clade D) and HeV/*Pteropus* spp./Australia/RML-084/2018 (Hendra 84, clade B) compared to the prototype HeV strain (HeVAustralia/horse/1994) and the non-pathogenic Cedar virus (CedV, CedV/*Pteropus* spp./Australia/AUSRML1/2021 (ref. 34)) on cell lines from multiple species, where applicable.

In human (HFL-1) and horse (NBL-6) cells, the replication kinetics of the prototype HeV strain and Hendra 71 were indistinguishable, while replication of Hendra 84 was attenuated (Fig. 5a). In human cells, Hendra 84 reached similar titres as the other two strains at 96 h post infection (hpi), while in horse cells, it remained attenuated throughout the time course (Fig. 5a). All three HeV strains replicated on *P. alecto* cells (PaKiT) with similar replication kinetics (Fig. 5a). At 24 hpi, CedV grew to similar titres as the HeV prototype on all cell lines but was attenuated significantly at later time points (Fig. 5a).

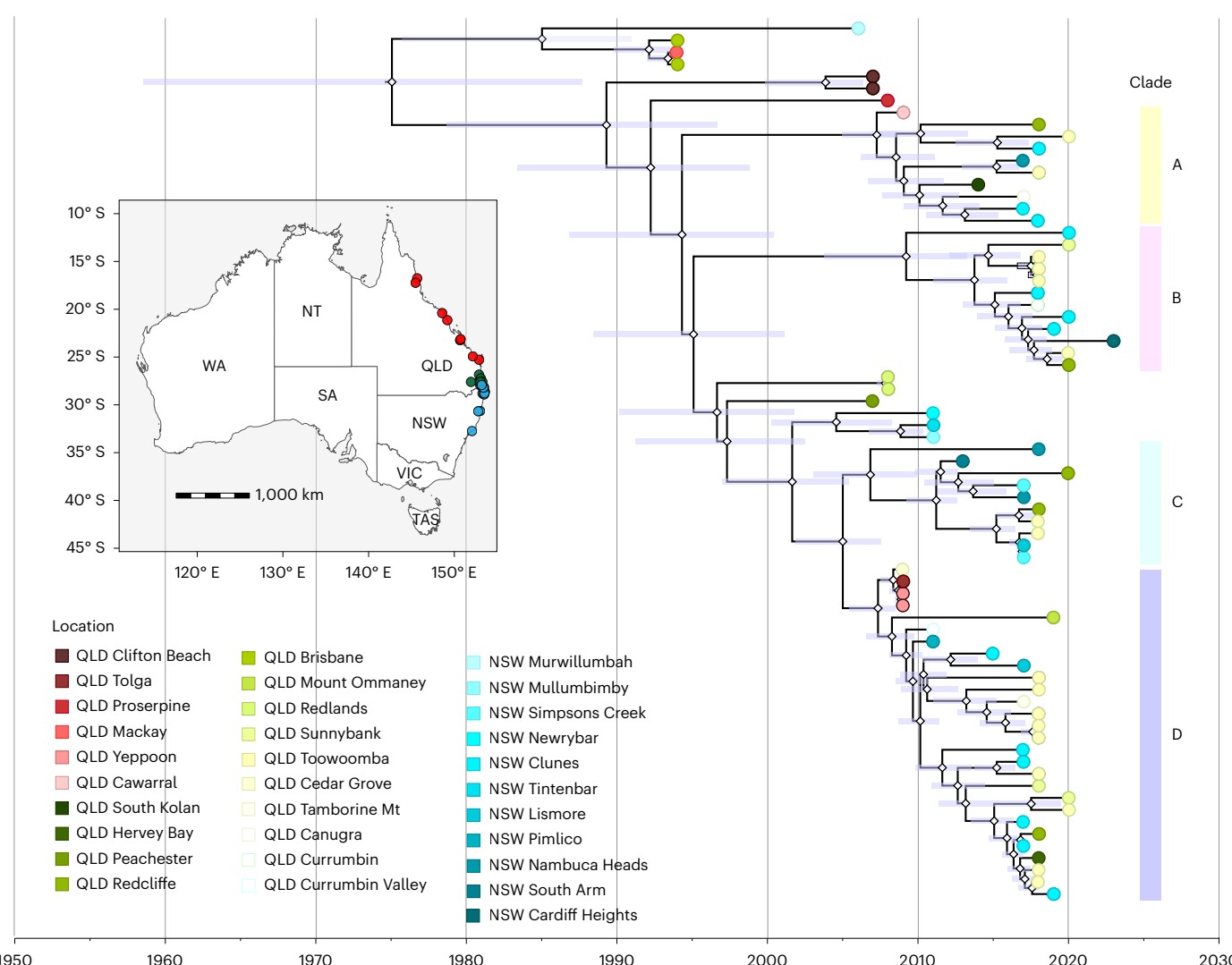

**Fig. 3 | Bayesian phylogenetic inference of HeV-g1.** A Bayesian tree was constructed using a relaxed clock and Skygrid coalescent model, incorporating all available Hendra virus genomes from horses, humans and bats. Tip colours represent sample location and horizontal node bars indicate 95% HPD intervals. Node shapes (diamond) represent the Bayesian posterior distribution. HeV-g1 clades circulating in Australia between 1994 and 2020 are indicated. Map shows sampling location (red, north and central Queensland; green, southeast Queensland; blue, north New South Wales; darker colours indicate sampling locations close to the border between Queensland and New South Wales. Maps were created in R[72].

The innate responses at 72 and 96 hpi were evaluated to determine whether the strains differed in their capacity to antagonize the type I interferon response. In HFL-1 cells, Hendra 84 induced lower levels of Ifnb1 mRNA transcripts than HeV prototype, but IFN-I stimulated genes (ISGs), *Ifit1* or *Mx1*, were higher or equivalent at 72 (*Ifit1* and *Mx1*) and 96 (*Mx1*) hpi, suggesting that Hendra 84 may not be as efficient at blocking IFN-I signalling (Fig. 5b and Extended Data Fig. 7a). *Il-6* was also less efficiently induced in Hendra 84-infected human cells (Fig. 5b and Extended Data Fig. 7a). In NBL-6 cells, *Ifnb1* and *Il-6* were similarly induced for all HeV strains despite the significantly lower viral titres for Hendra 84, suggesting impaired IFN-I induction antagonism in horse cells (Fig. 5c and Extended Data Fig. 7b). The ISG transcripts were induced poorly in NBL-6 cells at both time points, potentially due to replication being below the threshold to induce a robust IFN-I signalling response. Further, CedV-induced expression of all transcripts evaluated more strongly than the HeV strains, supporting that CedV does not effectively antagonize horse IFN-I induction or signalling. In PaKiT cells, the Ifnb1 transcripts were higher for Hendra 71 (72 and 96 hpi) and Hendra 84 (96 hpi) than for prototype HeV despite similar viral loads (Fig. 5d and Supplementary Fig. 10c). Il-6 transcripts followed a similar pattern. At both time points, the transcripts for both

ISGs were higher for Hendra 71 and 84 than for prototype HeV (Fig. 5d and Extended Data Fig. 7c). As observed in horse cells, all transcripts were induced the strongest in CedV-infected bat cells. Protein expression data assessed with western blot also supported that IFN-I signalling was induced in response to CedV, based on the increased expression of ISG *IFITM3* and total *STAT1* (Fig. 5e). Induction of ISGs did not vary much among the different HeV strains in any of the cell lines (Fig. 5e).

To compare the IFN-I sensitivity of the HeV strains, Vero cells were treated with universal IFN-I (U-IFN) at 5-fold serial dilutions before infection. All HeV strains showed sensitivity to U-IFN, but the higher doses appeared to attenuate the prototype HeV strains more effectively than the HeV strains isolated from bats (Fig. 5f). The effect of IFN-I on HeV infection appeared to level off for HeV prototype-infected cells treated with 500, 100 and 20 ng ml⁻¹ of U-IFN, suggesting that expression of ISG effectors is not sufficient to fully neutralize HeV replication.

To assess the antigenic relationship between the isolates (HeV prototype, Hendra 71 and Hendra 84) and the efficacy of the Equivac HeV vaccine, we performed neutralization assays with horse serum from Equivac HeV vaccine vaccinated horses[35]. All HeV strains were effectively neutralized by the vaccinated horse serum, with comparable neutralizing activity among the different strains (Extended Data Fig. 7d).

**Table 1 | Potential functionally relevant amino acid substitutions in HeV proteins**

| Protein | substitution | Predicted functional relevance | Type / clade | Key analysis |
|---|---|---|---|---|
| P/V/W | R149C | Within STAT1-binding region; could modulate IFN antagonism. | LD / A | STAT1-binding assays[86] |
| P/V/W | G219E | Within amyloidogenic motif; could influence fibrillization, proposed to be linked to immune evasion. | LD / C, D | Amyloid formation studies[33] |
| P/V/W | D304G | Within amyloidogenic motif; could influence fibrillization and immune evasion. | LD / A | Amyloid formation studies[33] |
| P/V/W | P352S | Alters local flexibility; flanks a conserved helical segment predicted to be a protein-binding motif. | LD / B | Structural modelling (Supplementary Discussion) |
| P/V/W | H139Y | Within STAT1-binding region; could modulate IFN antagonism. | non-LD / D | STAT1-binding assays[86] |
| P/V/W | R170G | Near exportin-1-binding region; could influence subcellular localization. | non-LD / A to D | Exportin-1 binding assays and mutagenesis[32] |
| F | A141T | Lies within the heptad repeat A; could influence fusion efficiency. | non-LD / D | Mutagenesis assays[87] |
| G | S175T | Located within a predicted T-cell epitope; may alter antigenicity or immune evasion. | LD / D | NiV immunogenic epitope mapping studies[32] |
| G | R248G | Near K246, which plays a critical role in fusion triggering; could affect syncytium formation and fusion efficiency. | non-LD / C | NiV G-F-binding screening and mutagenesis[88] |
| G | N306K | *N*-glycosylation site; probably affects fusion activity | non-LD / B | Mutagenesis assays[89] |
| L | Q321H | Adjacent to putative nucleotide NTP entry channel; could influence NTP affinity. | LD / C, D | Comparative structural analysis with cryo-EM structures of the NiV RNA polymerase[90,91] |
| L | H2081R | Near conserved regions involved in RNA capping; could indirectly influence RNA capping performance. | LD / A | Comparative structural analysis with cryo-EM structures of the NiV L[90,91] |
| L | I2110T | Near conserved regions involved in RNA capping; could indirectly influence RNA capping performance. | LD / A | Comparative structural analysis with cryo-EM structures of the NiV L[90,91] |

Detailed discussion is provided in the Supplementary Material.

## Discussion

Understanding how viral populations are structured and maintained within bats has been constrained by limited genomic data, in turn limiting our ability to predict and mitigate spillover risk from zoonotic viruses such as HeV[14,15]. Through extensive spatiotemporal sampling and viral-enrichment sequencing, we generated 48 new HeV genomes from bats, together with nine genomes from horses, expanding the available dataset nearly 5-fold (from 16 to 73). Our analyses revealed the presence of four distinct HeV clades co-circulating in bats, with no distinct spatiotemporal structuring or evidence of immune-driven strain evolution, providing insights into viral maintenance in bat populations.

Epidemiological theory predicts that selective pressures should drive viral lineage turnover, with dominant strains periodically replacing others, producing clear temporal or spatial clustering and reducing diversity[36]. We previously hypothesized that phylogenetic signatures could distinguish between competing models of HeV infection dynamics in bats[5]. Acute immunizing infections ('SIR dynamics', that is, susceptible–infectious–recovered) should yield low within-outbreak but high between-outbreak diversity, whereas longer infections ('SILI dynamics', that is, susceptible–infectious–latent–infectious) or partial cross-immunity could generate greater within-outbreak diversity. Our genomic dataset now permits partial evaluation of these hypotheses. Our results are inconsistent with the rapid lineage turnover or immune-driven evolution expected under acute infection with immune escape (flu-like dynamics). Instead, the co-circulation of divergent clades, minimal variation in the G glycoprotein and the absence of strong selection signals indicate limited immune-mediated lineage replacement. This broadly resembles measles virus phylodynamics where strong, long-lasting cross-immunity stabilizes lineages and persistence arises through epidemiological rather than immunological processes[36]. However, the underlying mechanisms are probably different, since current evidence suggests that infections in bats are characterized by weak or waning immunity rather than strong sterilizing immunity, making classical SIR dynamics unlikely. Importantly, both short-lived immunity ('SIRS dynamics',

that is, susceptible–infectious–recovered–susceptible) and low-level within-host persistence (SILI) can generate stable, multiyear clade co-existence. With no longitudinal resampling of individual bats, we cannot currently discriminate between these mechanisms. Although explicit phylodynamic modelling and within-host longitudinal data will ultimately be required to separate SIRS from SILI dynamics, our results nonetheless support population-level stability of HeV lineages and suggest that ecological factors, such as food stress or host condition, could drive episodic shedding.

This stable co-existence of divergent HeV lineages within *Pteropus* bats contrasts strongly with that of bat-borne rabies virus, where the phylogeny is structured by host species, with obvious temporal signal reflecting an evolutionary history of host shifts followed by predominantly within-species transmission[37–40]. However, our observations are somewhat consistent with those in the closely related NiV[41]. Two distinct NiV genogroups have been described, comprising four minor genotypes and a total of 15 genetic 'clusters'[42]. These genetic clusters overlap within the same country, representing an average of 2.41 genetic clusters per roost (analogous to estimates of 2.2 clades per roost here)[41]. However, some NiV genetic clusters spanned >2,100 km, comparable to observations within our study area[41]. While it remains possible that HeV might exhibit spatial structuring at continental scales beyond our study region, the comparable fine-scale patterns to those of NiV studies suggest that our findings reflect genuine ecological and evolutionary dynamics rather than sampling limitations[41].

Understanding the functional consequences of genetic diversity across co-circulating clades is essential for assessing their public health and therapeutic significance. Structure-based analysis identified several amino acid substitutions that may influence HeV phenotype and host interactions (Table 1, Fig. 4 and Supplementary Discussion). Many occur near known or predicted key regions involved in viral processes, including replication efficiency and immune modulation. Supporting this, our in vitro experiments revealed phenotypic differences linked to non-silent substitutions.

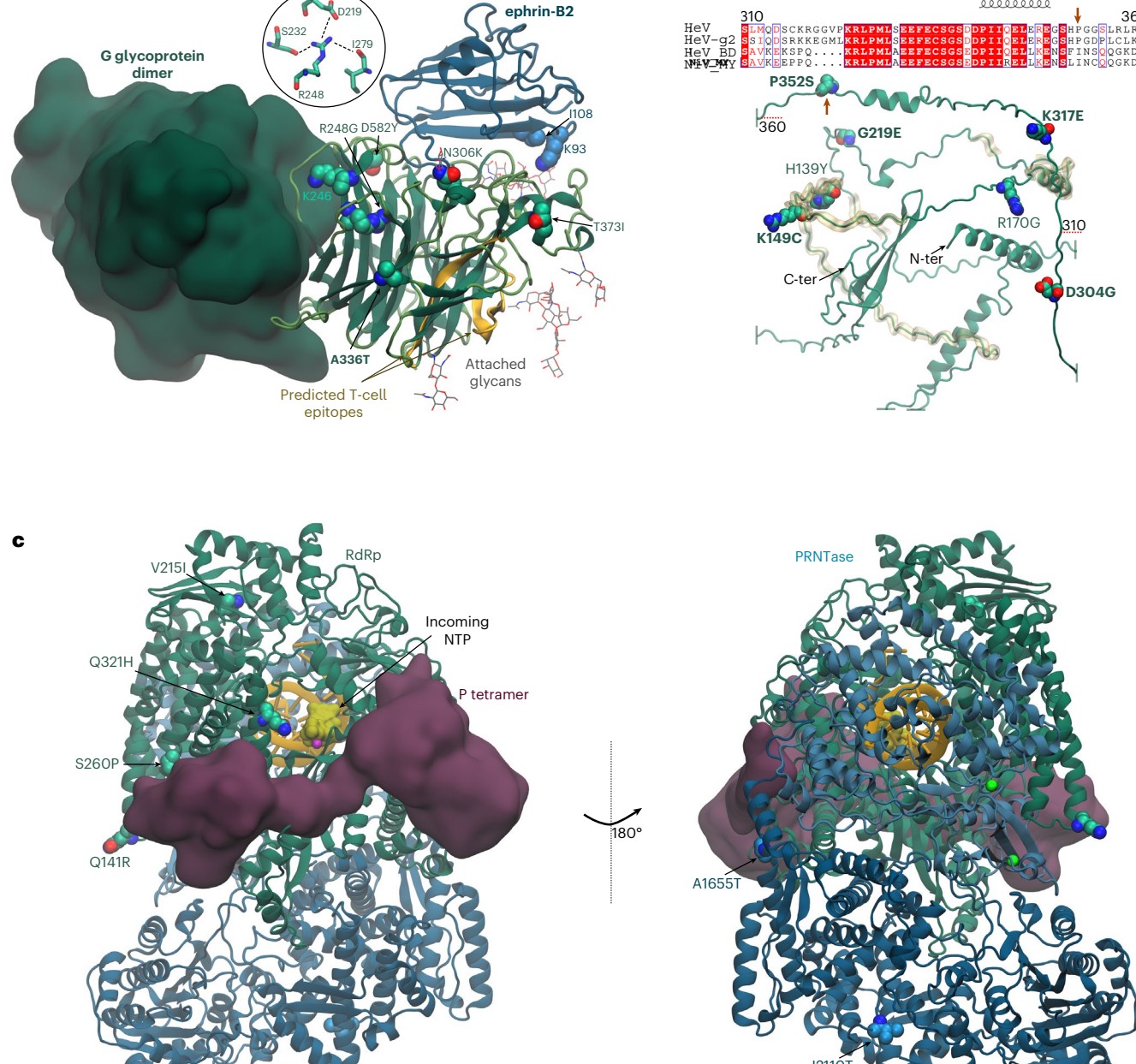

**Fig. 4 | Potential functional effects of amino acid substitutions in HeV proteins G, P/V/W and L. a**, Substitutions in the head domain of the HeV G glycoprotein are depicted as van der Waals spheres. The C/D-defining substitution A336T is highlighted in bold. Other recurrent substitutions in G are also depicted. Details of local interactions affected by the non-lineage-defining substitutions R248G (clade C: ARRED004_AVL_U_23_1) are shown: R248 forms hydrogen bonds with S232 and I279, and a salt bridge with D219. The substitution to glycine will disrupt these interactions and local secondary structure. The human ephrin-B2 receptor is represented as blue ribbons (PDB ID: 2VSK)[92], with residues K93 and I108 highlighted. The equivalent amino acids are K93/V108 in horse and T93/V107 in bat (*P. alecto*). The neighbouring protomer is shown as a dark-green surface (PDB ID: 2X9M)[93], and attached glycans are shown from PDB ID: 7SYY (ref. 94). Most substitutions in G are not predicted to perturb receptor binding or dimer/tetramer interface contacts (PDB ID: 8K0C and 8K0D)[88]. **b**, Recurrent substitutions in the HeV V are mapped onto an AlphaFold model and shown as van der Waals spheres. The lineage-defining (LD) substitutions, including P352S in clade B, are highlighted in bold. Regions of the intrinsically disordered domain known or predicted (by comparison with NiV) to mediate host interactions

are indicated with a yellow transparent surface, including recognition regions for STAT1 (aa 101–160), exportin-1 (aa 174–192), PLK1 (aa 199–201) and STAT2 (aa 231–236)[32]. The sequence alignment of V proteins from HeV, HeV-g2, NiV-Bangladesh (NiV$_{BD}$) and NiV-Malaysia (NiV$_{MD}$) in the aa 310–360 region shows high conservation near position 352 (red arrow), consistent with functional constraint at this site. The helical propensity in that region is indicated. **c**, LD substitutions in the HeV L protein are depicted as van der Waals spheres on its AlphaFold model. The RNA-dependent RNA-polymerase (RdRp) domain is coloured green; the polyribonucleotidyl-transferase (PRNTase) light blue; and the connector domain (CD), methyltranferase (MTase) and the C-terminal domain (CTD) domain dark blue. The model was superimposed with the NiV RNA polymerase complex cryo-EM structure (PDB ID: 9GJU)[90], showing the region of L- (P-tetramer) interaction (dark mauve surface), the RNA template/product in the active site (yellow cartoon) and an incoming nucleotide triphosphate (NTP, yellow surface). The LD substitutions M645T and K658R are not displayed because they lie in a flexible loop in RdRp that lacks resolved density in the reference NiV L structure. In our AlphaFold model, the loop overlaps with the P tetramer.

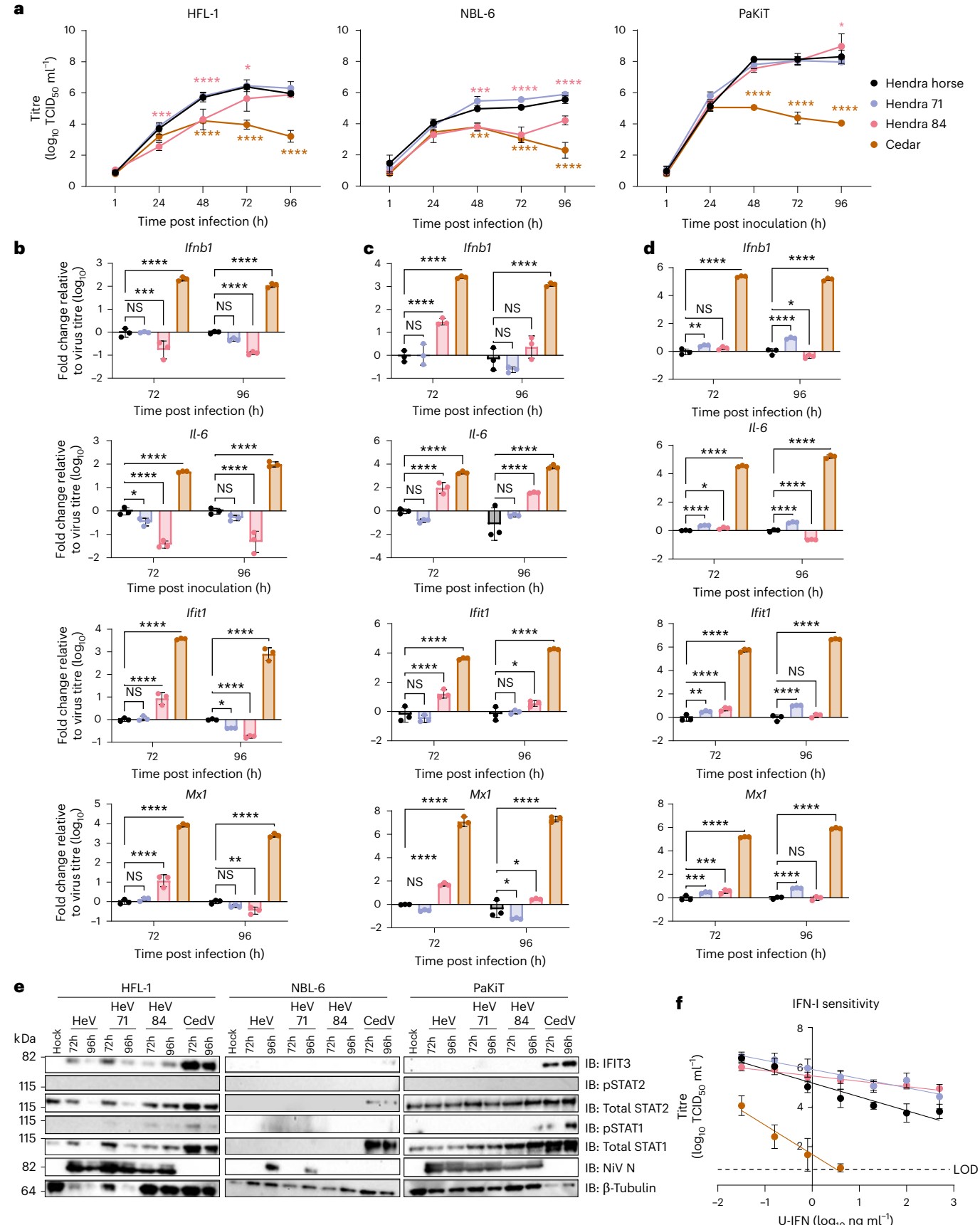

**Fig. 5 | Phenotypic differences of Hendra virus in human and horse cells.**
**a**–**e**, Human (HFL-1) (**a**,**b**,**e**), horse (NBL-6) (**a**,**c**,**e**) and black flying fox (PaKiT) (**a**,**d**,**e**) cells were infected with a multiplicity of infection (MOI) of 0.01 of Hendra virus prototype (Hendra virus/Australia/horse/1994), clade B (Hendra virus/*Pteropus* spp./Australia/RML-084/2018, 84), clade D (Hendra virus/*Pteropus* spp./Australia/RML-071/2018, 71) or Cedar virus (Cedar virus/*Pteropus* spp./Australia/AUSRML1/2021). **a**, Supernatants were collected 1 h after infection and every 24 h after, then titrated. Limit of detection (LOD), 0.8 TCID$_{50}$ ml$^{-1}$. $n = 3$, mean ± s.d. shown, representative of 2 independent experiments. For each cell line, each time point was compared to Hendra horse-infected cells using two-way analysis of variance (ANOVA) with Tukey's multiple comparisons test. **b**–**d**, RT–qPCR of host genes normalized to the virus titre. $\Delta C_T$ values were normalized to the average $\Delta C_T$ value of mock infected cells to calculate $\Delta\Delta C_T$, then divided by the virus titre. The $\Delta\Delta C_T$/TCID$_{50}$ ratio was normalized to Hendra prototype-infected cells then log$_{10}$ transformed. Fold change of each gene at each time point was compared to Hendra horse-infected cells using two-way ANOVA with Dunnett's multiple comparisons test. $n = 3$, mean ± s.d. shown, representative of 2 independent experiments. **e**, Immunoblot of protein lysates collected at 72 and 96 hpi. The panel is representative of 3 independent western blots from 2 independent experiments. Source data. **f**, Virus titres from Vero CCL-81 cells treated with increasing doses of U-IFN for 24 h before infection with MOI 0.01 for 24 h. $n = 6$, mean ± s.d. shown, data from 2 independent experiments. (**a**–**d**) $P$ values for comparisons to mock cells are indicated: ****$P < 0.0001$, ***$P < 0.001$, **$P < 0.01$, *$P < 0.05$; NS, not significant.

Hendra 84, belonging to clade B, exhibited lower fitness than the HeV prototype and the clade D Hendra 71. Conversely, Hendra 71 and the prototype showed similar profiles, suggesting that most clade D substitutions (Table 1) have limited impact in vitro. Subtle or context-dependent effects remain possible; for example, Hendra 71 also carries G S175T within a NiV T-cell epitope, whose effects would not appear in assays lacking adaptive immunity.

Attenuation of Hendra 84 in horse and human cells through at least 72 hpi, before induction of the innate immune response, suggests that this reduced fitness results from viral factors that affect replication across different host cell types. Further, in all cell lines, Hendra 84 showed elevated ISG expression relative to viral load compared to other strains, suggesting reduced efficiency in blocking IFN-I induction and/or signalling. This impaired IFN-I antagonism, evident even in natural reservoir host cells, may strongly influence viral fitness in vivo, although Clade B sequences are still capable of fatal infections in horses. The clade B-specific P352S substitution in P/V/W offers a plausible mechanism, as P is part of the RNA polymerase complex and V/W antagonizes IFN-I signalling (Fig. 4b, Supplementary Fig. 4 and Supplementary Discussion).

All three HeV strains suppressed the innate immune response more effectively than CedV, consistent with CedV's reduced pathogenicity[34]. Although the HeV strains exhibited some sensitivity to IFN-I at pre-treatment, even the highest doses did not fully block replication. Future work examining IFN-I responsiveness in cell lines from different species may help identify the ISGs most effective against HeV. Collectively, these phenotypic assessments reveal that Hendra 71 resembles the prototype strain, whereas Hendra 84 exhibits distinct, species-specific replication and immune-response characteristics. This indicates that the clades from which these viruses originate differ not only genetically but also phenotypically, underscoring functional diversity among HeV lineages.

Importantly, neutralization assays showed minimal antigenic differences among the three evaluated HeV clades, and the Equivac vaccine elicited comparable neutralizing antibody titres to all, suggesting that it would be equally efficacious across HeV clades[35].

Alongside the discovery of HeV-g2 (refs. 17,18), our results suggest that our understanding of the true diversity of HeV and related viruses in flying fox populations is incomplete, and intensified surveillance endeavours would probably uncover more divergent strains and variants. The stable maintenance of diverse viral clades over consecutive years, combined with minimal evidence of immune-driven selection, suggests long-term viral persistence with episodic shedding rather than epidemic waves with strain replacement. Most HeV-positive samples had low viral loads, limiting full genome recovery and underscoring the need for large longitudinal studies and viral-enrichment sequencing to generate more genomes for analyses. Finally, the detection of phenotypic differences across HeV clades demonstrates that ongoing surveillance of viral and phenotypic diversification, alongside assessment of countermeasure efficacy, remains critical for spillover prevention, even in well-studied systems.

## Methods

### Institutional Review Board statement
Sample collection from flying foxes was covered under Griffith University Animal Ethics Committee Approval ENV/10/16/AEC and ENV/07/20/AEC and is described in ref. 17. All virus isolates and field samples positive for Hendra virus (HeV) in this study were handled in a high-containment facility at the Rocky Mountain Laboratories (RML), NIH, and were inactivated according to validated and Institutional Biosafety Committee (IBC)-approved protocols before removal for downstream processing. The RML IBC approved work with HeV at Biosafety Level 4 (BSL-4) according to PRD-18-91. The RML IBC also approved work with field specimens at BSL-2 with BSL-3 practices according to PRD-19-158.

### Biosafety
For bats, personnel wore protective clothing, including long sleeves and pants (or long-sleeved coveralls), closed-in shoes and a hat. To protect mucous membranes, all personnel also wore P2/N95 respirators and eye protection. A high-visibility vest was recommended when placing drop sheets to improve visibility in the dark. All samplers had current rabies vaccination, although this was not essential for under-roost sampling, as under-roost sampling does not typically involve close contact with bats. A bat bite, scratch and needlestick injury protocol was provided during sampling sessions, and contact details for the relevant public health authorities in New South Wales and Queensland were supplied to ensure that appropriate advice could be obtained for post-exposure prophylaxis.

Bat samples were treated as diagnostic field samples and stored in a PC2 laboratory at Griffith University prior to shipment to RML. At RML, bat samples were similarly treated as diagnostic field samples and received at an enhanced BSL-2 (BSL-2$^+$) lab using BSL-3 practices. This lab is dedicated to handling field samples and all employees with access to this lab are enrolled in the RML Biosurety Program. Here, samples were inactivated and nucleic acid obtained and screened for HeV. Samples positive for HeV were reported to the Federal Select Agent program and transferred into BSL-4 for further analysis and the field sample room completely decontaminated. In BSL-4, all HeV-positive samples and isolates were handled according to the Federal Select Agent regulations.

For horses, clinical specimens from horses showing symptoms consistent with HeV infection were collected by veterinarians following national and state infection-prevention guidelines. Samples were submitted to the Australian Centre for Disease Preparedness (ACDP) for diagnostic testing. Virus isolation and inactivation of samples for downstream processing were performed in the BSL-4 laboratory at ACDP.

### Sample collection
Briefly, sampling was conducted at flying fox roosts in southeast Queensland and mid- to north-coast New South Wales between December 2016 and September 2020 (Supplementary Table 1 and Extended Data Fig. 1). This area encompasses the distributional overlap of key HeV reservoir species, contains the highest HeV prevalences in

flying foxes and represents a hotspot of spillover risk (encompassing 47 of 68 documented HeV spillover events)[16]. Urine was prioritized for sample collection and screening because HeV demonstrates renal tropism, with urine representing the primary route of viral excretion in naturally infected flying foxes and typically yielding the highest viral loads compared to blood samples from the same individuals[19]. Urine samples were collected both from individual bats captured in mist nets at their roost site (after bats were anaesthetized with isoflurane, starting at 5% then reducing to 1.5%) and from plastic sheets placed underneath roosting trees at pre-dawn[28]. For captured bats, urine samples were collected directly from the bat if it urinated while anaesthetized, or from a urine collection bag attached to its holding bag (the bottom third was lined with plastic). After sample collection, bats were released. Information on species, sex and age class (adult, subadult, juvenile) was also recorded (Supplementary Data 1). For samples collected from sheets, a single pooled urine sample was collected from each sheet, and the number and species of bats immediately above the sheet were recorded. Sheet placement was prioritized to target black flying-fox roosting locations, but other species were also present. For both sample types, urine was pipetted into a tube with AVL lysis buffer (Qiagen, target 140 µl of urine into 560 µl buffer), viral transport medium (VTM, between 200–1,000 µl of urine into 1,000 µl of VTM) or a plain cryovial. Samples were held on ice during collection, then transferred to a CryoShipper (<−80 °C) for transport and stored at −80 °C in the laboratory.

## RNA extraction and HeV screening of bat samples

RNA was extracted using the QIAamp Viral RNA kit using a QIAcube HT automated system (Qiagen) according to manufacturer instructions. RNA was eluted in 150 µl of TE buffer and used for HeV-g1 screening using quantitative PCR with reverse transcription (RT–qPCR) targeting the viral P gene (5′-CCCAACCAAGAAAGCAAGAG (forward), 5′-TTCATTCCTCGTGACAGCAC (reverse) and 5′-TTACTGCG GAGAATGTCCAACTGAGTG (probe)). RNA (10 µl) was tested with a *Taq*Man Fast Virus 1-Step master mix on a QuantStudio 6 Flex Real-Time PCR instrument (Applied Biosystems) according to manufacturer instructions. Samples were deemed positive with genome detection at cycle threshold ($C_t$) value ≤ 40. Spatiotemporal analyses of PCR results generated as part of the same study are described in ref. 28. Positive samples with $C_t$ values ≤ 32 were selected for further whole genome recovery. HeV-g2 was previously detected in a subset of these samples at significantly lower prevalence (0.16%; ref. 17); however, no full genomes could be recovered for phylogenetic analyses, so this study focused exclusively on HeV-g1.

## Virus isolation

Where positive samples had available sample aliquots in VTM (n = 177), virus isolation was performed to enhance genome sequencing success. Vero E6 cells (Vero CCL-81, ATCC) in a 24-well plate were inoculated with 250 µl original undiluted sample and a 1:10 dilution thereof in different wells. Diluted and undiluted samples were plated in duplicates. Plates were centrifuged for 30 min at 200 × *g* and incubated for 30 min at 37 °C and 5% $CO_2$. The inoculum was removed and replaced with 500 µl DMEM medium containing 2% FBS, 50 U ml⁻¹ penicillin and 50 µg ml⁻¹ streptomycin. At 2 and 3 days after inoculation, cytopathic effect (CPE) was scored. Blind passage was performed on samples without CPE as above; other than that, plates were not spun. Supernatants from plates with CPE present were analysed via RT–qPCR for Hendra virus RNA to rule out other causes of CPE. RNA of positive samples was subjected to full genome recovery.

## Partial and full genome recovery of bat HeV-g1

Libraries were prepared using either the HyperPrep DNA library kit, or RNA HyperPrep workflow (Roche Sequencing Solutions). Using the HyperPrep DNA library kit, 11 µl viral RNA was used as template for first-strand complementary (c)DNA synthesis using the Superscript IV First-Strand Synthesis system (ThermoFisher). Second-strand synthesis was performed by combining the RNAse-treated first-strand cDNA product with 20 units exo-Klenow Fragment and 1× NEB Buffer 2 (New England Biolabs), combined with 0.5 mM dNTPs and 3.35 µM random hexamers (ThermoFisher), and then incubated at 37 °C for 45 min. The resulting double-stranded cDNA (ds-cDNA) products were purified with AMPure XP beads following manufacturer-recommended procedure (Beckman Coulter). Sequencing libraries were generated from ds-cDNA following the SeqCap EZ HyperCap Workflow User's Guide v.2.3 (Roche). Briefly, ds-cDNA was sheared in 50 µl microtubes using the Covaris LE220 Focused Ultrasonicator (Covaris) to an average fragment size of 180–220 bp and converted to sequence-ready libraries using the Kapa HyperPrep DNA kit and dual-indexed adapters (Roche).

For the RNA HyperPrep workflow, sequencing libraries were generated from 11 µl viral RNA stock viral RNA using the RNA HyperPrep kit and dual-indexed adapters, according to manufacturer instructions (Roche). Additional PCR cycles were added if final product yields were low.

## Targeted enrichment

1. Probe-based enrichment and sequencing
   To enrich for viral sequences, Kapa libraries from above were quantified using the area under the curve on Bioanalyzer traces as well as UV spectrophotometry on the Nanodrop instrument (ThermoFisher). Pre-capture sequencing libraries were normalized and then combined in 4- to 12-plex reactions for solution capture hybridization using the version of VirCapSeq-VERT probe set described in ref. 43, or custom myBaits probe library. For the VirCapSeq-VERT probe set, Kapa libraires were enriched for virus following the SeqCap EZ HyperCap Workflow User's Guide v.2.3, while for the custom myBaits probe library, the myBaits Hybridization Capture for Targeted NGS protocol v.4.01 was used.

2. PCR-based amplicon enrichment and sequencing
   In addition, PCR-based amplicon enrichment was performed on the samples that did not yield full genomes following the methods above. First, tiled primers were designed using the primal scheme multiplex <FASTA> command line interface following existing methods[44]. Then, single-stranded cDNA was generated as described above and 2.5 µl was used as template[45] in a PCR assay using HeV-specific tiled primers that generated overlapping 400-bp amplicons spanning the HeV genome. Even and odd multiplex PCR conditions/amplicon generation using Q5 HotStart Polymerase PCR (ThermoFisher) was optimized following the ARTIC nCoV-2019 protocol with the following modifications: 2× primer concentration in each even/odd primer pool with 35 cycles at a 52 °C (1 min) annealing temperature and 72 °C (30 s) extension. Products from each primer pool were combined, AmPure XP purified and quantified with Qubit (ThermoFisher) fluorometric quantitation as per instructions. Following visual assessment on a TapeStation HS D1000 (Agilent), a total of 100–400 ng of positive PCR product was taken directly into TruSeq DNA PCR-Free Library Preparation Guide, Revision D (Illumina) beginning with the repair-ends step. Subsequent cleanup consisted of a double 1:1 AmPure XP/reaction ratio, and all steps followed manufacturer instructions including the Illumina TruSeq CD (96) Indexes. Final libraries were visualized on either a TapeStation HS D5000 or BioAnalyzer HS chip (Agilent) and quantified using a KAPA Library Quant kit - Illumina Universal qPCR Mix (Kapa Biosystems) on a CFX96 Real-Time System (BioRad) for PCR-based amplicon enrichment and sequencing.

3. Long-range PCR amplicon enrichment and sequencing
   Positive samples from which full genomes could be recovered by either method above were subjected to a long-range PCR

amplification to recover the attachment (G) glycoprotein. Single-stranded cDNA was generated as described above[45] and long-range semi-nested PCR was performed according to previously established protocol (Supplementary Table 6)[46]. Libraries were prepared as outlined for PCR-based amplicon enrichment and sequencing above.

After preparing each of the three sequencing library types, the libraries were normalized and sequenced as 2 × 150 bp fragments on Illumina MiSeq or NextSeq platforms, following manufacturer standard protocols (Illumina).

**Bioinformatics analysis.** Sequence adapters were removed from raw fastq files using cutadapt v.1.12, and reads were quality trimmed and filtered using the FAST-X toolkit[47]. Trimmed and filtered paired-end (PE) reads were processed through a custom viral and bacterial database screening and de novo assembly pipeline to identify and assemble reads mapping to the Hendra virus as well as other viruses of interest. The pipeline consisted of removing host, ribosomal and repeat sequences, and mapping remaining PE reads to databases containing all known viruses and bacteria. De novo assembly was also performed on host/rRNA/repeat-depleted PE reads using the Spades assembly program[48]. Similarly, PE reads were also processed through a custom metavirus pipeline[49].

To generate full and partial consensus sequences from samples with ≥50% of bases above 15× coverage, the following workflow was implemented: (1) adapter-trimmed/quality-filtered fastq files were mapped to the best-matched Hendra virus reference sequence using Bowtie2 v.2.2.9 with '−local−very sensitive−no-unal -X 1500' parameter settings[50]. The sam alignments were converted to bam files using samtools[51]; (2) variant calling was performed using GATK v.4.2.5.0 HaplotypeCaller with '-ploidy 1'[52]; (3) indels and single nucleotide polymorphism (SNP) variants were filtered using BCFtools and QD > 10 (ref. 53); (4) SNPs were inserted into the original reference to generate an initial consensus sequence using BCFtools consensus[51]; (4) adapter-trimmed/quality-filtered fastq files were mapped back to the initial consensus sequence as described above; (5) BEDtools genomcov was used to generate bed files of regions with no or >4× coverage; and (6) BEDtools maskfasta was used to generate the final consensus sequence.

**Host species confirmation**
To confirm field host species assignment, sequenced reads were processed as follows: paired-end reads were trimmed of adapters using cutadapt[54], reads 1 and 2 of the pair were combined into one file and aligned to a Cytochrome Oxidase subunit 1 (COX1) database using ref. 55, only allowing for a maximum of 1 hit per read. The COX1 database for species identification is based on and available in fasta format from ref. 56. The blast output was summarized per species and divided by 2 to calculate the frequency. The frequencies per sample were visualized using Krona[57] (Supplementary Fig. 5) and the top hit (defined as the one that had the highest frequency) per sample was taken as the putative host species.

**Full genome recovery of HeV-g1 from horses**
Clinical specimens, blood, serum and nasal swabs were also collected from horses with symptoms and submitted to the Australian Centre for Disease Preparedness (ACDP) for the diagnosis of HeV. Samples were provided as diagnostic samples from national surveillance efforts by the governments of New South Wales and Queensland. Virus isolation was conducted at the BSL-4 laboratory at ACDP. The samples were passed on Vero cells (ATCC CCL-81) for three consecutive times of 7 days per passage. The samples were then tested by RT–qPCR to confirm the presence of replicating HeV genome. Then, full genome sequencing was performed as reported in ref. 58. Briefly, viral RNA was extracted using the MagMax 96 Viral RNA kit (ThermoFisher) in a MagMAX Express

Magnetic Particle Processor (ThermoFisher) following manufacturer instructions. TruSeq RNA library Prep Kit V2 (Illumina) was utilized for DNA library preparation according to manufacturer instructions. The purified libraries were quantified using a Qubit Fluorometer and an Agilent 2100 Bioanalyzer (Agilent). The concentration of the final libraries was normalized and pooled in equimolar ratios. The library pool was then loaded into MiSeq Reagent Kit V2 (2 × 150 cycles; Illumina) and sequenced in the MiSeq platform (Illumina) according to manufacturer instructions. The NGS sequence data were analysed using CLC Genomic Workbench 20 (Qiagen) with standard parameters. The raw reads were quality trimmed before assembly by both reference mapping (using HeV genomes available) and de novo assembly. The resultant sequences were confirmed as HeV-g1 by blasting against the NCBI non-redundant nucleotide database (https://blast.ncbi.nlm.nih.gov/Blast.cgi).

**Phylogenetic and evolutionary analysis**
**Phylogenetic analysis.** To examine the genome-wide evolution of HeV-g1 in Australia, an alignment was prepared using 73 complete genome sequences with MAFFT[59] including 57 from this study and 16 reference strains from NCBI GenBank. A second alignment was also prepared with non-coding regions removed. Finally, to improve sampling from incomplete genomes available in our own dataset and in the public dataset, we prepared a third alignment of G glycoprotein sequences (coding region) that contained additional 19 sequences. The best model for distance estimates was identified with the ModelFinder function[60] in IQ-TREE2 (ref. 61) as the one with the lowest Bayesian information criterion (BIC). Once the optimal model was selected, IQ-TREE2 automatically applied it to construct the phylogenetic tree. A maximum-likelihood (ML) phylogenetic tree was also constructed using IQ-TREE2, and branch support was assessed using both ultrafast bootstrap approximation (ufBoot, 1,000 replicates)[62] and SH-like approximate likelihood ratio test (SH-aLRT). The tree was visualized in FigTree (http://tree.bio.ed.ac.uk/software/figtree/) and midpoint rooted for purposes of clarity.

**Identification of clade-defining substitutions.** JalView[63] (https://doi.org/10.1093/bioinformatics/btp033) was used for genome translation and for the identification of persistent amino acid substitutions in the viral proteins. ESPript was used for multiple sequence alignment visualization[64]. The oldest genome of HeV sampled from a human in our library (ID: KY425627) was used as reference to annotate the substitutions. Experimentally determined or AlphaFold-predicted[65] structures were used to locate substitutions and for a preliminary inference of potential functional impacts. Visual molecular dynamics was used for visual inspection[66]. Sequence comparison with HeV-g2 and Nipah virus from Bangladesh (NiV$_{BD}$) and Malaysia (NiV$_{MY}$) strains was conducted using MZ229748, AY988601.1 and CAF25498.1 as reference, respectively.

**Phylodynamic analysis.** First, we assessed the temporal signal in HeV-g1 data by linear regression of root-to-tip distances on the ML phylogeny against time of sampling using the program TempEst (http://tree.bio.ed.ac.uk/software). Exact dates were available for all sequences generated in this study. For two samples for which the exact day of sample collection was not known, mid-month dates were assigned. Given the apparent temporal structure in the genome-scale alignment data, we then made estimates of the rates of evolutionary change (that is, nucleotide substitutions per site per year) and the TMRCA using the Bayesian MCMC method available in BEAST (v.1.10.4)[67]. Here we chose to use the complete coding regions dataset to make use of the SRD06 codon substitution model[68] and tested four different temporal and demographic scenarios: relaxed and strict molecular clock, each with constant population and SkyGrid coalescent model[69], with path sampling used to rank the final models.

For each analysis, 3 independent chains of 50 million generations (sampled every 10,000 states) were run to ensure convergence and then combined with appropriate burn-in. Statistical uncertainty was reflected in values of the 95% HPD. The maximum clade credibility (MCC) tree was estimated from the posterior distribution of trees, with node heights scaled to mean values and posterior probabilities showing the statistical support for individual nodes.

For the phylogeographic analysis, the 31 sampling sites from which HeV sequences were obtained were grouped into 7 discrete regions: Brisbane–southeast Queensland, Lismore–Byron–Ballina, mid-coast New South Wales, north and central Queensland, southeast Queensland, Toowoomba, and Cardiff Heights (Supplementary Table 2). Discrete trait phylogeographic reconstruction was performed using a symmetric substitution model, estimating transition events between regions across the posterior distribution of phylogenies. The BSSVS procedure[70] was applied to identify the subset of transition rates that best explained the observed spatial diffusion patterns, with statistical support evaluated using Bayes factors.

**Determination of the nature of selection pressures.** Next, we sought to determine the nature of selection pressures acting on HeV-g1 using the Datamonkey web server of the HyPhy package[71] (http://www.data-monkey.org/). Accordingly, codon-based ML methods were used to estimate the ratio of non-synonymous to synonymous substitutions per site ($dN/dS$ ratio; also denoted $\omega$): fixed-effects likelihood (FEL); fast, unconstrained Bayesian approximation (FUBAR); single-likelihood ancestor counting (SLAC); and random-effects likelihood (REL). Those sites with $P$ values of <0.05 or with posterior probability values of >0.95 were considered to provide significant evidence of positive selection (Supplementary Table 7).

### In vitro HeV phenotypic characterization
**Viruses.** Hendra virus/Australia/horse/1994 (GenBank MN062017) was propagated on Vero E6 cells. Hendra virus/*Pteropus* spp./Australia/RML-071/2018 (GenBank PV730206), Hendra virus/*Pteropus* spp./Australia/RML-084/2018 (GenBank PV730215) and Cedar virus (Cedar virus/*Pteropus* spp./Australia/AUSRML1/2021) were isolated on Vero E6 cells from HeV-positive urine samples as part of this study.

**Cells.** Human lung fibroblasts (HFL-1, ATCC CCL-153), black flying fox kidney cells (PaKiT, (1)), horse dermal fibroblasts (NBL-6, ATCC CCL-57), Vero E6 and Vero CCL-81 cells were used in this study. HFL-1 and PaKiT cells were maintained in 10% FBS DMEM/F-12 media (Sigma-Aldrich, D8437) with 1× penicillin/streptomycin (P/S) (Gibco, 10378016) and 1× non-essential amino acids (NEAA) (Gibco, 11140050). NBL-6 cells were maintained in 10% FBS EMEM (ATCC, 30-2003) with 1× P/S, 1× NEAA and 1.5% sodium bicarbonate (Gibco, 25080094). Vero E6 and CCL-81 cells were maintained in 10% FBS DMEM with 1× P/S.

**Replication kinetics in cells.** HFL-1, NBL-6 or PaKiT cells were plated at 250,000 cells per ml in 0.5 ml of media into 24-well plates and incubated overnight at 37 °C and 5% $CO_2$. The next day, the media were removed, and cells were infected with 0.1 ml of Hendra virus (prototype strain, strain 71 or strain 84) at a multiplicity of infection (MOI) of 0.01. After 1 h incubation at 37 °C and 5% $CO_2$ with rocking every 15 min, the inoculum was removed, and cells were washed three times with 1× DPBS (Gibco, 14190144). Fresh media (0.5 ml) were added to the wells immediately after the final wash. A sample of media, 0.25 ml at 1 h and 0.1 ml every 24 h thereafter, was collected to measure infectious virus. Fresh media were added to replace the volume removed. At 72 and 96 hpi, 3 replicate wells were lysed in 4% SDS buffer with 20% β-mercaptoethanol (β-ME) for protein or RLT buffer (QIAGEN) with 1:100 β-ME for RNA. Protein samples were boiled at 100 °C for 10 min. Both supernatant and RLT samples were stored at −80 °C until processing. Supernatants were serially diluted 1:10 in 2% FBS DMEM and incubated at 37 °C and 5% $CO_2$ on Vero CCL-81 cells for 5 days before evaluating cytopathic effect.

**IFN-I sensitivity.** To assess the sensitivity of HeV to IFN-I, Vero CCL-81 cells were treated with 5-fold serial dilutions of U-IFN (0–500 ng ml$^{-1}$) for 24 h before infection with MOI 0.01 for 24 h. Supernatants were collected at 24 hpi and titrated on Vero CCL-81 cells as described above.

**Western blot analysis.** Inactivated cell lysates collected in SDS buffer were run on 4–12% Bis-Tris NuPAGE gels (Invitrogen) at 150 V for 1 h, then transferred onto methanol-activated PVDF membranes (BioRad). Following a 1-h block in 5% powdered milk, the membranes were washed in buffer (1× Tris-HCL with 0.1% Tween 20) three times. Membranes were probed with primary antibody overnight with rocking at 4 °C. The next day, blots were washed three times, incubated with an HRP-conjugated secondary antibody for 1 h with rocking at room temperature, and washed three times before development. Blots were incubated with a 1:1 ratio of peroxidase and enhancer reagent (Clarity Western ECL (Bio-Rad) or SuperSignal West Femto (ThermoFisher)) and developed on an iBright imaging system (ThermoFisher). To calculate relative expression, the area under the curve was determined for each band using Fiji68. Primary antibodies used were: 1:1,000 pTBK1-172 (Cell Signaling Technology, 5483T), 1:1,000 pSTAT1 – Y701 (Cell Signaling Technology, 9167S), 1:1,000 pSTAT2 – Y690 (Cell Signaling Technology, 88410S), 1:1,000 total STAT1 (Cell Signaling Technology, 14994S), 1:1,000 total STAT2 (Cell Signaling Technology, 72604S), Nipah nucleoprotein (ThermoFisher, HL1436) or 1:5,000 actin (GeneTex, GTX629630). Secondary antibodies used were: 1:10,000 donkey anti-rabbit (GE Healthcare, NA934) and 1:10,000 sheep anti-mouse (GE Healthcare, NA931).

**RNA extractions and RT–qPCR.** RLT lysates from cell monolayers were transferred to 70% ethanol before extraction of RNA using the RNeasy extraction kit (QIAGEN). Cellular RNA was used to measure host genes using RT–qPCR (Supplementary Table 8). Following extraction, 17 μl of RNA was treated with TURBO DNase (Invitrogen) according to manufacturer instructions. After DNase treatment, the RNA was diluted 1:5 in molecular-grade water (Invitrogen). DNase-treated RNA (5 μl) was used for each host gene assessed. Fold change was calculated for host genes by dividing the sample relative gene expression by the average relative gene expression of the healthy controls or mock-treated cells ($2 - \Delta\Delta C_t$) and presented as $\log_2$ fold change.

**Virus neutralization assay.** Horse sera from an Equivac HeV vaccine vaccinated horse was provided by ACDP and used to determine the antigenicity of the three HeV isolates. Vero E6 cells were seeded in flat-bottom 96-well plates 1 day before infection. On the day of the assay, serum samples were heat inactivated at 56 °C for 30 min. Antibodies were diluted 1:20 in duplicate and then serially 2-fold diluted across a round-bottom 96-well plate using infection media. Then, 2,000 TCID$_{50}$ ml$^{-1}$ of HeV (prototype strain, Hendra 71 or Hendra 84) was added to each well of the antibody dilution plate and incubated at 37 °C for 1 h. Following incubation, 100 μl of the serum–virus mixture was transferred to the Vero E6 cell plates. Plates were incubated for 5 days at 37 °C, and CPE was visually assessed. The virus neutralization titre was expressed as the reciprocal value of the highest dilution of the serum that still inhibited virus replication.

### Spatiotemporal analyses
We tested for temporal structure in the presence of different clades using a permutation test to address two specific questions: (1) How unlikely is it that clade B is not observed before 2018 if it was in fact present throughout the sampling period? (2) Do the relative proportions of clades change over time?

Both questions were addressed using a permutation test in which the clade labels were shuffled randomly among all samples, taking into

account region and species. This random shuffling simulated a situation where all clades were present at the same relative proportions throughout the sampling period. By repeating this permutation many times, it was possible to generate a distribution of metric (see below) values that could arise in this situation, which was then used to compare with the observed value of the metric of interest. If the observed value fell outside a chosen interval of the permuted distribution, it indicated statistical support for the notion that it was unlikely to obtain the observed value if the relative proportions of clades were consistent throughout the sampling period.

To address the two questions, we selected two metrics: (1) a binary number (0 or 1) indicating whether clade B was present before 2018 and (2) the relative proportions of each clade during each 6-month period. During each permutation iteration, the clade labels were randomly shuffled between samples, with shuffling allowed only between samples from the same host species and from the same region. This ensured that the host and spatial structure of the dataset were maintained. The two metrics were calculated during each iteration and stored for later comparison with the observed values. We performed 1,000 iterations.

### Visualization
Phylogenetic trees were visualized in FigTree (http://tree.bio.ed.ac.uk/software/figtree/). Maps were created in R (v.4.3.0)[72] using the following packages: tidyverse[73], magrittr[74], maps[75], mapproj[76], ozmaps[77], sf[78,79], scatterpie[80], stringr[81], ggnewscale[82], ggforce[83] and ggspatial[84].

### Statistics and reproducibility
Significance tests were performed as indicated where appropriate for the data using GraphPad Prism 10.2.0 or R 4.3.1. Unless stated otherwise, statistical significance levels were determined as follows: no symbol = $P > 0.05$; * = $P \leq 0.05$, ** = $P \leq 0.01$, *** = $P \leq 0.001$, **** = $P \leq 0.0001$. The statistical test used is specified where appropriate. Sample sizes were predetermined on the basis of expected viral prevalence and statistical power calculations. We excluded one sequence with poor coverage. Individuals were captured in nets at the roost site. All captured individuals that produced a urine sample during holding or processing were included in screening. Multiple urine samples were pooled from each under-roost sheet, and a pooled sample from each sheet was screened. Samples selected for further analyses (including sequencing) were those with a $C_t$ value < 32. Blinding of investigators was not applicable. Because this is a field study in wild populations, results cannot be directly replicated. However, findings were replicated across sites and years. Data distribution was assumed to be normal, but this was not formally tested.

### Reporting summary
Further information on research design is available in the Nature Portfolio Reporting Summary linked to this article.

## Data availability
All sequences reported here have been submitted in GenBank (accession numbers PV730172–PV730228 (Supplementary Data 2). All other figure data are available as source data and on Figshare at https://doi.org/10.6084/m9.figshare.30670544 (ref. 85). Source data are provided with this paper.

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

## Acknowledgements

The project was supported by the National Science Foundation (DEB1716698, EF-2231624), the DARPA PREEMPT program Cooperative Agreement #D18AC00031 and the Intramural Research Program of National Institute of Allergy and Infectious Diseases of the National Institutes of Health. The contributions of the NIH author(s) are considered Works of the United States Government. The findings and conclusions presented in this paper are those of the author(s) and do not necessarily reflect the views of the NIH or the US Department of Health and Human Services. T.J.L. was supported by an Endeavour Postgraduate Leadership Award and a Research Training Program scholarship sponsored by the Australian Government. A.J.P. was supported by a Sydney Horizon Fellowship, an ARC DECRA fellowship (DE190100710) and a Queensland Government Accelerate Postdoctoral Research Fellowship. We acknowledge the Bundjalung, Butchulla, Danggan Balun, Gomeroi, Gumbainggir, Kabi Kabi, Taribelang Bunda, Turrbal, Widjabul Wia-bal, Yugambeh and Yuggera Ugarapul people, who are the Traditional Custodians of the land upon which this work was conducted. We also thank government and private landholders for granting permission for fieldwork, and broader team members and volunteers for their contributions, including P. Eby, M. Kessler, A. Dale, M. Ruiz Aravena, L. Chirio, M. Allonby, R. Smethurst, R. Brooks, T. Pearson, L. McGuire, K. Silas, T. Padgett-Stewart, D. Karkkainen, J. Scaccia, A. Ananda, E. Glennon, H. Eiseman, C. Pietromonaco, S. LaTrielle, I. Knights, D. Riseley, E. Spence, S. M. Januario da Silva and many others. Hendra virus RT–qPCR primers and probe sequences were provided by the Public Health Agency of Canada. We thank D. Sturdevant for designing the primers for amplicon-based sequencing; J. R. Port, J. Purushottam, J. Schulz and Z. Weishampel for helping to screen bat samples; staff of the Diagnostic Virology Team, Australian Centre for Disease Preparedness, for technical supports, including virus isolation at a BSL-4 laboratory; the Queensland and New South Wales Chief Veterinary Officers for allowing us to include the HeV horse sequences in this paper; and the relevant veterinary and laboratory staff in each of these states for their contributions to the disease outbreak investigations.

This research used resources of the Oak Ridge Leadership Computing Facility at the Oak Ridge National Laboratory, which is supported by the Office of Science of the US Department of Energy under Contract No. DE-AC05-00OR22725.

## Author contributions

C.K.Y., A.J.P., J.W., K.M.-W., R.K.P. and V.J.M. conceived the research. C.K.Y., A.J.P., J-S.E., J.W., K.M.-W., R.K.P. and V.J.M. designed the research methodology. A.J.P., R.K.P. and V.J.M. acquired funding for the research. D.N.J.-S., C.A.F., T.J.L., J.W., K.M.-W., R.K.P. and A.J.P. collected samples. C.K.Y., A.J.P., J.W., K.M.-W., R.K.P., K.H. and V.J.M. led project administration. C.K.Y., J-S.E., E.T.P., S.v.T., B. Borremans, B. Bulloch, J.W., K.M.-W., S.L.A., T.B., K.B., C.M. and A.J.P. curated the data. C.K.Y., J-S.E., B. Borremans, B. Bulloch, E.T.P., S.v.T., A.V., M.S., M.P., W.C., D.N.J.-S., A.J.P. and V.J.M. analysed and visualized the data, E.T.P., A.V., M.S., M.P. and D.J. described clade substitutions. A.J.P., C.M., R.K.P. and V.J.M. provided supervision. C.K.Y., S.L.A., T.B. and K.B. conducted laboratory screening, testing and validation. R.K.P., A.J.P. and V.J.M. provided resources. C.K.Y., J-S.E., A.J.P. and V.J.M. drafted the manuscript. All authors reviewed and edited the manuscript.

## Competing interests

The authors declare no competing interests.

## Additional information

**Extended data** is available for this paper at https://doi.org/10.1038/s41564-025-02254-7.

**Correspondence and requests for materials** should be addressed to Vincent J. Munster.

[1]Laboratory of Virology, Division of Intramural Research, National Institutes of Health, Hamilton, MT, USA. [2]Centre for Virus Research, Westmead Institute for Medical Research, Westmead, New South Wales, Australia. [3]Sydney Institute for Infectious Diseases, The University of Sydney, Sydney, New South Wales, Australia. [4]Oak Ridge National Laboratory, Oak Ridge, TN, USA. [5]Genomics Research Section, Research Technologies Branch, Division of Intramural Research, National Institute of Allergy and Infectious Diseases, National Institutes of Health, Hamilton, MT, USA. [6]Australian Centre for Disease Preparedness, Commonwealth Scientific and Industrial Research Organisation (CSIRO), Canberra, Australia. [7]Wildlife Health Ecology Research Organization, San Diego, CA, USA. [8]Odum School of Ecology, University of Georgia, Athens, GA, USA. [9]Center for the Ecology of Infectious Diseases, University of Georgia, Athens, GA, USA. [10]Department of Public and Ecosystem Health, College of Veterinary Medicine, Cornell University, Ithaca, NY, USA. [11]Department of Ecology, Montana State University, Bozeman, MT, USA. [12]Biochemistry and Cellular and Molecular Biology, The University of Tennessee, Knoxville, TN, USA. [13]Department of Biology, Medgar Evers College, City University of New York, New York, NY, USA. [14]Sydney School of Veterinary Science, University of Sydney, Sydney, New South Wales, Australia. [15]Centre for Planetary Health and Food Security, Griffith University, Nathan, Queensland, Australia. [16]These authors contributed equally: Claude Kwe Yinda, John-Sebastian Eden, Erica T. Prates. [17]These authors jointly supervised this work: Alison J. Peel, Vincent J. Munster. ✉e-mail: vincent.munster@nih.gov

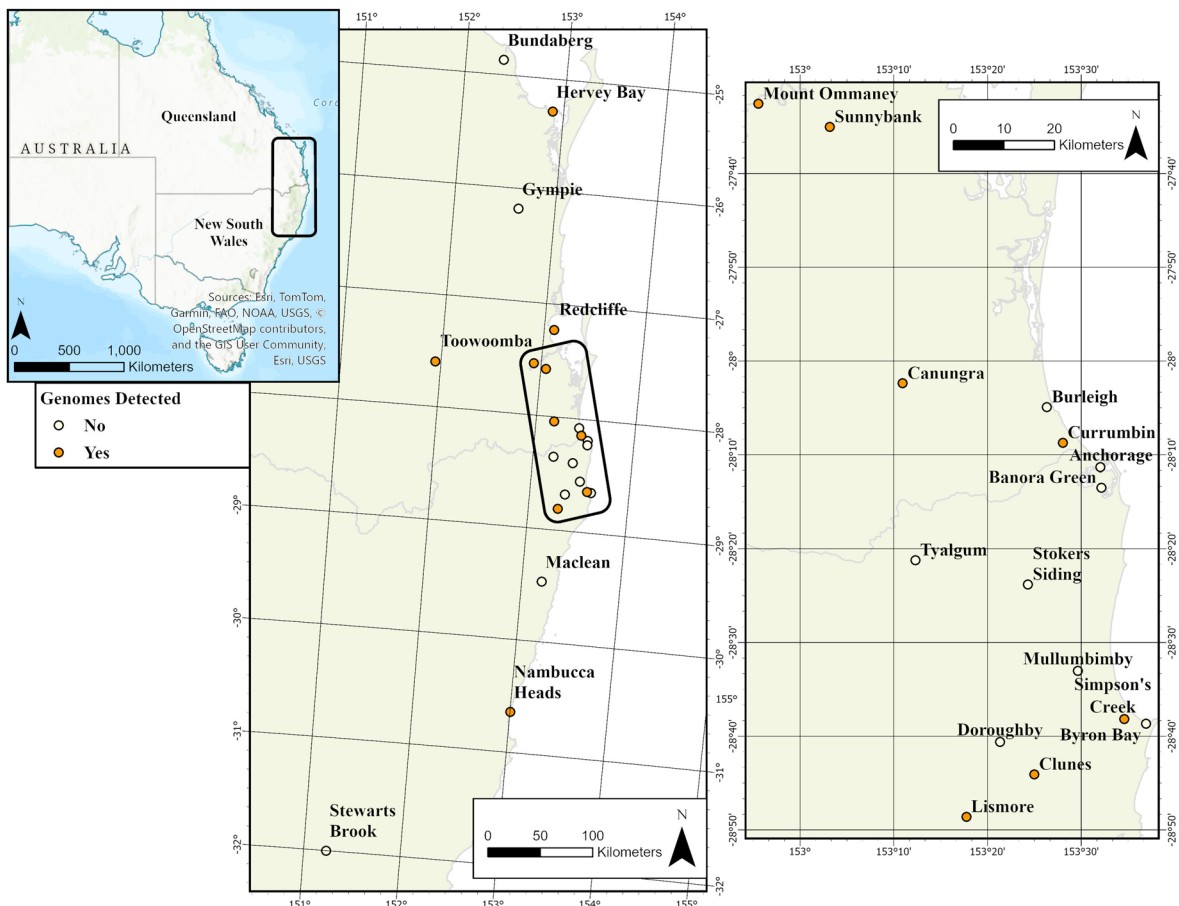

**Extended Data Fig. 1 | Maps of the study area and bat sampling locations.** Left: Eastern Australia, with the study area for bat sample collection highlighted by the black rectangle. Centre: Study area, showing bat sample collection sites (circles) and genome recovery success (brown fill representing sites where genomes were recovered). Right: Enlarged view of a subset of bat sampling locations from the area highlighted in the center map.

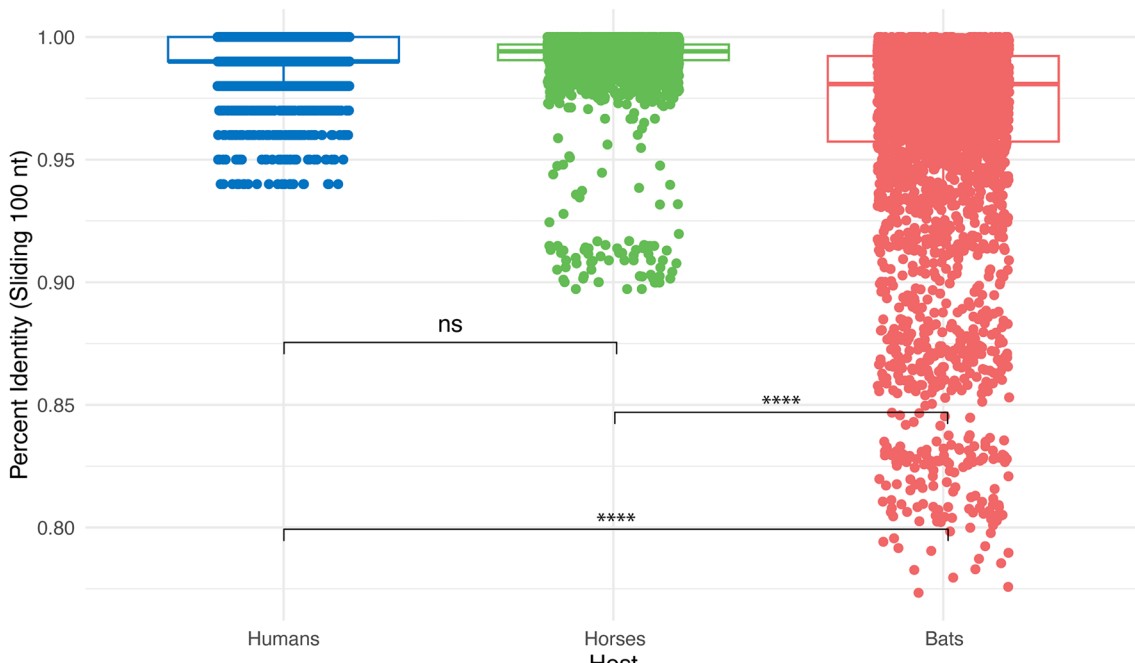

**Extended Data Fig. 2 | Genetic variation and base composition in HeV-g1 from different hosts.** (**a**) sequences from all hosts, (**b**) sequences from human, (**c**) sequences from horse, and (**d**) sequences from bat. The genome organization of HeV-g1 and the relative position of each gene are shown at the top of right and left panels. The genetic variation in the HeV-g1 genome was determined by plotting the average nucleotide identity of 100 bases using a sliding window in steps of 5 bases (left y axis, colored brown, horizontal dashed line represents average nucleotide identity) across the genome (x axis). Genome base composition also was plotted using a sliding window that calculated the average GC content over 100 bases in steps of 5 bases (right y axis, colored blue, horizontal dashed line represents average GC content). (**e**) Distribution of HeV-g1 genome percent identity across host species. Boxplots show the distribution of genome-wide percent identity values calculated in sliding 100-nt windows for HeV sequences derived from humans ($n = 2$), horses ($n = 19$) and bats ($n = 52$). Each box represents the interquartile range (IQR) with the horizontal line indicating the median. Whiskers extend to $1.5 \times$ IQR, and individual data points (jittered for visibility) represent individual sliding-window values. Colors indicate host species. Percent identity significance among hosts was determine using a non-parametric two-sided Kruskal–Wallis test followed by pairwise two-sided Wilcoxon rank-sum test to identify which pairs differed significantly. P values for comparisons of HeV genomes between hosts are indicated: **** <0.0001 and ns = not significant. Bat-derived genomes displayed significantly lower percent identity compared with horse and human genomes ($P < 0.0001$), whereas identities did not differ significantly between horse and human isolates ($P = 0.882$).

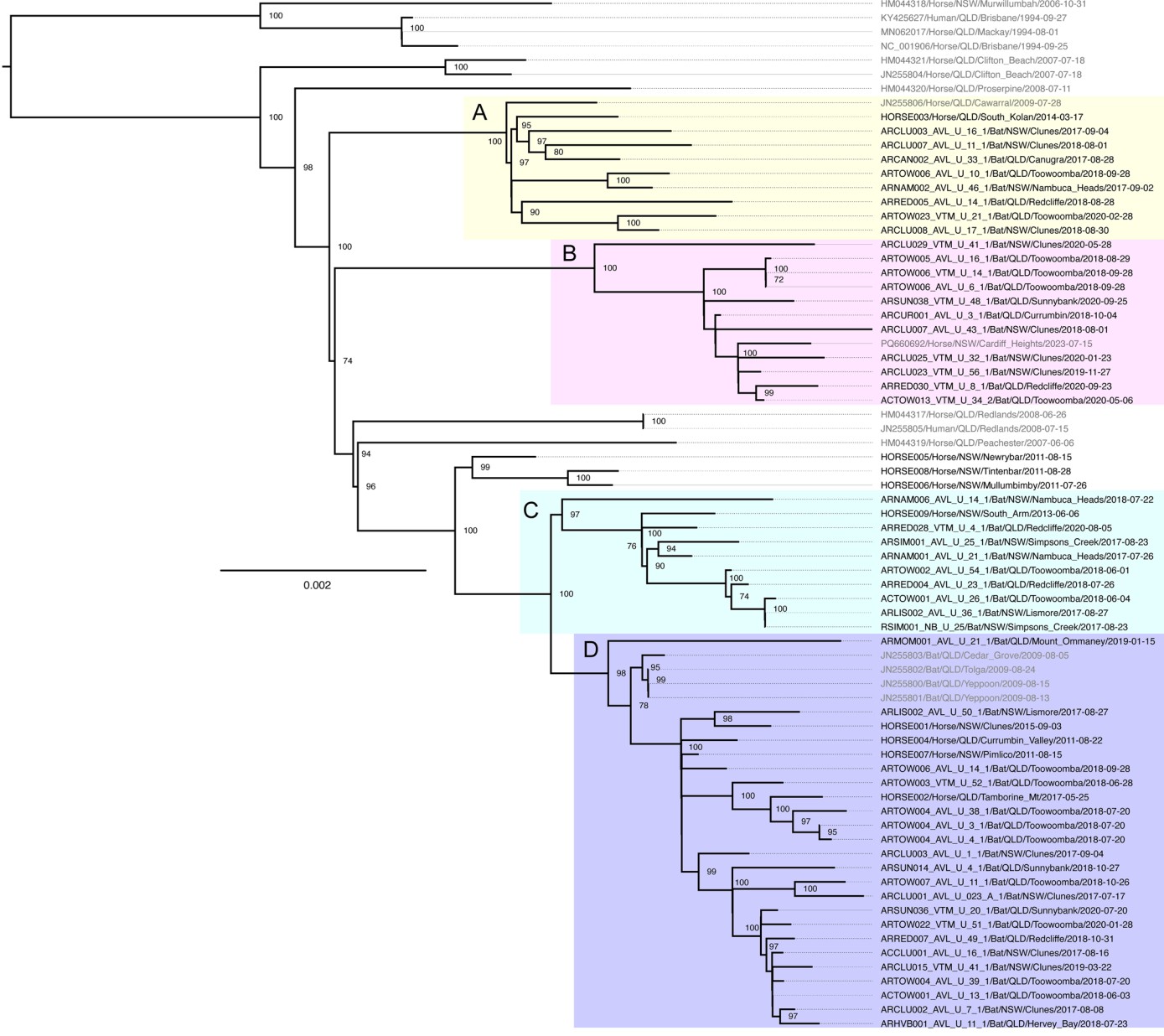

**Extended Data Fig. 3 | Whole genome phylogeny of HeV-g1.** Of the 73 genomes used in the phylogeny, 57 were newly generated in this study. The alignments were built with MAFFT (FFT-NS-1 algorithm[59], with the best model for distance estimates (TIM + F + I) identified with the ModelFinder function[60] as the one with the lowest Bayesian information criterion (BIC). Maximum likelihood phylogenetic tree was constructed using IG-TREE2[61] and branch support was assessed using both ultrafast bootstrap approximation (ufBoot, 1000 replicates)[62] and SH-like approximate likelihood ratio test (SH-aLRT). The tree was visualized in FigTree (http://tree.bio.ed.ac.uk/software/figtree/), and midpoint rooted for purposes of clarity. Only bootstrap support values greater of 69 are shown. Bars indicate nucleotide substitutions per site. Greyed tips represent sequences obtained from GenBank.

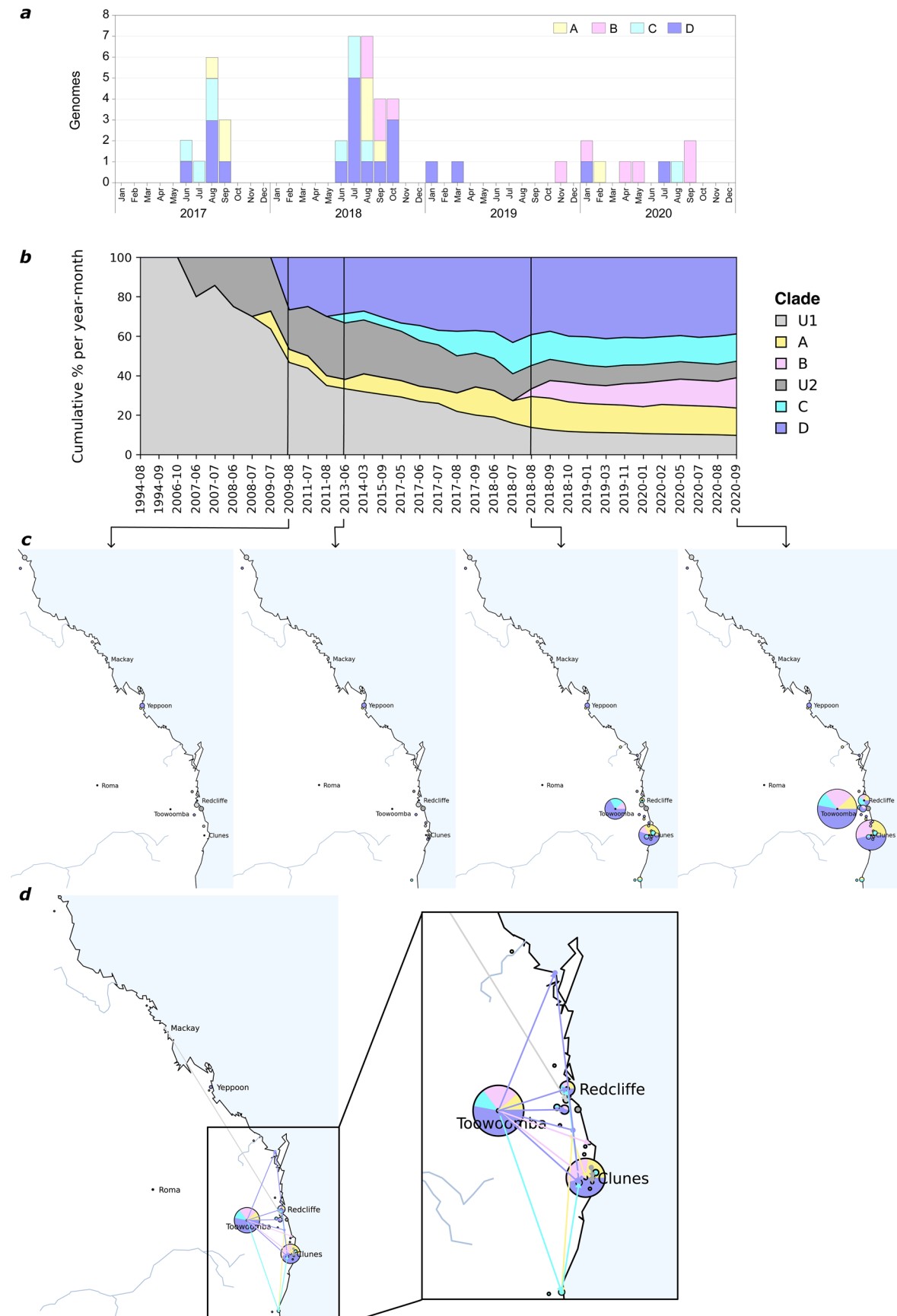

**Extended Data Fig. 4 | HeV-g1 clades circulating in Australia between 1994 and 2020. (a)** Clade distribution by month and year. **(b)** The cumulative total distribution of the identified clades at specific points in time. **(c)** A series of time-ordered snapshots indicating the spatial distribution of identified clades at different points in time. **(d)** The cumulative clade distribution in September 2020; arrows indicate potential migration paths. An arrow is drawn if the same clade was detected at a different location in the month prior. We did not include 2023 horse sequence in this analysis.

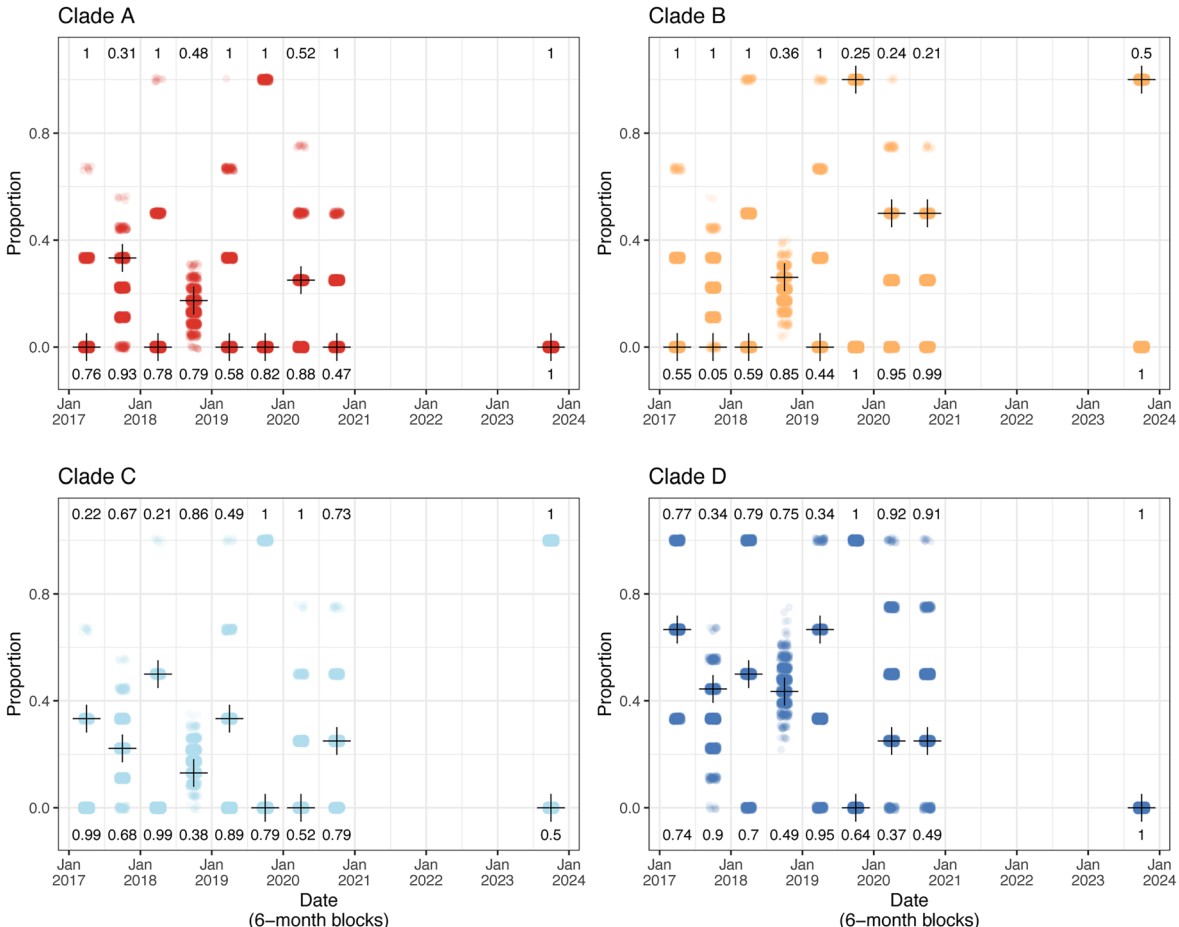

**Extended Data Fig. 5 | Temporal structure using permutations methods.** Permuted and observed (black plus sign) relative proportions of clade samples for each 6-month period. Numbers at the top and bottom are the proportions of permuted values ≥ and ≤ observed value, respectively.

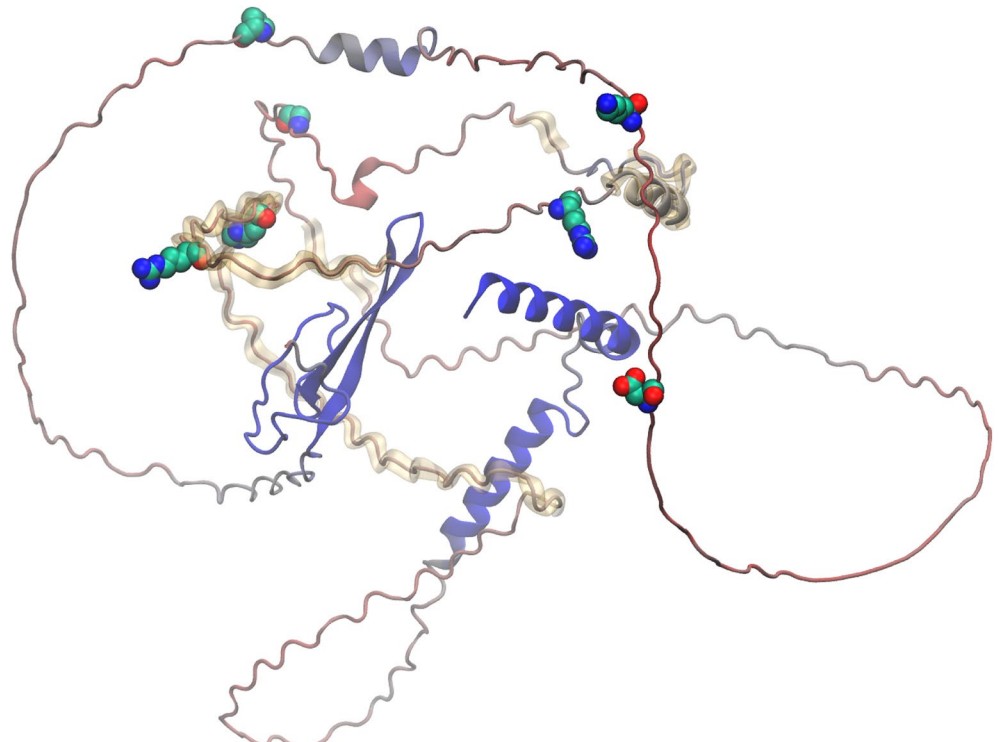

**Extended Data Fig. 6 | Clade-defining mutations are depicted in an AlphaFold model of protein V as van der Waals spheres.** The clade defining mutations, including P352S in clade B, are highlighted with a bold label. The protein backbone is colored to reflect the predicted local model quality score, pLDDT[95], ranging from 0-100 (red-blue).

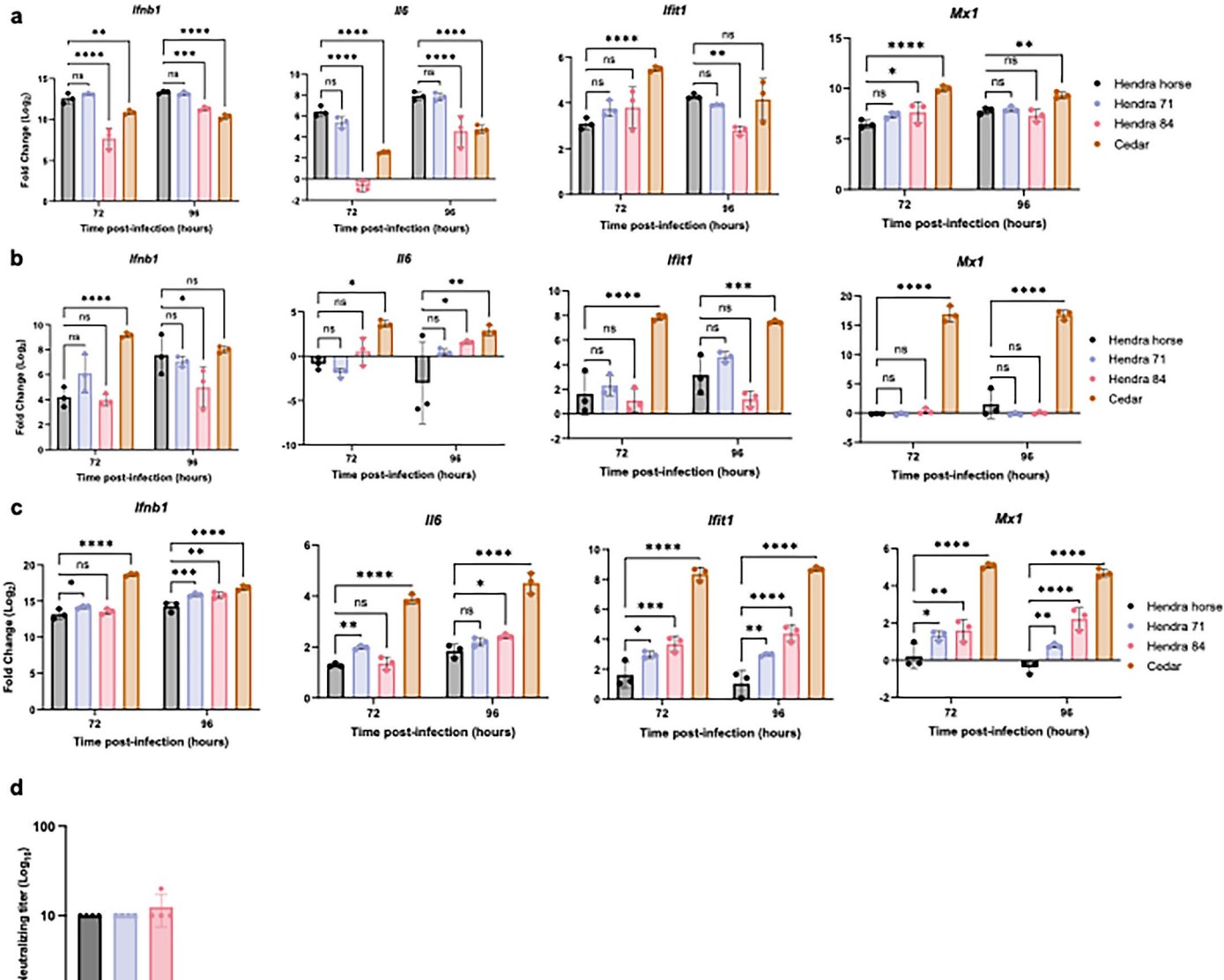

**Extended Data Fig. 7 | Innate immune response to and neutralization of Hendra virus.** Human (HFL-1) (**a**), horse (NBL-6) (**b**), and black flying fox (PaKiT) (**c**) cells were infected with a multiplicity of infection (MOI) of 0.01 of Hendra virus prototype (Hendra virus/Australia/horse/1994), -clade B (Hendra 84), or -clade D (Hendra 71), or Cedar virus. (a-c) RT-qPCR of host genes. ΔCT values were normalized to the average ΔCT value of mock infected cells to calculate ΔΔCT. Data presented as log2 fold change relative to uninfected mock cells. Fold change of each gene at each time point was compared to Hendra-infected cells using a two-way ANOVA with Dunnett's multiple test comparison. Data are presented as mean values (*n* = 3) with SD shown. Representative of two independent experiments. This data was used for the calculation of the host gene expression relative to virus titer shown in Fig. 5b-d. P values for comparisons to mock cells are indicated. **** <0.0001, *** <0.001, ** <0.01, * <0.05. (**d**) Neutralization of three virus strains by vaccinated horse serum. Serial dilutions of serum from a horse vaccinated with Equivac® HeV were tested in a CPE-based neutralization assay against three virus strains (Hendra virus prototype (horse), -clade B (84), or -clade D (71)). Neutralization titer was expressed as the reciprocal value of the highest dilution of the serum which still inhibited virus replication. Data are presented as mean values (*n* = 3) with SD shown.

**Extended Data Table 1 | Clades and their sub-clade-defining mutations**

| Protein | Substitution | A | B | C | D | U |
|---|---|---|---|---|---|---|
| C | T94I | 10/10 | 8/11 | 10/10 | 26/28 | 8/13 |
| F | S88L | | | 3/10 | | |
| F | A141T | | | | 3/28 | |
| G | N12D | | 3/11 | | | |
| G | S175T | | | | 24/28 | |
| G | T311I | | | | 3/28 | |
| G | A336T | | | 10/10 | 28/28 | 3/13 |
| G | T373I | | 3/11 | | | |
| L | Q141R | | 10/11 | | | |
| L | V215I | | 10/11 | | | |
| L | S260P | | | | 28/28 | |
| L | Q321H | | | 10/10 | 26/28 | |
| L | G325E | 10/10 | 11/11 | 10/10 | 27/28 | 10/13 |
| L | M645T | 10/10 | | | | |
| L | K658R | | | | 28/28 | |
| L | M693T | 10/10 | 11/11 | 10/10 | 28/28 | 9/13 |
| L | A1655T | | 11/11 | | | |
| L | H2081R | 10/10 | | | | |
| L | I2110T | 10/10 | | | | |
| L | P2178S | | | | 4/28 | |
| L | V2244I | | | 9/10 | 26/28 | 3/13 |
| M | N31D | 10/10 | 11/11 | 10/10 | 28/28 | 12/13 |
| N | D6E | 9/10 | 8/11 | 9/10 | 27/28 | 10/13 |
| N | E498G | 10/10 | 11/11 | 10/10 | 28/28 | 9/13 |
| N | R512G | 10/10 | 11/11 | 10/10 | 27/28 | 7/13 |
| N | P518S | 10/10 | 11/11 | 10/10 | 27/28 | 9/13 |
| P/V/W | H139Y | | | | 3/28 | |
| P/V/W | R149C | 10/10 | | | | |
| P/V/W | R170G | 10/10 | 11/11 | 10/10 | 21/28 | 10/13 |
| P/V/W | G219E | | | 10/10 | 27/28 | |
| P/V/W | E251K | | | | 5/28 | |
| P/V/W | V278A | | | 5/10 | | |
| P/V/W | D304G | 10/10 | | | | |
| P/V/W | K317E | | | 10/10 | 28/28 | 3/13 |
| P/V/W | P352S | | 10/11 | | | |
| P | L612V | | | 9/10 | | |

Letters from A to U represent individual clades. The denominators indicate the total number of Hendra Virus (HeV) genotypes within each clade, and the numerators specify the number of genotypes that exhibit the mutation listed in each row. The coloured shading corresponds to HeV clades as identified in the phylogeny in Fig. 1.

# Reporting Summary

## Statistics

For all statistical analyses, confirm that the following items are present in the figure legend, table legend, main text, or Methods section.

| n/a | Confirmed | |
|---|---|---|
| ☐ | ☒ | The exact sample size (*n*) for each experimental group/condition, given as a discrete number and unit of measurement |
| ☒ | ☐ | A statement on whether measurements were taken from distinct samples or whether the same sample was measured repeatedly |
| ☐ | ☒ | The statistical test(s) used AND whether they are one- or two-sided *Only common tests should be described solely by name; describe more complex techniques in the Methods section.* |
| ☒ | ☐ | A description of all covariates tested |
| ☐ | ☒ | A description of any assumptions or corrections, such as tests of normality and adjustment for multiple comparisons |
| ☒ | ☐ | A full description of the statistical parameters including central tendency (e.g. means) or other basic estimates (e.g. regression coefficient) AND variation (e.g. standard deviation) or associated estimates of uncertainty (e.g. confidence intervals) |
| ☐ | ☒ | For null hypothesis testing, the test statistic (e.g. *F*, *t*, *r*) with confidence intervals, effect sizes, degrees of freedom and *P* value noted *Give P values as exact values whenever suitable.* |
| ☐ | ☒ | For Bayesian analysis, information on the choice of priors and Markov chain Monte Carlo settings |
| ☒ | ☐ | For hierarchical and complex designs, identification of the appropriate level for tests and full reporting of outcomes |
| ☐ | ☒ | Estimates of effect sizes (e.g. Cohen's *d*, Pearson's *r*), indicating how they were calculated |

*Our web collection on statistics for biologists contains articles on many of the points above.*

## Software and code

Policy information about availability of computer code

| Data collection | NA |
|---|---|
| Data analysis | R 4.3.0 using the following packages; tidyverse, magrittr, maps, mapproj, ozmaps, sf, scatterpie, stringr ggnewscale, ggforce, and ggspatial<br>GraphPad Prism 10<br>IG-TREE2<br>MAFFT v7.505<br>JalView v10.0.5<br>TempEst v1.5.3<br>BEAST v1<br>FigTree v1.4.4<br>HyPhy package of datamonkey |

For manuscripts utilizing custom algorithms or software that are central to the research but not yet described in published literature, software must be made available to editors and reviewers. We strongly encourage code deposition in a community repository (e.g. GitHub). See the Nature Portfolio guidelines for submitting code & software for further information.

# Data

Policy information about availability of data

All manuscripts must include a data availability statement. This statement should provide the following information, where applicable:
- Accession codes, unique identifiers, or web links for publicly available datasets
- A description of any restrictions on data availability
- For clinical datasets or third party data, please ensure that the statement adheres to our policy

> All novel sequences reported here have been submitted in GenBank (accession numbers are in Extended Data Table 2).

# Research involving human participants, their data, or biological material

Policy information about studies with human participants or human data. See also policy information about sex, gender (identity/presentation), and sexual orientation and race, ethnicity and racism.

| | |
|---|---|
| Reporting on sex and gender | Research does not involve human participants |
| Reporting on race, ethnicity, or other socially relevant groupings | Research does not involve human participants |
| Population characteristics | Research does not involve human participants |
| Recruitment | Research does not involve human participants |
| Ethics oversight | Research does not involve human participants |

Note that full information on the approval of the study protocol must also be provided in the manuscript.

# Field-specific reporting

Please select the one below that is the best fit for your research. If you are not sure, read the appropriate sections before making your selection.

☐ Life sciences     ☐ Behavioural & social sciences     ☒ Ecological, evolutionary & environmental sciences

For a reference copy of the document with all sections, see nature.com/documents/nr-reporting-summary-flat.pdf

# Ecological, evolutionary & environmental sciences study design

All studies must disclose on these points even when the disclosure is negative.

| | |
|---|---|
| Study description | This study aimed to enhance the understanding of Hendra virus (HeV) evolution and transmission by conducting extensive spatiotemporal sampling and whole-genome sequencing of HeV-positive samples from bats and horses. |
| Research sample | Sampling was conducted at flying fox roosts in southeast Queensland and mid- to north-coast New South Wales. Horse Hendra virus sequences included in this study came from horses infected with Hendra virus and then diagnosed and and samples from positive horses were sequenced at the Australian Centre for Disease Preparedness (ACDP). |
| Sampling strategy | Sample sizes were predetermined based on expected viral prevalence and statistical power calculations. |
| Data collection | December 2016 - September 2020. Data were collected via field sampling of bat populations, molecular viral screening assays and sequencing. Field samples were handled by trained field teams, with data collection occurring either on paper data sheets, transferred to digital formats, or directly onto a tablet data collection form. |
| Timing and spatial scale | Field data were collected monthly over four consecutive years at multiple roosting sites across southeastern Australia. Laboratory data were gathered continuously over the course of the study. |
| Data exclusions | We excluded one sequence with poor coverage. |
| Reproducibility | Because this is a field study in wild populations, results cannot be directly replicated. However, findings were replicated across sites and years. |
| Randomization | Individuals were captured in nets at the roost site. All captured individuals that produced a urine sample during holding or processing were included in screening. Multiple urine samples were pooled from each under-roost sheet, and a pooled sample from each sheet was screened. Samples selected for further analyses (including sequencing) were those with a Ct value <32. |

| Blinding | The laboratory team was blind to the metadata associated with each sample. |

Did the study involve field work?  ☒ Yes  ☐ No

## Field work, collection and transport

| Field conditions | Fieldwork was conducted across all seasons and under varying climatic conditions, however rain-affected sampling sessions were abandoned and rescheduled. |
| Location | Sampling occurred in southeastern Australia, in roost sites located between 24.87°S and 32.0°S latitude . Full site details are provided in Supplementary Table 1 |
| Access & import/export | Samples were exported from Australia to the USA. All necessary permits for sample collection were obtained from relevant wildlife and government authorities. |
| Disturbance | Disturbance to bat populations was minimized by conducting individual sampling outside of the birthing and early lactation season and limiting the time of interaction with captured animals. |

# Reporting for specific materials, systems and methods

We require information from authors about some types of materials, experimental systems and methods used in many studies. Here, indicate whether each material, system or method listed is relevant to your study. If you are not sure if a list item applies to your research, read the appropriate section before selecting a response.

### Materials & experimental systems

| n/a | Involved in the study |
|---|---|
| ☐ | ☒ Antibodies |
| ☐ | ☒ Eukaryotic cell lines |
| ☒ | ☐ Palaeontology and archaeology |
| ☐ | ☒ Animals and other organisms |
| ☒ | ☐ Clinical data |
| ☒ | ☐ Dual use research of concern |
| ☒ | ☐ Plants |

### Methods

| n/a | Involved in the study |
|---|---|
| ☒ | ☐ ChIP-seq |
| ☒ | ☐ Flow cytometry |
| ☒ | ☐ MRI-based neuroimaging |

## Antibodies

| Antibodies used | pSTAT1 – Y701 (Cell Signaling Technology, 9167S)<br>pSTAT2 – Y690 (Cell Signaling Technology, 88410S)<br>total STAT1 (Cell Signaling Technology, 14994S)<br>total STAT2 (Cell Signaling Technology, 72604S)<br>pTBK1-172 (Cell Signaling Technology, 5483T)<br>Actin (GeneTex, GTX629630)<br>Donkey-anti-rabbit (GE Healthcare, NA934)<br>Sheep-anti-mouse (GE Healthcare, NA931) |
| Validation | All of the above antibodies were validated by the manufacturer via western blot, flow cytometry, and/or immunohistochemistry |

## Eukaryotic cell lines

Policy information about cell lines and Sex and Gender in Research

| Cell line source(s) | HFL-1, NBL-6 and Vero E6 cells were sourced from ATCC.<br>PaKiT cells were sourced from Michelle Baker at ACDP/AAHL in Australia. |
| Authentication | All cell lines were authenticated using Cyt sequencing. |
| Mycoplasma contamination | All cell lines were tested regularly for Mycoplasma contamination. |
| Commonly misidentified lines (See ICLAC register) | No commonly misidentified cell lines were used. |

# Animals and other research organisms

Policy information about studies involving animals; ARRIVE guidelines recommended for reporting animal research, and Sex and Gender in Research

| Laboratory animals | NA |
|---|---|
| Wild animals | Urine samples were collected from plastic sheets placed underneath flying fox roosts. Urine samples were also collected directly from individual bats captured in mist nests at their roost site. Bats were held in cotton bags with the bottom third lined with plastic and a urine collection bag attached to facilitate the collection of samples. Bats were anaesthetised for further sample collection and their species, sex, and age class (adult, subadult, juvenile) were recorded. Urine samples were collected directly from the bat if it urinated while under anaesthetic, or from the urine collection bag. Bats were released after sample collection. |
| Reporting on sex | Bats were anaesthetised for further sample collection and their species, sex, and age class (adult, subadult, juvenile) were recorded. However, the sequences analysis in this study does not include sex or age. |
| Field-collected samples | Urine samples from roosts or individual bats were kept frozen until extracted and screened for Hendra virus. |
| Ethics oversight | Griffith University Animal Ethics Committee Approval ENV/10/16/AEC and ENV/07/20/AEC. Personal Protective Equipment and disinfection protocols followed best practice guidelines (e.g. IUCN Bat Specialist Group, 2021; Wildlife Health Australia, 2020). |

Note that full information on the approval of the study protocol must also be provided in the manuscript.

# Plants

| Seed stocks | No plant seed stocks were used. |
|---|---|
| Novel plant genotypes | No plants were used |
| Authentication | No plants were used |

