## [Peer Review File · Nature Microbiology]

Spatio-temporal dynamics of Hendra virus in Australia reveals stable maintenance of diverse viral clades among Pteropus bats

Corresponding Author: Dr Vincent Munster

Version 0:

Reviewer comments:

Reviewer #1

(Remarks to the Author)

This manuscript entitled 'Spatio-temporal dynamics of Hendra virus in Pteropus bats in Australia reveals high evolutionary diversity linked with spillover' by C.K. Yinda et al. closes a remarkable knowledge gap regarding the spatiotemporal evolution of Hendra virus lineages in bats, horses and humans in the field. The analysis was done thoroughly by analyzing almost 7000 bat and the available horse samples, and the results are presented and discussed clearly. It is a pity that a lot of also highly valuable data had to be added as Supporting Information. My main concerns regarding the presentation is that due to lengths constraints, some information that would be very useful to understand the context are presented too briefly, which can easily be corrected. Secondly, a number of possible explanations for spillover, mutation and transmission events are excluded throughout the manuscript without offering clear hypothesis as to how the real mechanisms may be composed.

My comments in particular:

Abstract:

52-54: I would prefer to read a thesis how shedding events do occur, instead of reading how they, based on these data, do not occur.

54-56: This conclusion is not very well based on specific data from this study. When presenting a conclusion like this, I would expect possible mechanisms to be presented in the discussion, which are lacking or not stressed sufficiently clearly.

Significance statement:

62-64: again, it is stated how spillover events are not driven, but a data-based hypothesis on how they occur would be much more helpful here.

64-67: the perspective on improving our options for outbreak management and prevention read very nicely, but how exactly do these data translate into management options. I would prefer reading clearer statements here.

Introduction:

74-81: I would suggest to also briefly mention Nipah virus, just to give a broader context.

82-84: mention that the g2 variants were first detected after the sampling for this study was finished, otherwise it does not become clear why HeV-g2 was excluded from this study.

89: delete 'hundreds of' (repetition)

Results:

130 ff: add a short explanation as to how the clades A, B, C, D were ordered, it is not chronological.

212: LD needs to be explained here, explanation follows in line 267

283: 'our results suggest'... (delete s)

293: again, we learn what is not happening 'implies that immune pressure is insufficient to drive the evolution of these strains'. I cannot see the data supporting this in this manuscript, and as stated above, I would much rather see hypotheses highlighting the possible mechanisms underneath the observations.

Methods:

312-313: it is stated 'but other species were also present', how many samples were exactly analyzed from other bat species?

353-361 and 378-388: why were two different sequencing approaches used for bat and horse samples?

404: as reference, 'the oldest genome of HeV sampled from a human in our library' was used, why not the oldest genome in general?

415-416: this is a repetition to what has been described in lines 189-193.

Figure 1: I was wondering why only three human sequences were included in this analysis. With four human fatal cases, are there only three sequences available?

Reviewer #2

(Remarks to the Author)

Major Concerns:

1. Sampling Strategy Limitations

Given the profound influence of natural environments on viral evolution, please clarify: 1) Why sampling was restricted to the eastern coastline. 2) Whether non-urine samples (e.g., blood) were considered for collection, especially given the low viral loads (Ct values >32) in most HeV-positive samples.

2. HeV-g2 Subtype Analysis Omission

Since the study team previously reported HeV-g2 spillover events (Refs 16 and 17), please explicitly address: 1) Whether HeV-g2 was detected in the current samples. If not, is this due to geographic or host restrictions? 2) If detected, please clarify its phylogenetic classification (e.g., as an independent clade or subclade).

3. Phylogenetic Divergence of Spillover Sequences

Most spillover sequences from horses/humans (13/21) fall outside the four main clades (Fig. 1a). It is recommended to calculate host-specific selection pressure and analyze differences in key mutations between bat-derived and spillover strains.

4. Insufficient Mutation Annotation and Phenotypic Correlation

1) What are the mutation profiles of the other three structural proteins besides P/L/G?

2) The two LD mutations identified in the G protein, S175T and A336T, although not located at the G-EB2/3 interaction interface, may affect viral function by influencing the expression efficiency and conformation of G tetramers or the interaction with the F protein.

3) More LD mutations were observed in the L and P proteins. Given the recently solved structure of the NiV L-P complex (Cell. 2025 Feb 6;188(3):688-703.e18.), the authors should provide a more detailed discussion on the potential impacts of these mutations.

4) In-depth analysis of the effects of mutations on viral virulence, cell tropism, and immune evasion is critical for assessing the public health significance of these variants.

5. Fig. 1c is not cited in the Main Text

Minor Issues:

1. Replace hyphens with en-dashes for ranges (e.g., "Suppl. Fig. 9–11").

2. Line 44: Replace the comma after "genetic diversity" with a semicolon in the keywords section.

3. Line 156: Add "is" after "Clade B".

4. Line 194: Correct "clade C and D" to "clades..."

5. Line 221: Add "in" before "Australia".

6. In Fig.3, there is a free node between the 2000–2010 grid lines.

Reviewer #3

(Remarks to the Author)

In this paper, the authors generate critical genomic resources for the bat-borne zoonosis, Hendra virus, increasing the total number of available full genome sequences for HeV from 16 to 73, with the vast majority of newly contributed sequences derived from *Pteropus alecto* flying foxes, the natural reservoir hosts for HeV. The authors also obtain three isolates of HeV from their recovered samples (though the paper shares limited information about the virus isolates other than their sequences). After sequencing, the authors build a series of phylogenetic trees to explore the HeV genetic diversity in time and space. They identify a clear molecular clock signal in the data and highlight evidence of 4 co-circulating HeV clades with no clear spatial structuring. This is clearly important work, providing commendable genomic contributions to the literature. However, the conclusions and claims of the paper feel a bit overstated to me. Key concerns include:

1. Overstated title: The manuscript title, 'Spatio-temporal dynamics of Hendra virus in *Pteropus* bats in Australia reveals high evolutionary diversity linked with spillover,' made me think the paper would share evidence of expanded HeV diversity in reservoir bats associated with the timing and location of spillover events to horses— but this is not the case. Indeed, no clear links to spillover are actually demonstrated in this paper. A more accurate and understated title would be more appropriate.

2. Overstated abstract: similar to above, the abstract claims that the paper provides "crucial insights into how bats generate and maintain their extraordinary viral diversity, with direct implications for zoonotic disease emergence and pandemic threats". In truth, it does not really offer any explanation for how bats generate and maintain viral diversity, nor does it link this diversity to cross species emergence or pandemic threat. These are big claims that need to be better supported if they are to be made.

3. Both the abstract and the discussion argue that the absence of identifiable spatiotemporal genotypic structuring in the HeV sequences generated here provides support for a hypothesis of persistent infection of HeV in bat hosts and contrasts hypotheses of lineage turnover and selection through immune evasion. Given the sparse number and frequency of genomes recovered in time and space, I wonder if the data are really strong enough to say this? Certainly, the authors do find evidence of a molecular clock structure in their phylogeny, which suggests some role for virus evolution and possible immune escape. I would expect deeper tMRCA roots for a population-level phylogeny of persistent infections especially for hosts as long-lived as bats (e.g. as seen with population level phylogenies for the chronic pathogen, HIV). I would also expect that persistent infections would muddle the temporal signal in root-to-tip phylogenies because some bats would be sampled earlier vs. later in their infection history, with different degrees of within-host evolution. Some kind of forward simulation phylodynamic model with comparison against the data will be needed to really make a claim this grandiose. At a minimum, the role for within-host virus evolution in muddling the phylogenetic inference should be discussed.

4. Structural modeling seems out of place: While only a minor concern, Fig. 4 and the AlphaFold structural predictions felt a bit disconnected from the rest of the manuscript, especially considering that the conclusions of this modeling work are fairly muted, as HeV is a known pathogen, and differences between the strains identified here and those previously identified in GenBank is fairly minimal.

5. Some additional clarity in some of the methods would also be helpful. For example, the authors note in the results that *P. alecto* genome segments recovered from the pooled urine sampling suggest this bat species to be the host of the new HeV sequences --but, unless I missed it, I could not find any additional methods or descriptors about how the host inference was made. At a different point, the authors also mention sequencing some new horse genomes but these are not clearly identified as new contributions in the rest of the manuscript.

6. Finally, most of the inference presented here is derived from maximum likelihood phylogenies, though the authors do include a BEAST analysis in main text Fig. 3. I wish these two approaches had been better integrated: for example, in Supp. Fig 5., the authors show some possible migration paths between localities and strains, which appears to be derived from the ML analysis. There are more robust ways to undertake phylogeographic analysis and joint tree and trait-based inference in BEAST. Given the small number of genomes in the dataset, it is unclear why these approaches were not attempted here.

Line by line comments are as follows:

L49: suggestion to break this into two sentences – it is a bit of a mouthful as is

L77: suggestion to clarify that the HeV genome encodes 6 structural proteins; V,W, and C are also encoded but not mentioned here.

L97: suggestion to specify that this natural reservoir refers to bats (specifically *Pteropus* spp. flying foxes)

L121: Was *P. alecto* the only bat identified on these sheets or just the most common? Unless I missed it, methods for how these host assignments were made seem to be missing from the paper.

L122: The text mentions 10 novel HeV sequences from horses – where do these pop up in the rest of the analyses and in the figures? It would be nice to understand where they fall in the phylogenies.

L136: Is Supp Fig 5 really what you want to reference here? It is not a phylogeny

L171: Are there other horse genomes elsewhere?

L234: See main comment above -- Is the spatial-temporal extent of sampling sufficient to truly evaluate this hypothesis?

L254: The reference paper does show that NiV clusters spatially at large scales... Is it possible that the region sampled here for HeV is too small to demonstrate this spatial structuring given the large ranges of these bats?

L269: Has it been shown that HeV uses ephrin B1/B2 receptors in horses and bats? Some discussion of the ortholog receptors in these hosts would be helpful since these are the source of most of the sequences.

Figures and tables:

- What nucleotide or amino acid substitution model was used for the different trees? Fig 1 and Supp Fig 3, 4, 5. I don't see this listed in the caption or reported in the methods.

- Fig 1: I think there is an error in the caption for panel C – the symbol is a purple star, not a triangle for clade D. Also in panel d, it is fine to have the stylized prevalence in the black line but some label on the y-axis would be helpful. Also, it would help to the year of sampling for the horse and human sequences in panel a – for example the text discusses the 2009 vs. 2014 horse sequences in clade A but it is challenging to figure out which is which. Finally, could you indicate which sequences come from this study and which are from GenBank?

- Fig 3. Does this include horse and human sequences? If not, why not?

- Supp Table 2 : where are the NCBI accession numbers? These should be listed prior to publication

- Supp Fig 2: Is this the average nucleotide identity between all of the 73 genomes combined? It would be more helpful to see how these differ across host species (e.g. bat, horse, human) and possibly roost site. What do the dashed horizontal lines correspond to?

- Supp Fig 5d. The colors are too faint to make it possible to read the migration paths. Also, how were these inferred? I don't see phylogeographic methods anywhere.

- Supplementary text – need to define IDR

- Fig. 10 and 11 are a bit blurry in the file I was provided.

Decision Letter:

30th April 2025

Dear Vincent,

Thank you for your patience while your manuscript "Spatio-temporal dynamics of Hendra virus in Pteropus bats in Australia reveals high evolutionary diversity linked with spillover" was under peer-review at Nature Microbiology. It has now been seen by 3 referees, whose expertise and comments you will find at the end of this email. Although they find your work of some potential interest, they have raised a number of concerns that will need to be addressed before we can consider publication of the work in Nature Microbiology.

In particular, the referees request phenotypic validation, as well as more in depth analysis of the virus evolution aspects.

Should further experimental data allow you to address these criticisms, we would be happy to look at a revised manuscript.

Please include a data availability statement as a separate section after Methods but before references, under the heading "Data Availability". This section should inform readers about the availability of the data used to support the conclusions of your study. This information includes accession codes to public repositories (data banks for protein, DNA or RNA sequences, microarray, proteomics data etc...), references to source data published alongside the paper, unique identifiers such as URLs to data repository entries, or data set DOIs, and any other statement about data availability. At a minimum, you should include the following statement: "The data that support the findings of this study are available from the corresponding author upon request", mentioning any restrictions on availability. If DOIs are provided, we also strongly encourage including these in the Reference list (authors, title, publisher (repository name), identifier, year). For more guidance on how to write this section please see: <http://www.nature.com/authors/policies/data/data-availability-statements-data-citations.pdf>

* If you have not done so already we suggest that you begin to revise your manuscript so that it conforms to our Article format instructions at <http://www.nature.com/nmicrobiol/info/final-submission>. Refer also to any guidelines provided in this letter.

When submitting the revised version of your manuscript, please pay close attention to our [href="https://www.nature.com/nature-portfolio/editorial-policies/image-integrity">Digital Image Integrity Guidelines](https://www.nature.com/nature-portfolio/editorial-policies/image-integrity) and to the following points below:

EXTENDED DATA FIGURES

Link Redacted

Note: This url links to your confidential homepage and associated information about manuscripts you may have submitted or be reviewing for us. If you wish to forward this e-mail to co-authors, please delete this link to your homepage first.

Nature Microbiology is committed to improving transparency in authorship. As part of our efforts in this direction, we are now requesting that all authors identified as 'corresponding author' on published papers create and link their Open Researcher and Contributor Identifier (ORCID) with their account on the Manuscript Tracking System (MTS), prior to acceptance. This applies to primary research papers only. ORCID helps the scientific community achieve unambiguous attribution of all scholarly contributions. You can create and link your ORCID from the home page of the MTS by clicking on 'Modify my Springer Nature account'. For more information please visit www.springernature.com/orcid.

If you wish to submit a suitably revised manuscript we would hope to receive it within 6 months. If you cannot send it within this time, please let us know. We will be happy to consider your revision, even if a similar study has been accepted for publication at Nature Microbiology or published elsewhere (up to a maximum of 6 months).

Yours sincerely,

Reviewer Expertise:

Referee #1: Henipaviruses
Referee #2: Structural Biology
Referee #3: Virus ecology and evolution

Reviewer Comments:

Reviewer #1 (Remarks to the Author):

This manuscript entitled 'Spatio-temporal dynamics of Hendra virus in Pteropus bats in Australia reveals high evolutionary diversity linked with spillover' by C.K. Yinda et al. closes a remarkable knowledge gap regarding the spatiotemporal evolution of Hendra virus lineages in bats, horses and humans in the field. The analysis was done thoroughly by analyzing almost 7000 bat and the available horse samples, and the results are presented and discussed clearly. It is a pity that a lot of also highly valuable data had to be added as Supporting Information. My main concerns regarding the presentation is that due to lengths constraints, some information that would be very useful to understand the context are presented too briefly, which can easily be corrected. Secondly, a number of possible explanations for spillover, mutation and transmission events are excluded throughout the manuscript without offering clear hypothesis as to how the real mechanisms may be composed.

My comments in particular:

Abstract:

52-54: I would prefer to read a thesis how shedding events do occur, instead of reading how they, based on these data, do not occur.

54-56: This conclusion is not very well based on specific data from this study. When presenting a conclusion like this, I would expect possible mechanisms to be presented in the discussion, which are lacking or not stressed sufficiently clearly.

Significance statement:

62-64: again, it is stated how spillover events are not driven, but a data-based hypothesis on how they occur would be much more helpful here.

64-67: the perspective on improving our options for outbreak management and prevention read very nicely, but how exactly do these data translate into management options. I would prefer reading clearer statements here.

Introduction:

74-81: I would suggest to also briefly mention Nipah virus, just to give a broader context.

82-84: mention that the g2 variants were first detected after the sampling for this study was finished, otherwise it does not become clear why HeV-g2 was excluded from this study.

89: delete 'hundreds of' (repetition)

Results:

130 ff: add a short explanation as to how the clades A, B, C, D were ordered, it is not chronological.

212: LD needs to be explained here, explanation follows in line 267

283: 'our results suggest'... (delete s)

293: again, we learn what is not happening 'implies that immune pressure is insufficient to drive the evolution of these strains'. I cannot see the data supporting this in this manuscript, and as stated above, I would much rather see hypotheses highlighting the possible mechanisms underneath the observations.

Methods:

312-313: it is stated 'but other species were also present', how many samples were exactly analyzed from other bat species?

353-361 and 378-388: why were two different sequencing approaches used for bat and horse samples?

404: as reference, 'the oldest genome of HeV sampled from a human in our library' was used, why not the oldest genome in general?

415-416: this is a repetition to what has been described in lines 189-193.

Figure 1: I was wondering why only three human sequences were included in this analysis. With four human fatal cases, are there only three sequences available?

Reviewer #2 (Remarks to the Author):

Major Concerns:

1. Sampling Strategy Limitations

Given the profound influence of natural environments on viral evolution, please clarify: 1) Why sampling was restricted to the eastern coastline. 2) Whether non-urine samples (e.g., blood) were considered for collection, especially given the low viral loads (Ct values >32) in most HeV-positive samples.

2. HeV-g2 Subtype Analysis Omission

Since the study team previously reported HeV-g2 spillover events (Refs 16 and 17), please explicitly address: 1) Whether HeV-g2 was detected in the current samples. If not, is this due to geographic or host restrictions? 2) If detected, please clarify its phylogenetic classification (e.g., as an independent clade or subclade).

3. Phylogenetic Divergence of Spillover Sequences

Most spillover sequences from horses/humans (13/21) fall outside the four main clades (Fig. 1a). It is recommended to calculate host-specific selection pressure and analyze differences in key mutations between bat-derived and spillover strains.

4. Insufficient Mutation Annotation and Phenotypic Correlation

1) What are the mutation profiles of the other three structural proteins besides P/L/G?

2) The two LD mutations identified in the G protein, S175T and A336T, although not located at the G-EB2/3 interaction interface, may affect viral function by influencing the expression efficiency and conformation of G tetramers or the interaction with the F protein.

3) More LD mutations were observed in the L and P proteins. Given the recently solved structure of the NiV L-P complex (Cell. 2025 Feb 6;188(3):688-703.e18.), the authors should provide a more detailed discussion on the potential impacts of these mutations.

4) In-depth analysis of the effects of mutations on viral virulence, cell tropism, and immune evasion is critical for assessing the public health significance of these variants.

5. Fig. 1c is not cited in the Main Text

Minor Issues:

1. Replace hyphens with en-dashes for ranges (e.g., "Suppl. Fig. 9–11").

2. Line 44: Replace the comma after "genetic diversity" with a semicolon in the keywords section.

3. Line 156: Add "is" after "Clade B".

4. Line 194: Correct "clade C and D" to "clades...".

5. Line 221: Add "in" before "Australia".

6. In Fig.3, there is a free node between the 2000–2010 grid lines.

Reviewer #3 (Remarks to the Author):

In this paper, the authors generate critical genomic resources for the bat-borne zoonosis, Hendra virus, increasing the total number of available full genome sequences for HeV from 16 to 73, with the vast majority of newly contributed sequences derived from *Pteropus alecto* flying foxes, the natural reservoir hosts for HeV. The authors also obtain three isolates of HeV from their recovered samples (though the paper shares limited information about the virus isolates other than their sequences). After sequencing, the authors build a series of phylogenetic trees to explore the HeV genetic diversity in time and space. They identify a clear molecular clock signal in the data and highlight evidence of 4 co-circulating HeV clades with no clear spatial structuring. This is clearly important work, providing commendable genomic contributions to the literature. However, the conclusions and claims of the paper feel a bit overstated to me. Key concerns include:

1. Overstated title: The manuscript title, 'Spatio-temporal dynamics of Hendra virus in *Pteropus* bats in Australia reveals high evolutionary diversity linked with spillover,' made me think the paper would share evidence of expanded HeV diversity in reservoir bats associated with the timing and location of spillover events to horses— but this is not the case. Indeed, no clear links to spillover are actually demonstrated in this paper. A more accurate and understated title would be more appropriate.

2. Overstated abstract: similar to above, the abstract claims that the paper provides "crucial insights into how bats generate and maintain their extraordinary viral diversity, with direct implications for zoonotic disease emergence and pandemic threats". In truth, it does not really offer any explanation for how bats generate and maintain viral diversity, nor does it link this diversity to cross species emergence or pandemic threat. These are big claims that need to be better supported if they are to be made.

3. Both the abstract and the discussion argue that the absence of identifiable spatiotemporal genotypic structuring in the HeV sequences generated here provides support for a hypothesis of persistent infection of HeV in bat hosts and contrasts hypotheses of lineage turnover and selection through immune evasion. Given the sparse number and frequency of genomes recovered in time and space, I wonder if the data are really strong enough to say this? Certainly, the authors do find evidence of a molecular clock structure in their phylogeny, which suggests some role for virus evolution and possible immune escape. I would expect deeper tMRCA roots for a population-level phylogeny of persistent infections especially for hosts as long-lived as bats (e.g. as seen with population level phylogenies for the chronic pathogen, HIV). I would also expect that persistent infections would muddle the temporal signal in root-to-tip phylogenies because some bats would be sampled earlier vs. later in their infection history, with different degrees of within-host evolution. Some kind of forward simulation phylodynamic model with comparison against the data will be needed to really make a claim this grandiose. At a minimum, the role for within-host virus evolution in muddling the phylogenetic inference should be discussed.

4. Structural modeling seems out of place: While only a minor concern, Fig. 4 and the AlphaFold structural predictions felt a bit disconnected from the rest of the manuscript, especially considering that the conclusions of this modeling work are fairly muted, as HeV is a known pathogen, and differences between the strains identified here and those previously identified in GenBank is

fairly minimal.

5. Some additional clarity in some of the methods would also be helpful. For example, the authors note in the results that *P. alecto* genome segments recovered from the pooled urine sampling suggest this bat species to be the host of the new HeV sequences --but, unless I missed it, I could not find any additional methods or descriptors about how the host inference was made. At a different point, the authors also mention sequencing some new horse genomes but these are not clearly identified as new contributions in the rest of the manuscript.

6. Finally, most of the inference presented here is derived from maximum likelihood phylogenies, though the authors do include a BEAST analysis in main text Fig. 3. I wish these two approaches had been better integrated: for example, in Supp. Fig 5., the authors show some possible migration paths between localities and strains, which appears to be derived from the ML analysis. There are more robust ways to undertake phylogeographic analysis and joint tree and trait-based inference in BEAST. Given the small number of genomes in the dataset, it is unclear why these approaches were not attempted here.

Line by line comments are as follows:

L49: suggestion to break this into two sentences – it is a bit of a mouthful as is

L77: suggestion to clarify that the HeV genome encodes 6 structural proteins; V,W, and C are also encoded but not mentioned here.

L97: suggestion to specify that this natural reservoir refers to bats (specifically *Pteropus* spp. flying foxes)

L121: Was *P. alecto* the only bat identified on these sheets or just the most common? Unless I missed it, methods for how these host assignments were made seem to be missing from the paper.

L122: The text mentions 10 novel HeV sequences from horses – where do these pop up in the rest of the analyses and in the figures? It would be nice to understand where they fall in the phylogenies.

L136: Is Supp Fig 5 really what you want to reference here? It is not a phylogeny

L171: Are there other horse genomes elsewhere?

L234: See main comment above -- Is the spatial-temporal extent of sampling sufficient to truly evaluate this hypothesis?

L254: The reference paper does show that NiV clusters spatially at large scales... Is it possible that the region sampled here for HeV is too small to demonstrate this spatial structuring given the large ranges of these bats?

L269: Has it been shown that HeV uses ephrin B1/B2 receptors in horses and bats? Some discussion of the ortholog receptors in these hosts would be helpful since these are the source of most of the sequences.

Figures and tables:

- What nucleotide or amino acid substitution model was used for the different trees? Fig 1 and Supp Fig 3, 4, 5. I don't see this listed in the caption or reported in the methods.

- Fig 1: I think there is an error in the caption for panel C – the symbol is a purple star, not a triangle for clade D. Also in panel d, it is fine to have the stylized prevalence in the black line but some label on the y-axis would be helpful. Also, it would help to the year of sampling for the horse and human sequences in panel a – for example the text discusses the 2009 vs. 2014 horse sequences in clade A but it is challenging to figure out which is which. Finally, could you indicate which sequences come from this study and which are from GenBank?

- Fig 3. Does this include horse and human sequences? If not, why not?

- Supp Table 2 : where are the NCBI accession numbers? These should be listed prior to publication

- Supp Fig 2: Is this the average nucleotide identity between all of the 73 genomes combined? It would be more helpful to see how these differ across host species (e.g. bat, horse, human) and possibly roost site. What do the dashed horizontal lines correspond to?

- Supp Fig 5d. The colors are too faint to make it possible to read the migration paths. Also, how were these inferred? I don't see phylogeographic methods anywhere.

- Supplementary text – need to define IDR

- Fig. 10 and 11 are a bit blurry in the file I was provided.

Version 1:

Reviewer comments:

Reviewer #1

(Remarks to the Author)

I wish to thank the authors for their thorough revision of this manuscript. This revision has greatly increased the clarity and the relevance of this manuscript.

I can confirm that all my suggestions and also the other reviewers' suggestions have been fully addressed. I have no further suggestions to make for this manuscript.

Reviewer #2

(Remarks to the Author)

I would like to commend you for your thorough and thoughtful responses to my comments. The revisions and additional explanations have significantly improved the manuscript, addressing all the points I raised in a clear and satisfactory manner. The paper is now stronger and more compelling. I have no further concerns and am pleased to recommend its acceptance in its current form. Congratulations on your excellent work.

Reviewer #3

(Remarks to the Author)

This is my second review of this manuscript, for which major revisions were requested in April of this year. The authors have done a commendable job addressing the substantial critiques of all three reviewers, in particular (1) extending the analysis to include a thorough address of the possible functional consequences of observed amino acid substitutions in the different HeV clades identified, as well as (2) thoroughly characterizing the replication kinetics of novel HeV isolates recovered in this paper and the corresponding innate immune responses in human, horse, and bat cell lines. In addition, (3) the authors included a more thorough phylogeographic analysis in BEAST. The paper is generally in excellent shape, though there are a few remaining issues that need addressing:

1. I appreciate the authors' careful consideration of my concerns regarding the overstated claims of the first version of the manuscript, specifically with respect to an inference of support for a hypothesis of persistent infection. Nonetheless (and here, I am clearly in disagreement with Reviewer 1), there are a few additional places where I find the text to still be overstated—particularly the Significance Statement (L63-67) and the Discussion (L326-365). Indeed, though R1 requests that the authors double down on a single hypothesis, I think that the most the authors can do with the data at hand is to rule out the hypothesis of repeated immune escape and selection. The authors reference citation #38 (Grenfell et al. 2004) where the concept of phylodynamics is first presented; however, they appear to offer two options only: a persistent infection hypothesis and an acute infection hypothesis. In actuality, Grenfell et al. offer three (four if immune enhancement is included), where two types of acute infection are recognized: acute infections with strong cross-immunity (e.g. measles) and acute infections with partial cross immunity (e.g. flu). While I agree that the persistence of strains for long time horizons seems to rule out the flu-like hypothesis, this does not guarantee that this is evidence for persistent infections without repeated sequences from resampled individuals. Indeed, as mentioned in my previous reviews, the clocklike signal is more pronounced in the phylogeny and the tMRCA between sequences more shallow than I would expect if that were the case—unless within-host evolution is extremely slow in persistently infected bats (which may be but we do not know).

Indeed, Grenfell et al. explicitly state with measles-like viruses that "...an immune response that is equally potent against all strains will not generate selection. Therefore, many strains coexist, with relative frequencies determined predominantly by nonselective epidemiological processes. This does not exclude the sporadic occurrence of selection, immunologically mediated or otherwise. Rather, the measles phylogeny indicates that selection is not operating sufficiently consistently to leave its imprint. Instead, the phylogeny is determined by global spatiotemporal strain dynamics: Some lineages persist in regions with low vaccination coverage, whereas others are globally distributed and represent localized outbreaks initiated by imported strains in regions with higher coverage. The high infectiousness of measles permits the rapid geographic spread of these strains, and their phylogenetic lineages reveal substantial spatial mixing."

Given above, I would recommend that the authors tone down the emphasis on persistent infections in the Significance Statement and Discussion and present this as a possibility but not a certainty.

2. I very much appreciate the addition of detailed in vitro characterization of the HeV isolates in this new version of the manuscript. This is important work that fills much-needed gaps in the field. However, I am confused as it says on L118-119 that three bat samples generated isolates, while for the rest of the paper, only two HeV-71 and HeV-84 are characterized. What happened to the third isolate? It is fine if it was not viable for some reason, but this discrepancy needs to be explained.

3. I remain confused about the methods used for species identification from the NGS data. On lines L514-516, the authors address this vaguely, including a citation of a paper that tests different approaches for species ID from NGS data; however, it would be helpful to have some more concrete information about what was done here. It seems that reads were pulled from sequencing, BLASTed against a cytb reference sequence set, and the highest frequency host match was assigned to the sampling session in question. However, no information is given for thresholds in frequency of host match that were used to make species calls. I'd love to see these 'visualizations' as supplementary figures with this paper.

4. Perhaps this can't be helped (e.g. if this a journal requirement), but I found it very challenging to page back and forth among three different localities for the figures and tables: the main text, the extended data, and the supplementary data. Would it be possible to combine the non-main text information into one alternative location rather than two?

Minor, line-by-line comments are as follows:

L67: As mentioned above, I appreciate the authors' attempt to tone down the language on the persistent infection hypothesis; however, some of that language is still maintained in the Significance statement here, e.g.: "surveillance should be prioritized during periods of ecological stress when shedding from persistent infections is most likely". I suggest to back off that hypothesis here as well

L83: clarify that these 68 spillovers correspond to spillovers to horse

L132-134: I agree it appears that the bat sequences are more heterogeneous than human or horse, and this edited extra figure is exciting and helpful. However, it would be stronger, given variable numbers of sequences in each dataset, if you could say this with some statistical certainty? Confidence intervals on the mean identity in the figure and/or some statistics to pair with this statement would help.

L144: Is Supp Table 3 the correct reference here? I am not sure I understand the connection. Maybe this should be Extended Data Table 2?

L182-183: It is not very clear from the current wording that this refers to physical movement of HeV across the landscape. Please reword this sentence to make it clear what is being tested and why.

L254: As mentioned above, I am confused by the sudden focus on two new HeV strains? I thought there were supposed to be three

L306: typo – "HeV-positives samples"

L312-314: As mentioned above, this is not a correct interpretation of Grenfell et al, as it is certainly possible for immunizing viruses (e.g. measles) to maintain many clades simultaneously. Sure, this is not a ladder-like arms race but it does not guarantee a persistent infection either. Also, this sentence appears to have some extra words – "due to" should be removed

L314: typo – should read "replacing one another"

L319: typo – extra comma

L332-334: As noted above, please remember that acute infection is not the same as immune escape – e.g. both flu and measles are acute but have different phylogenies

L337-339: Yes, thank you – the authors offer the alternative, measles-like hypothesis here but it is largely ignored for the rest of the discussion.

L359-361: I would caution against discounting the hypothesis of species-specific structuring, as HeV does infect multiple species of Australian Pteropus, and sequences of only one are analysed here. Isn't it possible the HeV phylogeny is also structured by species and that you just lack the data needed to evaluate this here?

L410: typo "specie"

L469: typo – missing closing parenthesis

L514-516: why were these host read visualizations not also included as supplementary materials in the paper? It would be helpful to have some actual methods here rather than the citation that is provided. It seems that reads were pulled from sequencing, BLASTed against a cyt b reference sequence set, and the highest frequency host match was assigned to the sampling session in question. However, no information is given for thresholds in frequency of host match that were used to make species calls

Extended Data Fig 3: Some of the sequences list a sample number (e.g. HORSE003) rather than an accession number. While searching GenBank with the sample number does seem to pull up the right sequences, for consistency, I suggest that the phylogeny should identify them all by accession number.

Extended Data Fig 5 – Apologies if I misunderstood the permutation analysis but can you explain why the proportions of permuted samples $>$ and $<$ the observed value do not add up to 1?

Extended Data Fig 6 and 7 – The order of these should be switched to reflect the order they are presented in the main text of the paper.

Extended Data Fig. 6 – Please indicate the cell type somewhere in the plot, either on the left or right side of rows a/b/c. Additionally, it would be more intuitive to also reverse this plot, with cell type in the columns and gene expression in the rows, since this is how it is oriented in the complementary Fig 2 f the main text.

Fig 6: In the main text, the authors refer to the 'prototype Hendra' strain, but in the figure here (and in Extended Fig. 6) they refer to it as 'Hendra horse'. The same language should be used in both the main text and the figure to avoid confusion.

Supplement

L44: references supplementary Fig. 8 which does not appear to exist

Decision Letter:

1st October 2025

Dear Vincent,

Thank you for your patience while your manuscript "Spatio-temporal dynamics of Hendra virus in Pteropus bats in Australia reveals stable maintenance of diverse viral clades" was under peer-review at Nature Microbiology. It has now been seen by 3 referees, whose expertise and comments you will find at the of this email. You will see from their comments below that while they find your work of interest, some important points are raised. We are very interested in the possibility of publishing your study in Nature Microbiology, but would like to consider your response to these concerns in the form of a revised manuscript before we make a final decision on publication.

In particular, you will see that referee #3 asks to tone down your statements on persistence in the discussion, as well as to provide more details in the methods. The rest referees' reports are clear and the remaining issues should be straightforward to address.

If you have not done so already please begin to revise your manuscript so that it conforms to our Article format instructions at <http://www.nature.com/nmicrobiol/info/final-submission/>

The usual length limit for a Nature Microbiology Article is six display items (figures or tables) and 3,000 words. We have some flexibility, and can allow a revised manuscript at 3,500 words, but please consider this a firm upper limit. There is a trade-off of ~250 words per display item, so if you need more space, you could move a Figure or Table to Supplementary Information.

Some reduction could be achieved by focusing any introductory material and moving it to the start of your opening 'bold' paragraph, whose function is to outline the background to your work, describe in a sentence your new observations, and explain your main conclusions. The discussion should also be limited. Methods should be described in a separate section following the discussion, we do not place a word limit on Methods.

Nature Microbiology titles should give a sense of the main new findings of a manuscript, and should not contain punctuation. Please keep in mind that we strongly discourage active verbs in titles, and that they should ideally fit within 90 characters each (including spaces).

Please include a data availability statement as a separate section after Methods but before references, under the heading "Data Availability". This section should inform readers about the availability of the data used to support the conclusions of your study. This information includes accession codes to public repositories (data banks for protein, DNA or RNA sequences, microarray, proteomics data etc...), references to source data published alongside the paper, unique identifiers such as URLs to data repository entries, or data set DOIs, and any other statement about data availability. At a minimum, you should include the following statement: "The data that support the findings of this study are available from the corresponding author upon request", mentioning any restrictions on availability. If DOIs are provided, we also strongly encourage including these in the Reference list (authors, title, publisher (repository name), identifier, year). For more guidance on how to write this section please see: <http://www.nature.com/authors/policies/data/data-availability-statements-data-citations.pdf>

To improve the accessibility of your paper to readers from other research areas, please pay particular attention to the wording of the paper's opening bold paragraph, which serves both as an introduction and as a brief, non-technical summary in about 150 words. If, however, you require one or two extra sentences to explain your work clearly, please include them even if the paragraph is over-length as a result. The opening paragraph should not contain references. Because scientists from other sub-disciplines will be interested in your results and their implications, it is important to explain essential but specialised terms concisely. We suggest you show your summary paragraph to colleagues in other fields to uncover any problematic concepts.

If your paper is accepted for publication, we will edit your display items electronically so they conform to our house style and will reproduce clearly in print. If necessary, we will re-size figures to fit single or double column width. If your figures contain several parts, the parts should form a neat rectangle when assembled. Choosing the right electronic format at this stage will speed up

the processing of your paper and give the best possible results in print. We would like the figures to be supplied as vector files - EPS, PDF, AI or postscript (PS) file formats (not raster or bitmap files), preferably generated with vector-graphics software (Adobe Illustrator for example). Please try to ensure that all figures are non-flattened and fully editable. All images should be at least 300 dpi resolution (when figures are scaled to approximately the size that they are to be printed at) and in RGB colour format. Please do not submit Jpeg or flattened TIFF files. Please see also 'Guidelines for Electronic Submission of Figures' at the end of this letter for further detail.

Figure legends must provide a brief description of the figure and the symbols used, within 350 words, including definitions of any error bars employed in the figures.

EXTENDED DATA FIGURES

Please include a statement before the acknowledgements naming the author to whom correspondence and requests for materials should be addressed.

Finally, we require authors to include a statement of their individual contributions to the paper -- such as experimental work, project planning, data analysis, etc. -- immediately after the acknowledgements. The statement should be short, and refer to authors by their initials. For details please see the Authorship section of our joint Editorial policies at http://www.nature.com/authors/editorial_policies/authorship.html

* include a point-by-point response to any editorial suggestions and to our referees. Please include your response to the editorial suggestions in your cover letter, and please upload your response to the referees as a separate document.

* ensure it complies with our format requirements for Letters as set out in our guide to authors at www.nature.com/nmicrobiol/info/gta/

* state in a cover note the length of the text, methods and legends; the number of references; number and estimated final size of figures and tables

* resubmit electronically if possible using the link below to access your home page:

Link Redacted

*This url links to your confidential homepage and associated information about manuscripts you may have submitted or be reviewing for us. If you wish to forward this e-mail to co-authors, please delete this link to your homepage first.

Please ensure that all correspondence is marked with your Nature Microbiology reference number in the subject line.

Nature Microbiology is committed to improving transparency in authorship. As part of our efforts in this direction, we are now requesting that all authors identified as 'corresponding author' on published papers create and link their Open Researcher and Contributor Identifier (ORCID) with their account on the Manuscript Tracking System (MTS), prior to acceptance. This applies to primary research papers only. ORCID helps the scientific community achieve unambiguous attribution of all scholarly contributions. You can create and link your ORCID from the home page of the MTS by clicking on 'Modify my Springer Nature account'. For more information please visit www.springernature.com/orcid.

We hope to receive your revised paper within two weeks. If you cannot send it within this time, please let us know.

Yours sincerely,

Reviewer Expertise:

Referee #1: Henipaviruses
Referee #2: Henipaviruses, structural biology
Referee #3: Virus ecology

Reviewers Comments:

Reviewer #1 (Remarks to the Author):

I wish to thank the authors for their thorough revision of this manuscript. This revision has greatly increased the clarity and the relevance of this manuscript.

I can confirm that all my suggestions and also the other reviewers' suggestions have been fully addressed. I have no further suggestions to make for this manuscript.

Reviewer #2 (Remarks to the Author):

I would like to commend you for your thorough and thoughtful responses to my comments. The revisions and additional explanations have significantly improved the manuscript, addressing all the points I raised in a clear and satisfactory manner. The paper is now stronger and more compelling. I have no further concerns and am pleased to recommend its acceptance in its current form. Congratulations on your excellent work.

Reviewer #3 (Remarks to the Author):

This is my second review of this manuscript, for which major revisions were requested in April of this year. The authors have done a commendable job addressing the substantial critiques of all three reviewers, in particular (1) extending the analysis to include a thorough address of the possible functional consequences of observed amino acid substitutions in the different HeV clades identified, as well as (2) thoroughly characterizing the replication kinetics of novel HeV isolates recovered in this paper and the corresponding innate immune responses in human, horse, and bat cell lines. In addition, (3) the authors included a more thorough phylogeographic analysis in BEAST. The paper is generally in excellent shape, though there are a few remaining issues that need addressing:

1. I appreciate the authors' careful consideration of my concerns regarding the overstated claims of the first version of the manuscript, specifically with respect to an inference of support for a hypothesis of persistent infection. Nonetheless (and here, I am clearly in disagreement with Reviewer 1), there are a few additional places where I find the text to still be overstated—particularly the Significance Statement (L63-67) and the Discussion (L326-365). Indeed, though R1 requests that the authors double down on a single hypothesis, I think that the most the authors can do with the data at hand is to rule out the hypothesis of repeated immune escape and selection. The authors reference citation #38 (Grenfell et al. 2004) where the concept of phylodynamics is first presented; however, they appear to offer two options only: a persistent infection hypothesis and an acute infection hypothesis. In actuality, Grenfell et al. offer three (four if immune enhancement is included), where two types of acute infection are recognized: acute infections with strong cross-immunity (e.g. measles) and acute infections with partial cross immunity (e.g. flu). While I agree that the persistence of strains for long time horizons seems to rule out the flu-like hypothesis, this does not guarantee that this is evidence for persistent infections without repeated sequences from resampled individuals. Indeed, as mentioned in my previous reviews, the clocklike signal is more pronounced in the phylogeny and the tMRCA between sequences more shallow than I would expect if that were the case—unless within-host evolution is extremely slow in persistently infected bats (which may be but we do not know).

Indeed, Grenfell et al. explicitly state with measles-like viruses that "...an immune response that is equally potent against all strains will not generate selection. Therefore, many strains coexist, with relative frequencies determined predominantly by nonselective epidemiological processes. This does not exclude the sporadic occurrence of selection, immunologically mediated or otherwise. Rather, the measles phylogeny indicates that selection is not operating sufficiently consistently to leave its imprint. Instead, the phylogeny is determined by global spatiotemporal strain dynamics: Some lineages persist in regions with low vaccination coverage, whereas others are globally distributed and represent localized outbreaks initiated by imported strains in regions with higher coverage. The high infectiousness of measles permits the rapid geographic spread of these strains, and their phylogenetic lineages reveal substantial spatial mixing."

Given above, I would recommend that the authors tone down the emphasis on persistent infections in the Significance Statement and Discussion and present this as a possibility but not a certainty.

2. I very much appreciate the addition of detailed in vitro characterization of the HeV isolates in this new version of the manuscript. This is important work that fills much-needed gaps in the field. However, I am confused as it says on L118-119 that three bat samples generated isolates, while for the rest of the paper, only two HeV-71 and HeV-84 are characterized. What happened to the third isolate? It is fine if it was not viable for some reason, but this discrepancy needs to be explained.

3. I remain confused about the methods used for species identification from the NGS data. On lines L514-516, the authors address this vaguely, including a citation of a paper that tests different approaches for species ID from NGS data; however, it would be helpful to have some more concrete information about what was done here. It seems that reads were pulled from sequencing, BLASTed against a cyt b reference sequence set, and the highest frequency host match was assigned to the sampling session in question. However, no information is given for thresholds in frequency of host match that were used to make species calls. I'd love to see these 'visualizations' as supplementary figures with this paper.

4. Perhaps this can't be helped (e.g. if this a journal requirement), but I found it very challenging to page back and forth among three different localities for the figures and tables: the main text, the extended data, and the supplementary data. Would it be possible to combine the non-main text information into one alternative location rather than two?

Minor, line-by-line comments are as follows:

L67: As mentioned above, I appreciate the authors' attempt to tone down the language on the persistent infection hypothesis; however, some of that language is still maintained in the Significance statement here, e.g.: "surveillance should be prioritized during periods of ecological stress when shedding from persistent infections is most likely". I suggest to back off that hypothesis here as well

L83: clarify that these 68 spillovers correspond to spillovers to horse

L132-134: I agree it appears that the bat sequences are more heterogeneous than human or horse, and this edited extra figure is exciting and helpful. However, it would be stronger, given variable numbers of sequences in each dataset, if you could say this with some statistical certainty? Confidence intervals on the mean identity in the figure and/or some statistics to pair with this statement would help.

L144: Is Supp Table 3 the correct reference here? I am not sure I understand the connection. Maybe this should be Extended Data Table 2?

L182-183: It is not very clear from the current wording that this refers to physical movement of HeV across the landscape. Please reword this sentence to make it clear what is being tested and why.

L254: As mentioned above, I am confused by the sudden focus on two new HeV strains? I thought there were supposed to be three

L306: typo – "HeV-positives samples"

L312-314: As mentioned above, this is not a correct interpretation of Grenfell et al, as it is certainly possible for immunizing viruses (e.g. measles) to maintain many clades simultaneously. Sure, this is not a ladder-like arms race but it does not guarantee a persistent infection either. Also, this sentence appears to have some extra words – "due to" should be removed

L314: typo – should read "replacing one another"

L319: typo – extra comma

L332-334: As noted above, please remember that acute infection is not the same as immune escape – e.g. both flu and measles are acute but have different phylogenies

L337-339: Yes, thank you – the authors offer the alternative, measles-like hypothesis here but it is largely ignored for the rest of the discussion.

L359-361: I would caution against discounting the hypothesis of species-specific structuring, as HeV does infect multiple species of Australian Pteropus, and sequences of only one are analysed here. Isn't it possible the HeV phylogeny is also structured by species and that you just lack the data needed to evaluate this here?

L410: typo "specie"

L469: typo – missing closing parenthesis

L514-516: why were these host read visualizations not also included as supplementary materials in the paper? It would be helpful to have some actual methods here rather than the citation that is provided. It seems that reads were pulled from sequencing, BLASTed against a cyt b reference sequence set, and the highest frequency host match was assigned to the sampling session in question. However, no information is given for thresholds in frequency of host match that were used to make species calls

Extended Data Fig 3: Some of the sequences list a sample number (e.g. HORSE003) rather than an accession number. While searching GenBank with the sample number does seem to pull up the right sequences, for consistency, I suggest that the phylogeny should identify them all by accession number.

Extended Data Fig 5 – Apologies if I misunderstood the permutation analysis but can you explain why the proportions of permuted samples > and < the observed value do not add up to 1?

Extended Data Fig 6 and 7 – The order of these should be switched to reflect the order they are presented in the main text of the paper.

Extended Data Fig. 6 – Please indicate the cell type somewhere in the plot, either on the left or right side of rows a/b/c. Additionally, it would be more intuitive to also reverse this plot, with cell type in the columns and gene expression in the rows, since this is how it is oriented in the complementary Fig 2 f the main text.

Fig 6: In the main text, the authors refer to the 'prototype Hendra' strain, but in the figure here (and in Extended Fig. 6) they refer to it as 'Hendra horse'. The same language should be used in both the main text and the figure to avoid confusion.

Supplement

L44: references supplementary Fig. 8 which does not appear to exist

Version 2:

Reviewer comments:

Reviewer #3

(Remarks to the Author)

I thank the authors for their thorough address of my commentary and find the manuscript much improved. There are a few small remaining errors that should be corrected prior to publication:

L123: typos in newly added material—should be changed to “Between hosts, there were significant differences...”

L180: run-on sentence---should be made into two sentences or comma should be substituted for a semicolon: “Only a small number of transitions were statistically supported originating in South-East Queensland (Suppl. Tables 5, Extended Data Table 3); these involved very limited data (n=2 genomes) and likely reflect artefacts of sparse sampling.”

L357: Please change “most bat sequences are from Pteropus species” to “Pteropus alecto” – it is possible that HeV clades segregate by host species within the Pteropus genus, but since your samples are almost-exclusively derived from P. alecto, you cannot identify that signal with the current dataset. Sampling and sequencing of HeV from the other 3 Pteropus species in Australia would be a valuable contribution to the field.

L44 of the Supporting Information still references “Supp Fig. 8 show the permuted and observed proportions of each clade out of the four clades within each 6-month period” but no Supp Fig 8 is included in the manuscript. The rebuttal suggests this is an erroneous observation, but the most recent collection of materials presented for this review still includes this error.

Decision Letter:

Our ref: NMICROBIOL-25030941B

5th November 2025

Dear Vincent,

Thank you for submitting your revised manuscript "Spatio-temporal dynamics of Hendra virus in Pteropus bats in Australia reveals stable maintenance of diverse viral clades" (NMICROBIOL-25030941B). It has now been seen by the original referees and their comments are below. The reviewers find that the paper has improved in revision, and therefore we'll be happy in principle to publish it in Nature Microbiology, pending minor revisions to satisfy the referees' final requests and to comply with our editorial and formatting guidelines.

If any mandatory data are not currently live at NCBI Sequence Read Archive or Genbank and you are experiencing any slowdown in service due to the US government shutdown then please deposit your data with the European Nucleotide Archive (ENA) or DNA Data Bank of Japan (DDBJ) as alternative repositories within our policies (<https://www.springernature.com/gp/authors/research-data-policy/repositories-mandates/19540364>). Mandatory data will have to be publicly available before acceptance of the manuscript (<https://www.nature.com/nmicrobiol/editorial-policies/reporting->

standards).

Thank you again for your interest in Nature Microbiology Please do not hesitate to contact me if you have any questions.

Sincerely,

Reviewer #3 (Remarks to the Author):

I thank the authors for their thorough address of my commentary and find the manuscript much improved. There are a few small remaining errors that should be corrected prior to publication:

L123: typos in newly added material—should be changed to “Between hosts, there were significant differences...”

L180: run-on sentence---should be made into two sentences or comma should be substituted for a semicolon: “Only a small number of transitions were statistically supported originating in South-East Queensland (Suppl. Tables 5, Extended Data Table 3); these involved very limited data (n=2 genomes) and likely reflect artefacts of sparse sampling.”

L357: Please change “most bat sequences are from Pteropus species” to “Pteropus alecto” – it is possible that HeV clades segregate by host species within the Pteropus genus, but since your samples are almost-exclusively derived from P. alecto, you cannot identify that signal with the current dataset. Sampling and sequencing of HeV from the other 3 Pteropus species in Australia would be a valuable contribution to the field.

L44 of the Supporting Information still references “Supp Fig. 8 show the permuted and observed proportions of each clade out of the four clades within each 6-month period” but no Supp Fig 8 is included in the manuscript. The rebuttal suggests this is an erroneous observation, but the most recent collection of materials presented for this review still includes this error.

Version 3:

Decision Letter:

17th December 2025

Dear Vincent,

I am pleased to accept your Article "Spatio-temporal dynamics of Hendra virus in Australia reveals stable maintenance of diverse viral clades among Pteropus bats" for publication in Nature Microbiology. Thank you for having chosen to submit your work to us and many congratulations.

Authors may need to take specific actions to achieve compliance with funder and institutional open access mandates. If your research is supported by a funder that requires immediate open access (e.g. according to [Plan S principles](https://www.springernature.com/gp/open-science/plan-s-compliance) or the [NIH public access policy](https://www.springernature.com/gp/open-science/us-federal-agency-compliance)) then you should select the gold OA route, and we will direct you to the compliant route where possible. Because authors warrant under our subscription licensing terms that they haven't committed to licensing any version of their article under a licence inconsistent with the terms of our agreement – including the applicable embargo period – publication under the subscription model isn't suitable for authors whose funders require no embargo.

Congrats again to you and your co-authors. I am looking forward to seeing your paper published.

With kind regards,

P.S. Click on the following link if you would like to recommend Nature Microbiology to your librarian
<http://www.nature.com/subscriptions/recommend.html#forms>

** Visit the Springer Nature Editorial and Publishing website at http://editorial-jobs.springernature.com?utm_source=ejP_NMicro_email&utm_medium=ejP_NMicro_email&utm_campaign=ejP_NMicro for more information about our career opportunities. If you have any questions please click [here](mailto:editorial.publishing.jobs@springernature.com).**

Reviewer Expertise:

Referee #1: Henipaviruses

Referee #2: Structural Biology

Referee #3: Virus ecology and evolution

Reviewer Comments:

Reviewer #1 (Remarks to the Author):

1. This manuscript entitled ‘Spatio-temporal dynamics of Hendra virus in Pteropus bats in Australia reveals high evolutionary 1 diversity linked with spillover’ by C.K. Yinda et al. closes a remarkable knowledge gap regarding the spatiotemporal evolution of Hendra virus lineages in bats, horses and humans in the field. The analysis was done thoroughly by analyzing almost 7000 bat and the available horse samples, and the results are presented and discussed clearly. It is a pity that a lot of also highly valuable data had to be added as Supporting Information. My main concerns regarding the presentation is that due to lengths constraints, some information that would be very useful to understand the context are presented too briefly, which can easily be corrected. Secondly, a number of possible explanations for spillover, mutation and transmission events are excluded throughout the manuscript without offering clear hypothesis as to how the real mechanisms may be composed.

Response: We would like to thank the reviewer for these comments. We agree that due to length constraints and the multidisciplinary nature of the work sometimes makes it challenging to capture all the nuances. In addition, we appreciate your recognition of the rigor of our analysis and acknowledge your concerns regarding the need for clearer mechanistic hypotheses. We have tried to address these important points in detail in our responses to the specific comments below.

My comments in particular:

Abstract:

2. 52-54: I would prefer to read a thesis how shedding events do occur, instead of reading how they, based on these data, do not occur.

Response: We have now revised the abstract to focus on the positive findings regarding spillover mechanisms rather than emphasizing what doesn't occur. The sentence now reads:

“The high HeV diversity within shedding pulses and temporal stability of co-circulating clades suggests that viral dynamics are driven by episodic shedding of existing lineages maintained at the population level, rather than immune-driven strain replacement dynamics.”

3. 54-56: This conclusion is not very well based on specific data from this study. When presenting a conclusion like this, I would expect possible mechanisms to be presented in the discussion, which are lacking or not stressed sufficiently clearly.

Response: We appreciate this comment, which has helped to improve the communication of the key findings within the abstract and discussion. The sentence in the abstract has been revised to:

“The high HeV diversity within shedding pulses and temporal stability of co-circulating clades suggests that viral dynamics are driven by episodic shedding of existing lineages maintained at the population level, rather than immune-driven strain replacement dynamics.”

This revision directly highlights the phylogenetic patterns we observed and connects to theoretical frameworks for viral maintenance in reservoir hosts. We have also strengthened the discussion section by adding clearer mechanistic explanations, specifically explaining how the observed patterns (high diversity within pulses, temporal stability across years, lack of strain replacement) align with predictions for viral maintenance rather than acute infection dynamics.

The relevant section in the discussion (347 - 361) now reads:

“At the time, the large-scale genomic surveillance data required to explore these hypotheses did not exist; however, the current dataset allows us to begin to evaluate them. An acute infection model would be expected to produce transient lineage dominance and frequent strain replacement that should yield phylogenies with strong temporal clustering but limited patterns of co-circulation. Instead, although our data shows a robust molecular clock signal, it also reveals long-term coexistence of multiple divergent clades across years, inconsistent with the rapid lineage turnover expected with an acute infection model. These patterns could therefore reflect persistent or low-level infections within individual bats, but they also might be explained by repeated re-introductions within a large, well-mixed metapopulation (Fig 3). Distinguishing between these mechanisms would require explicit phylodynamic modelling and, most importantly, longitudinal sampling of individual bats for within-host data, all which remain major gaps. Our data here supports the observation of population-level stability of HeV lineages, with limited evidence for immune-driven evolution. The persistence of multiple clades alongside episodic shedding pulses points to ecological triggers such as food stress or host condition, while the underlying mechanism, whether extended infections in individual bats or repeated re-infection across a connected metapopulation, remains unresolved.”

Significance statement:

4. 62-64: again, it is stated how spillover events are not driven, but a data-based hypothesis on how they occur would be much more helpful here.

Response: We have now revised the Significance Statement to better reflect our hypotheses on how diverse HeV lineages are stably maintained in bat populations. The sentence now reads:

“The co-circulation of multiple clades within and across years suggests that HeV is maintained within bat populations through persistent infections of diverse viral lineages, creating a stable reservoir from which spillover events can occur.”

5. 64-67: the perspective on improving our options for outbreak management and prevention read very nicely, but how exactly do these data translate into management options. I would prefer reading clearer statements here.

Response: We have now updated the sentence to provide more specific management implications based on our findings. It now reads:

“The results demonstrate the significant genetic diversity of HeV maintained in bats and that surveillance should be prioritized during periods of ecological stress when shedding from persistent infections is most likely, providing a framework for ecologically-informed spillover prevention in bat-virus systems globally.”

Introduction:

6. 74-81: I would suggest to also briefly mention Nipah virus, just to give a broader context.

Response: We have now added Nipah virus in the following sentence:

“Hendra virus (HeV, genus Henipavirus) provides an ideal system for addressing this gap (and a closely related highly pathogenic virus in this genus is Nipah virus (NiV)).

7.82-84: mention that the g2 variants were first detected after the sampling for this study was finished, otherwise it does not become clear why HeV-g2 was excluded from this study.

Response: We reported the screening of g2 variant in a subset of these samples in in Peel, Yinda et al., 2022 (PMID: 35447052). The prevalence was generally low in these samples and unfortunately, we didn't recover full genomes from the few positive samples we identified. A more detailed response to this comment is provided below under Reviewer 2, Comment 2.

8. 89: delete ‘hundreds of’ (repetition)

Response: We have deleted this.

Results:

10. 130 ff: add a short explanation as to how the clades A, B, C, D were ordered, it is not chronological.

Response: The clades were ordered based on genetic similarity using maximum likelihood genetic analyses and using the original 1994 Hendra viruses as an outgroup.

11. 212: LD needs to be explained here, explanation follows in line 267

Response: This is resolved (line 246).

12. 283: ‘our results suggest’... (delete s)

Response: This is resolved.

13. 293: again, we learn what is not happening ‘implies that immune pressure is insufficient to drive the evolution of these strains’. I cannot see the data supporting this in this manuscript, and as stated above, I would much rather see hypotheses highlighting the possible mechanisms underneath the observations.

Response: We appreciate the reviewer’s comment and note that this has been addressed in our responses to comments 2–5 above.

Methods:

14. 312-313: it is stated ‘but other species were also present’, how many samples were exactly analyzed from other bat species?

*Response: We agree with the reviewer that this statement was confusing. When we stated that other species were present, we were referring to bat species visually observed in the bat roosts where the under roost sampling occurred. Additionally, we have now performed sequence-based species identification using cytochrome b sequences obtained from the samples, showing that all the samples were obtained from *Pteropus alecto* (black flying fox). The two isolates (Hendra 71 and Hendra 84) were obtained from culturing HeV positive urine on Vero E6 cells (African Green monkey, *Chlorocebus sabaeus*) and as such we recovered African Green monkey Cyt B sequences for these two isolate derived sequences. We have added this data to the manuscripts in the Methods section (lines 529–533) and in Supplementary Table 2.*

*Line 125 – 126: “Host genomic sequences detected within the sequenced libraries confirmed *P. alecto* as the source of these samples (Suppl. Table 2).”*

15. 353-361 and 378-388: why were two different sequencing approaches used for bat and horse samples?

Response: This is because the bat and the horse samples were processed in different labs and at different times. The bat sequences were generated at the NIAID Rocky Mountain Laboratories in the USA, and the horse sequences were generated at the Australian Centre for Disease Preparedness (ACDP) as part of their diagnostic mandate.

16. 404: as reference, ‘the oldest genome of HeV sampled from a human in our library’ was used, why not the oldest genome in general?

Response: The oldest available horse-derived sequence is MN062017, reported only two months prior to KY425627, which we used in our analysis. The two sequences are nearly identical, differing only by the substitution in the N-terminus of the M protein: asparagine to aspartate at position 31 in MN062017 relative to KY425627. This substitution is discussed in the Supplementary Discussion (lines 164-176), where we note its presence in all genotypes except KY425627 and its convergence with HeV-g2 and NiV. Given the minimal difference, the choice between MN062017 and KY425627 as the reference sequence does not affect any of our results. Furthermore, their close sampling dates do not provide strong evidence on which sequence appeared first in the population.

17. 415-416: this is a repetition to what has been described in lines 189-193.

Response: Line 582-585 gives a detail method:

“Given the apparent temporal structure in the genome-scale alignments data, we then made estimates of the rates of evolutionary change (i.e., nucleotide substitutions per site per year) and the time to most recent common ancestor (TMRCA) using the Bayesian Markov chain Monte Carlo (MCMC) method available in BEAST (version 1.10.4)”

While lines 218-222 state the outcome of the analysis:

“We next used a Bayesian Markov chain Monte Carlo (MCMC) approach to more rigorously assess the evolutionary dynamics of HeV. Based on the best fit model (Suppl. Table 4), the coding sequences showed a refined estimate for the mean rate of evolution of 1.43×10^{-4} (95% HPD, 1.07×10^{-4} to 1.80×10^{-4}) subs/site/year. The time to the most recent common ancestor (TMRCA) estimate was 1975 (95% HPD, 1954.80 to 1988.40) across the whole tree.”

We briefly mention the method at the start of the results section (“We next used a Bayesian Markov chain Monte Carlo (MCMC) approach to more rigorously assess the evolutionary dynamics of HeV”) to provide essential context and improve clarity for readers before presenting the results.

18. Figure 1: I was wondering why only three human sequences were included in this analysis. With four human fatal cases, are there only three sequences available?

Response: The reviewer is correct, there are only three sequences of human origin available in the databases.

Reviewer #2 (Remarks to the Author):

Major Concerns:

1. Sampling Strategy Limitations

Given the profound influence of natural environments on viral evolution, please clarify: 1) Why sampling was restricted to the eastern coastline. 2) Whether non-urine samples (e.g., blood) were considered for collection, especially given the low viral loads (Ct values >32) in most HeV-positive samples.

*Response: Thank you for the opportunity to clarify these points. Firstly, our sampling was concentrated along the eastern coastline for three key reasons: the region (1) encompasses the distributional range of Australian flying foxes, in particular, the overlap in distributions of known HeV reservoir species – the black flying fox (*Pteropus alecto*) and grey-head flying foxes (*P. poliocephalus*), (2) has been shown to have the highest prevalences of HeV in flying foxes (PMCID: PMC4666458, 36277112), and (3) is a hotspot of HeV spillover risk in horses (encompassing 47 of 58 documented HeV spillover events). This coastal corridor contains the majority of significant flying fox roosts and represents the critical ecological interface where human-animal-environment interactions create spillover risk.*

We have added clarification to the Methods section (lines 457-462) to explaining that our sampling region encompasses the primary distributional range of HeV reservoir species and the major spillover risk corridor in Australia.

“This area encompasses the distributional overlap of key HeV reservoir species, contains the highest HeV prevalences in flying foxes, and represents a hotspot of spillover risk (encompassing 47 of 58 documented HeV spillover events)”

Secondly, we prioritized urine collection because HeV demonstrates renal tropism, with urine representing the primary route of viral excretion in naturally infected flying foxes (PMID: 26469523). This tissue tropism appears consistent across related henipaviruses (PMID: 40554741). Consequently, urine sampling provides the most reliable method for detecting actively shedding individuals and understanding transmission dynamics within roosts. Additionally, this provides benefits over blood in that it can be collected efficiently and non-invasively from under-roost sampling, enabling larger sample sizes for population-level monitoring.

The observed low viral loads (Ct values >32 in most samples) are consistent with previous studies on natural ecology of HeV infections in flying foxes, where viral loads are typically low and individuals do not show clinical signs. Out of serum, packed hemocytes, urine, rectal, nasal and oral samples, the lowest Ct values (highest viral load) were observed in urine (min 22.26, mean 32.56), with blood samples typically having higher Ct values within the same individuals (min Ct 33.53 and 31.70, mean Ct 35.01 and 36.66, for serum and hemocyte respectively)

(PMID: 26625128).

We have also added clarification to the Methods section explaining our focus on urine sampling based on HeV's renal tropism and superior viral detection compared to blood samples (lines 459-462):

“Urine was prioritized for sample collection and screening because HeV demonstrates renal tropism, with urine representing the primary route of viral excretion in naturally infected flying foxes and typically yielding the lowest Ct values (highest viral loads) compared to blood samples from the same individuals”

2. HeV-g2 Subtype Analysis Omission

Since the study team previously reported HeV-g2 spillover events (Refs 16 and 17), please explicitly address: 1) Whether HeV-g2 was detected in the current samples. If not, is this due to geographic or host restrictions? 2) If detected, please clarify its phylogenetic classification (e.g., as an independent clade or subclade).

*Response: We have previously studied HeV-g2 in a subset of the samples analyzed in this study, comprising 4,539 pooled urine samples collected from 129 underroost sampling sessions and 1,674 urine samples collected from individual bats over 39 catching sessions during July 2017–September 2020 (PMID: 35447052). In that study, only eight pooled urine samples and 2 samples from individual flying foxes tested positive for HeV-g2 and no full genomes were able to be sequenced. The significantly lower prevalence of HeV-g2 than HeV-g1 could indicate actual lower prevalence in the sampled population, or a species bias between the two variants, given our primary focus on collecting urine samples from *P. alecto* in our sampling design.*

Regarding phylogenetic classification, HeV-g2 represents a distinct genotype rather than a clade within HeV-g1, sharing only 84% genome-wide nucleotide identity with HeV-g1 (PMID: 35202527) compared to the ~96-100% identity within HeV-g1 identified here.

We have added clarification to the Methods section explaining why HeV-g2 was excluded from this analysis (lines 485-487).

“HeV-g2 was previously detected in a subset of these samples at significantly lower prevalence (0.16%¹⁶), however no full genomes could be recovered for phylogenetic analyses, so this study focused exclusively on HeV-g1.”

3. Phylogenetic Divergence of Spillover Sequences

Most spillover sequences from horses/humans (13/21) fall outside the four main clades (Fig. 1a). It is recommended to calculate host-specific selection pressure and analyze differences in key mutations between bat-derived and spillover strains.

Response: While this is an interesting avenue, we feel it is beyond the scope of the current study. In particular, robust inference of host-specific selection would require balanced and well-

powered sample sizes across host species, ideally with multiple independent lineages per host. In our dataset, the number of available genomes from horses and especially humans is very limited, which would constrain the statistical power and interpretability of such analyses. Given the exploratory nature of our current study and the limited sampling in some host groups, we have chosen to focus on comparative genomic and phylogenetic features that are more directly supported by the available data. However, we agree that future studies with expanded sampling could usefully address host-specific selection dynamics.

4. Insufficient Mutation Annotation and Phenotypic Correlation

Response: We appreciate the reviewer's suggestions of expanding the analysis of the mutations and their correlation with phenotypic characteristics. In response, we conducted a detailed analysis of recurrent mutations in our dataset, using structural information and priori literature to evaluate their potential functional impact (lines 245-267, 382-398, and Supplementary Discussion: lines 52-299). Our analysis indicates that a subset of the LD and non-LD mutations identified in the P/V/W, G, F, and L proteins might contribute to major or subtle functional differences in viral biology and host interactions. These mutations are discussed extensively in the Supplementary Material due to word limit constraints, but we have added a new table (Table 1) summarizing the mutations most likely to have functional significance.

In addition, we evaluated the functional implications of genetic divergence with in vitro assays. Specifically, we assessed replication fitness, innate immune responses, IFN-I sensitivity, and neutralization capacity of two new HeV strains from clades B and D, relative to the prototype HeV strain and Cedar virus, using cell lines from multiple species where applicable. Accordingly, we have added a new section to the Results, Methods, and Discussion detailing these analyses (lines 245-267, 382-398)

1) What are the mutation profiles of the other three structural proteins besides P/L/G?

Response: The M, N, and F proteins do not contain lineage-defining mutations. Recurrent substitutions are described in the Supplementary Discussion. Consistent with their highly constrained structural roles in virion assembly (M) and genome packaging (N), both proteins are highly conserved; notably, all observed substitutions match the corresponding residues found in HeV-g2 and NiV genomes. In the F protein, we detected substitutions (S88L and A141T) that may influence membrane fusion or interactions with other proteins, including the G protein, as well as modulate recognition by neutralizing antibodies (Supplementary Discussion: lines 178-200, Table 1).

2) The two LD mutations identified in the G protein, S175T and A336T, although not located at the G-EB2/3 interaction interface, may affect viral function by influencing the expression efficiency and conformation of G tetramers or the interaction with the F protein.

Response: While these are plausible hypotheses, we did not find structural or functional evidence to support them. Based on structural comparison with the NiV G tetramer, both in the compact and loose conformations (PDB ids: 8K0C and 8K0D, respectively), both substitutions are located away from the known interfaces within the G tetramer and are distant from K246, a residue shown to play a critical role in F triggering (PMID: 38773072). We cite the structures and study and corresponding study used for this analysis (Supplementary Discussion: lines 210-226).

In contrast, the non-lineage-defining mutations R248G (present in ARRED004_AVL_U_23_1, Clade C) and N306K (present in ARCLU007_AVL_U_43_1, Clade B) are more likely to influence F triggering, based on proximity to functionally characterized regions and prior structural and mutagenesis data. In addition, we do note, however, that S175T overlaps a previously reported linear T-cell epitope (aa. 164-178), suggesting possible role in modulating immune recognition.

We have now expanded the discussion of the potential functional relevance of these and other G substitutions (Supplementary Discussion: lines 52-299).

3) More LD mutations were observed in the L and P proteins. Given the recently solved structure of the NiV L-P complex (Cell. 2025 Feb 6;188(3):688-703.e18.), the authors should provide a more detailed discussion on the potential impacts of these mutations.

Response: We have expanded our discussion of the LD mutations as well as other recurrent substitutions in the L and P proteins, incorporating insights from the recently resolved NiV L-P complex (new Fig. 4c). In L, we highlight three substitutions with higher potential functional relevance (lines 389-390). Notably, L Q321H is positioned near the predicted nucleoside triphosphate entry channel and adjacent to the region that interfaces with the XD domain of the P protein (Suppl. Discussion: lines 276-281; Table 1). Given the role of L-P interaction in the polymerase activity, this substitution could plausibly influence replicase function. Nonetheless, strain 71, used in our in vitro assays, did not exhibit significant differences in replication kinetics relative to the prototype strain, suggesting that L Q321H weakening the hypothesized relevance of L Q321H (lines 269-302).

In P, we highlighted six putative non-silent substitutions and discussed their potential functional implications in both the main text (lines 255-261, 386-389; Table 1) and Suppl Discussion (lines 95-161). However, none of these mutations map to regions currently known to mediate direct interaction with L, based on available structural data. As such, while we cannot rule out more subtle or context-dependent effects, the current evidence does not support a direct structural impact on the L-P interface.

We have updated the manuscript accordingly to clarify these points and to reference the NiV L-P structures more explicitly.

4) In-depth analysis of the effects of mutations on viral virulence, cell tropism, and immune evasion is critical for assessing the public health significance of these variants.

Response: Once more we thank the reviewer for emphasizing this important point, which has helped strengthen our manuscript. In response, we have expanded our analysis to discuss the potential functional implications, including viral virulence, host tropism, immune evasion, of all recurrent substitutions, linking these hypotheses to the phenotypic differences observed in our in vitro assays. To balance the level of detail with space limitations, we have provided a comprehensive discussion of individual mutations in the Supplementary Material and highlighted those with the strongest potential relevance in the main text.

5. Fig. 1c is not cited in the Main Text

Response: We have corrected this:

“The largest number of genomes were recovered in 2017 and 2018 between June and September, corresponding to the peak of HeV shedding within the natural reservoir during the Australian winter months (Fig. 1c-d, Extended Data. Fig. 4)”

“Overall, the HeV-g1 phylogeny showed a spatially diffuse structure with a high diversity of strains maintained in the bat population. Co-circulation of multiple strains within and across clades was detected within the same roosts at the same time and over multiple consecutive years (Fig. 1c-d, Fig. 2).”

Minor Issues:

1. Replace hyphens with en-dashes for ranges (e.g., "Suppl. Fig. 9–11").

Response: We have changed this.

2. Line 44: Replace the comma after “genetic diversity” with a semicolon in the keywords section.

Response: We have corrected this.

3. Line 156: Add "is" after "Clade B".

Response: We have corrected this.

4. Line 194: Correct "clade C and D" to "clades..."

Response: We have corrected this.

5. Line 221: Add "in" before "Australia".

Response: This have been corrected

6. In Fig.3, there is a free node between the 2000–2010 grid lines.

Response: Thank you. We have corrected this.

Reviewer #3 (Remarks to the Author):

In this paper, the authors generate critical genomic resources for the bat-borne zoonosis, Hendra virus, increasing the total number of available full genome sequences for HeV from 16 to 73, with the vast majority of newly contributed sequences derived from *Pteropus alecto* flying foxes, the natural reservoir hosts for HeV. The authors also obtain three isolates of HeV from their recovered samples (though the paper shares limited information about the virus isolates other than their sequences). After sequencing, the authors build a series of phylogenetic trees to explore the HeV genetic diversity in time and space. They identify a clear molecular clock signal in the data and highlight evidence of 4 co-circulating HeV clades with no clear spatial structuring. This is clearly important work, providing commendable genomic contributions to the literature. However, the conclusions and claims of the paper feel a bit overstated to me. Key concerns include:

1. Overstated title: The manuscript title, ‘Spatio-temporal dynamics of Hendra virus in *Pteropus* bats in Australia reveals high evolutionary diversity linked with spillover,’ made me think the paper would share evidence of expanded HeV diversity in reservoir bats associated with the timing and location of spillover events to horses– but this is not the case. Indeed, no clear links to spillover are actually demonstrated in this paper. A more accurate and understated title would be more appropriate.

*Response: We have modified the title into: “Spatio-temporal dynamics of Hendra virus in *Pteropus* bats in Australia reveals stable maintenance of diverse viral clades”*

2. Overstated abstract: similar to above, the abstract claims that the paper provides “crucial insights into how bats generate and maintain their extraordinary viral diversity, with direct implications for zoonotic disease emergence and pandemic threats”. In truth, it does not really offer any explanation for how bats generate and maintain viral diversity, nor does it link this diversity to cross species emergence or pandemic threat. These are big claims that need to be better supported if they are to be made.

Response: We have adjusted the title of the manuscript and abstract in response to the reviewers' comments. See comment 1 and 2 Reviewer #1.

3. Both the abstract and the discussion argue that the absence of identifiable spatiotemporal genotypic structuring in the HeV sequences generated here provides support for a hypothesis of persistent infection of HeV in bat hosts and contrasts hypotheses of lineage turnover and selection through immune evasion. Given the sparse number and frequency of genomes recovered in time and space, I wonder if the data are really strong enough to say this? Certainly, the authors do find evidence of a molecular clock structure in their phylogeny, which suggests some role for virus evolution and possible immune escape. I would expect deeper tMRCA roots for a population-level phylogeny of persistent infections especially for hosts as long-lived as bats (e.g. as seen with population level phylogenies for the chronic pathogen, HIV). I would also expect that persistent infections would muddle the temporal signal in root-to-tip phylogenies because some bats would be sampled earlier vs. later in their infection history, with different degrees of within-host evolution. Some kind of forward simulation phylodynamic model with comparison against the data will be needed to really make a claim this grandiose. At a minimum, the role for within-host virus evolution in muddling the phylogenetic inference should be discussed.

Response: We thank the reviewer for this important observation. Our intention was not to assert direct evidence for persistent infection at the individual bat level, but rather to describe a pattern consistent with long-term maintenance of multiple viral lineages within the host population over time. To clarify this and avoid overinterpretation, we have revised the abstract and discussion to make this clear. This phrasing better reflects the available data and acknowledges that the observed temporal stability of multiple co-circulating lineages could arise from a range of ecological and epidemiological processes, including but not limited to persistent infections.

Abstract, lines 53-55:

“The high HeV diversity within shedding pulses and temporal stability of co-circulating clades suggests that viral dynamics are driven by episodic shedding of existing lineages maintained at the population level, rather than immune-driven strain replacement dynamics.”

Discussion, lines 345-361:

“By contrast, persistent infections should show higher viral diversity within each pulse due to within-host evolution, but low diversity between pulses as long-lived infections seed repeated shedding events from the same lineages. At the time, the large-scale genomic surveillance data required to explore these hypotheses did not exist; however, the current dataset allows us to begin to evaluate them. An acute infection model would be expected to produce transient lineage

dominance and frequent strain replacement that should yield phylogenies with strong temporal clustering but limited patterns of co-circulation. Instead, although our data shows a robust molecular clock signal, it also reveals long-term coexistence of multiple divergent clades across years, inconsistent with the rapid lineage turnover expected with an acute infection model. These patterns could therefore reflect persistent or low-level infections within individual bats, but they also might be explained by repeated re-introductions within a large, well-mixed metapopulation (Fig 3). Distinguishing between these mechanisms would require explicit phylodynamic modelling and, most importantly, longitudinal sampling of individual bats for within-host data, all which remain major gaps. Our data here supports the observation of population-level stability of HeV lineages, with limited evidence for immune-driven evolution. The persistence of multiple clades alongside episodic shedding pulses points to ecological triggers such as food stress or host condition, while the underlying mechanism, whether extended infections in individual bats or repeated re-infection across a connected metapopulation, remains unresolved.”

4. Structural modeling seems out of place: While only a minor concern, Fig. 4 and the AlphaFold structural predictions felt a bit disconnected from the rest of the manuscript, especially considering that the conclusions of this modeling work are fairly muted, as HeV is a known pathogen, and differences between the strains identified here and those previously identified in GenBank is fairly minimal.

Response: As requested by another reviewer, we have substantially expanded our analysis of functional implications to all recurrent mutations, beyond the initial focus on those defining Clade B. Using structural information (from modeling or available structures, such as the recently solved structure of the NiV L-P complex, PDB id: 9BDQ) and prior literature, we identified at least 13 substitutions with higher potential for functional relevance (now summarized in Table 1).

We contextualized these substitutions within the framework of viral protein structure and known functional domains, and where appropriate, link them to phenotypic differences observed in our in vitro assays using strain 84 (Clade B), strain 71 (Clade D), and the prototype strain MN062017 (undefined clade). For example, our results indicate that the Clade B-defining mutation P/V/W P352S may contribute with attenuated replication and impaired innate immune antagonism.

Importantly, although HeV exhibits relatively low sequence divergence and lacks clear signatures of immune-driven selection, our findings highlight how even modest genetic variation may yield meaningful biological divergence. We have worked on the text flow and believe that our structure-based analysis now better fits the narrative. We emphasize the importance of continued surveillance and systematic characterization of HeV variants, particularly in light of their public health relevance.

To balance the level of detail with space limitations, we have provided a comprehensive discussion of individual mutations in the Supplementary Material (lines 51-295) and highlighted those with the strongest potential relevance in the main text (lines 226-248 and Table 1).

In the revised version, we have replaced Fig. 4 by a new version, which now maps amino acid substitutions in the G, V, and L proteins, which are highlighted in this study for carrying LD substitutions.

5. Some additional clarity in some of the methods would also be helpful. For example, the authors note in the results that *P. alecto* genome segments recovered from the pooled urine sampling suggest this bat species to be the host of the new HeV sequences --but, unless I missed it, I could not find any additional methods or descriptors about how the host inference was made. At a different point, the authors also mention sequencing some new horse genomes but these are not clearly identified as new contributions in the rest of the manuscript.

Response:

*We agree with the reviewer that this statement can be confusing. When we stated that other species were present, we were referring to bat species visually observed in the bat roosts where the under roost sampling occurred. Additionally, we performed sequence-based species identification using cytochrome b sequences obtained from the sample, showing that all the samples were obtained from *Pteropus alecto* (black flying fox). The two isolates (Hendra 71 and Hendra 84) were obtained from culturing HeV positive urine on Vero E6 cells (African Green monkey, *Chlorocebus sabaeus*) and as such we recovered African Green monkey Cyt B sequences for these two isolate derived sequences. We have added this data to the manuscripts in the Methods section (lines 498–502) and in Supplementary Table 2.*

*Line 125 – 127: “Host genomic sequences detected within the sequenced libraries confirmed *P. alecto* as the source of these samples (Suppl. Table 2).”*

*In addition, Although HeV-g1 has been detected in tissues from all 4 flying fox species in continental Australia, excretion of the virus has been confirmed only in the black flying fox (*P. alecto*) and the spectacled flying fox (*P. conspicillatus*), suggesting these species are sources of transmission to horses (PMID: 26469523, 26060997). Given that the endangered *P. conspicillatus* only exists within a small region of far northern Australia, *P. alecto* is understood to be the primary reservoir host of HeV.*

*We have made this more clear within the introduction (lines 87-92), so that it now reads: "Australian flying foxes (*Pteropus spp*) are natural reservoir hosts of HeV, with serological assays, virus isolation and PCR indicating that HeV circulates among all four flying fox species in continental Australia: the black flying fox (*P. alecto*), the spectacled flying fox (*P. conspicillatus*), the little red flying fox (*P. scapulatus*), and the grey-headed flying fox (*P.**

poliocephalus), but the *P. alecto* is the primary reservoir in southeast Queensland and northeast New South Wales, where the majority of HeV spillovers have occurred."

*The first paragraph of the methods in the previous version describes our under-roost sampling method, including that "Sheet placement was prioritized to target black flying-fox roosting locations, but other species were also present." (lines 468-472). To confirm our field expectations that samples came from *P. alecto*, we examined sequenced data for CytB sequences. We have added this to our results section and the methodology has been added to the material and method section (line 125 and lines 530-533, respectively).*

We have now also made it clearer that the 10 additional horse HeV genomes in this study is a new contribution:

"We also generated a further 10 HeV genomes from spillover events in horses, representing a new contribution to the available genomic data. The new bat and horse derived HeV genomes were combined with 15 previously published sequences from NCBI GenBank (Extended Data Table 1)."

6. Finally, most of the inference presented here is derived from maximum likelihood phylogenies, though the authors do include a BEAST analysis in main text Fig. 3. I wish these two approaches had been better integrated: for example, in Supp. Fig 5., the authors show some possible migration paths between localities and strains, which appears to be derived from the ML analysis. There are more robust ways to undertake phylogeographic analysis and joint tree and trait-based inference in BEAST. Given the small number of genomes in the dataset, it is unclear why these approaches were not attempted here.

Response: We thank the reviewer for this helpful suggestion. Initially, due to the limited number of genomes and the absence of strong spatial structure at the level of individual sampling sites, we chose not to conduct a formal phylogeographic analysis. However, in response to the reviewer's comment, we have restructured the analysis to increase statistical power by grouping the 31 sampling sites into seven broader geographic regions. This allowed us to perform a robust discrete-trait phylogeographic inference in BEAST, incorporating Bayesian stochastic search variable selection (BSSVS) to identify well-supported migration routes between regions.

The results of this analysis are presented in lines 195–201:

"To formally test for structured movement, we applied a Bayesian stochastic search variable selection (BSSVS) framework to our genome (codon) dataset. Here, a small number of transitions were statistically supported originating in South-East Queensland (Suppl. Tables 5, Extended Data Table 3), these involved very limited data (n=2 genomes) and likely reflect artefacts of sparse sampling. Beyond this, no other migration routes were strongly supported, consistent with the spatially diffuse structure observed in the phylogenies. Further supporting the

strong mixing between roosts was the observations of the co-circulation of multiple strains within and across clades was detected within the same roosts at the same time and over multiple consecutive years (Fig. 1c-d, Fig. 2)."

The corresponding methods are detailed in lines 594–601:

"For the phylogeographic analysis, the 31 sampling sites from which Hendra virus sequences were obtained were grouped into seven discrete regions: Brisbane–Southeast Queensland, Lismore–Byron–Ballina, Mid Coast New South Wales, North and Central Queensland, Southeast Queensland, Toowoomba, and Cardiff Heights (Suppl. Table 5). Discrete trait phylogeographic reconstruction was performed using a symmetric substitution model, estimating transition events between regions across the posterior distribution of phylogenies. A Bayesian stochastic search variable selection (BSSVS) procedure⁶⁸ was applied to identify the subset of transition rates that best explained the observed spatial diffusion patterns, with statistical support evaluated using Bayes factors."

Line by line comments are as follows:

7. L49: suggestion to break this into two sentences – it is a bit of a mouthful as is

Response: We have changed this into two sentences

"We conducted extensive spatiotemporal sampling and whole-genome sequencing of HeV-positive samples from bats and horses. Genomic analyses revealed four distinct clades and additional cryptic clades."

8. L77: suggestion to clarify that the HeV genome encodes 6 structural proteins; V,W, and C are also encoded but not mentioned here.

Response: We have now included this in lines 78-80:

"HeV is a non-segmented, negative-strand RNA virus with an 18kb genome encoding six proteins: nucleocapsid, phosphoprotein (the same gene also encodes for V, W and C protein), matrix protein, fusion glycoprotein, attachment glycoprotein, and large polymerase."

9. L97: suggestion to specify that this natural reservoir refers to bats (specifically Pteropus spp. flying foxes).

Response: We have modified the sentence as such:

"Most of these existing genomes were obtained from horses (n = 9), with a few from flying foxes, the natural reservoir (n = 4)."

10. L121: Was P. alecto the only bat identified on these sheets or just the most common? Unless I

missed it, methods for how these host assignments were made seem to be missing from the paper.

Response: To supplement the description of field identification of species (lines 451-455, noting that “sheet placement was prioritized to target black flying-fox roosting locations” and that “the number and species of bats immediately above the sheet were recorded”), we have added an additional section within the methods that describes how host genomic sequences were obtained and analyzed (lines 530-533). That text now reads:

“Host species confirmation

To confirm field host species assignment, sequenced reads were used as described in 49. Briefly, adapter trimmed reads were BLASTed 50 against a CytB database and the resulting average hit reads were converted into a table and using custom scripts. Visualization of the relative frequencies of all potential host species were visualized using Krona 51.”

The complete result is reported in Suppl. Table 2 and the text reads:

*“Host genomic sequences detected within the sequenced libraries confirmed *P. alecto* as the source of these samples (Suppl. Table 2)”*

11. L122: The text mentions 10 novel HeV sequences from horses – where do these pop up in the rest of the analyses and in the figures? It would be nice to understand where they fall in the phylogenies.

Response: We have explained the phylogenetic positioning of the horse sequences throughout the results section, e.g.,

“Clade A is made up of eight novel bat strains and two horse strains (from the 2009 and 2014 spillover events, both in Queensland). While the 2014 horse sequence (HORSE003/Horse/QLD/South_Kolan/2014-03-17) forms a monophyletic clade with the bat sequences, the 2009 strain is distantly related with a percentage nucleotide identity of 99.3-99.8% to the rest of the clade.”

“Clade B is grouped into two subclades. One subclade contains just one strain, collected in 2018 from New South Wales, that is 99.5-99.7% identical to the rest of the clade. The rest of the clade shares a nucleotide identity of 99.8-100%, comprising ten bat strains and one horse strain arising from a 2023 HeV spillover in New South Wales. Similarly, clade C is made up of 10 strains, including nine bat strains and one spillover horse strain (HORSE009/Horse/NSW/South_Arm/2013-06-06), all identified in this study, and sharing a nucleotide identity of 99.6–99.7%.”

“This clade also comprises four strains from spillover events in horses, including two in Queensland (HORSE002/Horse/QLD/Tamborine_Mt/2017-05-25 from May 2017 and

HORSE004/Horse/QLD/Currumbin_Valley/2011-08-22 from August 2011) and two in New South Wales (HORSE001/Horse/NSW/Clunes/2015-09-03 from September 2015 and HORSE007/Horse/NSW/Pimlico/2011-08-15 from August 2011)."

12. L136: Is Supp Fig 5 really what you want to reference here? It is not a phylogeny

Response: We are aware that Suppl. Fig. 5 (Extended Data Fig 4) does not include any phylogeny. However, the figure shows the circulating clades over time including cryptic sequences not belonging to clades A-D. Therefore, it supports the sentence "A few cryptic sequences not belonging to clades A-D were detected in horses and humans between 1994 and 2008, during a period when limited bat sampling was conducted."

13. L171: Are there other horse genomes elsewhere?

Response: To the best of our knowledge, these sequences are not available in any public database. To clarify this point, we have revised the sentence as follows:

"Importantly, we did not identify any examples where we could link a spillover strain directly to the sampled bat diversity (both temporally and by sequence identity); however, only 20 of 89 known horse spillovers had sequences available (either from this study or in public databases) for inclusion in the phylogeny."

14. L234: See main comment above -- Is the spatial-temporal extent of sampling sufficient to truly evaluate this hypothesis?

Response: We acknowledge the reviewer's point. As it relates closely to the reviewer's next comment, we have addressed both points together in a combined response.

15. L254: The reference paper does show that NiV clusters spatially at large scales... Is it possible that the region sampled here for HeV is too small to demonstrate this spatial structuring given the large ranges of these bats?

Response: We address this point in lines 362-375 of the discussion, where we compare our findings to NiV studies at comparable spatial scales. However, we have made the inferences from these points more explicit by adding a sentence that explicitly acknowledges that continental-scale spatial structuring remains a possibility (lines 372-375)

"While it remains possible that HeV might exhibit spatial structuring at continental scales beyond our study region, the comparable patterns to those at similar spatial scales in NiV studies suggest our findings reflect genuine ecological dynamics rather than sampling limitations."

16. L269: Has it been shown that HeV uses ephrin B2/B3 receptors in horses and bats? Some

discussion of the ortholog receptors in these hosts would be helpful since these are the source of most of the sequences.

Response: Bossart et al., 2008 (PMID: 18054977) reported that there are no significant differences in receptor function from different species or receptor usage by HeV (an NiV) for the host receptors ephrin-B2 and ephrin-B3 between human, horse, pig, cat dog, bats (Pteropus alecto and Pteropus vampyrus) and mouse. For clarity, we included a brief statement with this reference in lines 151-154.

Figures and tables:

17. What nucleotide or amino acid substitution model was used for the different trees? Fig 1 and Supp Fig 3, 4, 5. I don't see this listed in the caption or reported in the methods.

Response: We used IQ-TREE, which includes the built-in ModelFinder function (<https://www.nature.com/articles/nmeth.4285>). This function automatically selects the best-fit model based on the lowest Bayesian Information Criterion (BIC) and builds the tree using the selected model. We have made this clear in the methods section lines 552-548:

“The best model for distance estimates were identified with the ModelFinder function ⁴⁷ in IQ-TREE2 ⁴⁸ as the one with the lowest Bayesian information criterion (BIC). Once the optimal model was selected, IQ-TREE2 automatically applied it to construct the phylogenetic tree.”

Also, we have added the specific models in the legend of the Fig 1 and Suppl. Fig 3, 4, and 6.

18. Fig 1: I think there is an error in the caption for panel C – the symbol is a purple star, not a triangle for clade D. Also in panel d, it is fine to have the stylized prevalence in the black line but some label on the y-axis would be helpful. Also, it would help to add the year of sampling for the horse and human sequences in panel a – for example the text discusses the 2009 vs. 2014 horse sequences in clade A but it is challenging to figure out which is which. Finally, could you indicate which sequences come from this study and which are from GenBank?

Response: The caption has now been corrected, and axis labels have been added to the prevalence curve in panel D. Additionally, we have added the year of sampling for the horse and human sequences to Fig 1a. In Suppl 3, 4 and 6 we have indicated which sequences are from this study and which are from GenBank. We have now made these clear in the figure legend of these figures:

“.... Greyed tips represent sequences obtained from GenBank.”

19. Fig 3. Does this include horse and human sequences? If not, why not?

Response: This figure includes all HeV sequences from horse, human and bats. We have made this clear in the legend:

“... Bayesian tree was constructed using a relaxed clock and Skygrid coalescent model, incorporating all available Hendra virus genomes to date from horses, humans, and bats.”

20. Supp Table 2 : where are the NCBI accession numbers? These should be listed prior to publication

Response: We have added accession numbers to a new Suppl. Table 2.

21. Supp Fig 2: Is this the average nucleotide identity between all of the 73 genomes combined? It would be more helpful to see how these differ across host species (e.g. bat, horse, human) and possibly roost site. What do the dashed horizontal lines correspond to?

Response: We have now computed nucleotide identity and GC content for bat, human, and horse sequences, and included the results in Supplementary Fig. 2. The dashed horizontal lines represent the average nucleotide identity and average GC content, respectively. We have updated the figure legend to reflect this new analysis:

“Supplementary Fig. 2: Genetic variation and base composition in HeV-g1 for all sequences, bat, human and horse sequences. The genome organization of HeV-g1 and the relative position of each gene are shown at the top of the graph. The genetic variation in the HeV-g1 genome was determined by plotting the average nucleotide identity of 100 bases using a sliding window in steps of 5 bases (left y axis, colored brown, horizontal dashed line represents average nucleotide identity) across the genome (x axis). Genome base composition also was plotted using a sliding window that calculated the average GC content over 100 bases in steps of 5 bases (right y axis, colored blue, horizontal dashed line represents average GC content).”

We note that there is more variation in the bat HeV sequences compared either to the human or horse HeV sequence. We have included this in our results section, lines 135-134:

“Notably, HeV-g1 sequences from bats show greater genetic diversity than those obtained from human or horse spillover cases.”

22. Supp Fig 5d. The colors are too faint to make it possible to read the migration paths. Also, how were these inferred? I don't see phylogeographic methods anywhere.

Response: To address the issue of faint colors, we increased both the color saturation and the line width, making the migration paths easier to distinguish. The inference of the migration paths is described in the figure caption: “An arrow is drawn if the same clade was detected at a different location in the month prior.” A more detailed response to the inclusion of

phylogeographic analysis is provided in response to Key Concern 6.

23. Supplementary text – need to define IDR

Response: This is done (line 111).

24. Supplementary Fig. 10 and 11 are a bit blurry in the file I was provided.

Response: Supplementary Fig. 10 (current Supplementary Fig. 9) was updated with better resolution and Fig. 11 was updated and is now in the main text as Fig. 4c.

Reviewer Expertise:

Referee #1: Henipaviruses

Referee #2: Henipaviruses, structural biology

Referee #3: Virus ecology

Reviewers Comments:

Reviewer #1 (Remarks to the Author):

I wish to thank the authors for their thorough revision of this manuscript. This revision has greatly increased the clarity and the relevance of this manuscript.

I can confirm that all my suggestions and also the other reviewers' suggestions have been fully addressed. I have no further suggestions to make for this manuscript.

Response: We thank the reviewer for this positive comment

Reviewer #2 (Remarks to the Author):

I would like to commend you for your thorough and thoughtful responses to my comments. The revisions and additional explanations have significantly improved the manuscript, addressing all the points I raised in a clear and satisfactory manner. The paper is now stronger and more compelling. I have no further concerns and am pleased to recommend its acceptance in its current form. Congratulations on your excellent work.

Response: We appreciate the reviewer's positive comments.

Reviewer #3 (Remarks to the Author):

This is my second review of this manuscript, for which major revisions were requested in April of this year. The authors have done a commendable job addressing the substantial critiques of all three reviewers, in particular (1) extending the analysis to include a thorough address of the possible functional consequences of observed amino acid substitutions in the different HeV clades identified, as well as (2) thoroughly characterizing the replication kinetics of novel HeV isolates recovered in this paper and the corresponding innate immune responses in human, horse, and bat cell lines. In

addition, (3) the authors included a more thorough phylogeographic analysis in BEAST. The paper is generally in excellent shape, though there are a few remaining issues that need addressing:

1. I appreciate the authors' careful consideration of my concerns regarding the overstated claims of the first version of the manuscript, specifically with respect to an inference of support for a hypothesis of persistent infection. Nonetheless (and here, I am clearly in disagreement with Reviewer 1), there are a few additional places where I find the text to still be overstated—particularly the Significance Statement (L63-67) and the Discussion (L326-365). Indeed, though R1 requests that the authors double down on a single hypothesis, I think that the most the authors can do with the data at hand is to rule out the hypothesis of repeated immune escape and selection. The authors reference citation #38 (Grenfell et al. 2004) where the concept of phylodynamics is first presented; however, they appear to offer two options only: a persistent infection hypothesis and an acute infection hypothesis. In actuality, Grenfell et al. offer three (four if immune enhancement is included), where two types of acute infection are recognized: acute infections with strong cross-immunity (e.g. measles) and acute infections with partial cross immunity (e.g. flu). While I agree that the persistence of strains for long time horizons seems to rule out the flu-like hypothesis, this does not guarantee that this is evidence for persistent infections without repeated sequences from resampled individuals. Indeed, as mentioned in my previous reviews, the clocklike signal is more pronounced in the phylogeny and the tMRCA between sequences more shallow than I would expect if that were the case—unless within-host evolution is extremely slow in persistently infected bats (which may be but we do not know).

Indeed, Grenfell et al. explicitly state with measles-like viruses that “...an immune response that is equally potent against all strains will not generate selection. Therefore, many strains coexist, with relative frequencies determined predominantly by nonselective epidemiological processes. This does not exclude the sporadic occurrence of selection, immunologically mediated or otherwise. Rather, the measles phylogeny indicates that selection is not operating sufficiently consistently to leave its imprint. Instead, the phylogeny is determined by global spatiotemporal strain dynamics: Some lineages persist in regions with low vaccination coverage, whereas others are globally distributed and represent localized outbreaks initiated by imported strains in regions with higher coverage. The high infectiousness of measles permits the rapid geographic spread of these strains, and their phylogenetic lineages reveal substantial spatial mixing.”

Given above, I would recommend that the authors tone down the emphasis on

persistent infections in the Significance Statement and Discussion and present this as a possibility but not a certainty.

Response: We have toned down our emphasis on persistent infection in the Significant Statement and in the Discussion sections:

Significant statement (now Bold Paragraph)

“The detection of multiple co-circulating clades across different years highlights the complex and dynamic nature of HeV within bat populations, and observed differences in in vitro phenotypes among the viruses suggest that circulating clades may differ in biologically relevant ways beyond their genetic variation. The results demonstrate the significant genetic diversity of HeV maintained in bats and that surveillance should be prioritized during periods of ecological stress to better anticipate and prevent spillover events, providing a framework for ecologically-informed spillover prevention in bat-virus systems globally.

Discussion section

“By contrast, other models—including those involving longer infections or partial cross-immunity—could produce higher within-pulse diversity. At the time, the large-scale genomic surveillance data required to explore these hypotheses did not exist; however, the current dataset allows us to begin to evaluate parts of these possibilities. An acute infection model with strong immune escape would be expected to generate rapid lineage turnover, strong temporal clustering, and evidence of repeated selection. Instead, although our data show a robust molecular clock signal, they also reveal long-term coexistence of divergent clades across years and limited evidence for immune-driven evolution. These features appear inconsistent with flu-like dynamics involving frequent immune escape but are broadly consistent with more stable, measles-like dynamics where strains persist through epidemiological rather than immunological processes.”

2. I very much appreciate the addition of detailed in vitro characterization of the HeV isolates in this new version of the manuscript. This is important work that fills much-needed gaps in the field. However, I am confused as it says on L118-119 that three bat samples generated isolates, while for the rest of the paper, only two HeV-71 and HeV-84 are characterized. What happened to the third isolate? It is fine if it was not viable for some reason, but this discrepancy needs to be explained.

Response: The reviewer is right. We could only obtain useful virus stock for phenotypic analyses from two isolates. The third isolate was a mixed infection between a HeV and another bat paramyxovirus, so we are at the moment not able to use this for phenotypic

comparison (we are working on limiting dilution strategies to get a clean isolate for both viruses).

We have added this to the manuscript in line 121

3. I remain confused about the methods used for species identification from the NGS data. On lines L514-516, the authors address this vaguely, including a citation of a paper that tests different approaches for species ID from NGS data; however, it would be helpful to have some more concrete information about what was done here. It seems that reads were pulled from sequencing, BLASTed against a cyt b reference sequence set, and the highest frequency host match was assigned to the sampling session in question. However, no information is given for thresholds in frequency of host match that were used to make species calls. I'd love to see these 'visualizations' as supplementary figures with this paper.

Response: We have now provided a more detail description of the method:

To confirm field host species assignment, sequenced reads were processed as follows: paired end reads were trimmed of adapters using cutadapt, (ref1) reads 1 and 2 of the pair were combined into one file, and aligned to a Cytochrome Oxidase subunit 1 (COX1) database using ⁵⁴ only allowing for a maximum of 1 hit per read. The COX1 database for species identification is based on and available in fasta format from Leray et al., 2019 ⁵³. The blast output was summarized per species and divided by 2 to calculate the frequency. The frequencies per sample were visualized using Krona (ref2) (Suppl. Fig. 5) and the top hit (defined as the one that had the highest frequency) per sample was taken as the putative host species.

In addition, and to clarify the results, we have added to Supplementary Table 2, two tabs: Total read pair classified and Percent top species of read pair and provided a visualization of the Krona as supplementary Fig 5.

4. Perhaps this can't be helped (e.g. if this a journal requirement), but I found it very challenging to page back and forth among three different localities for the figures and tables: the main text, the extended data, and the supplementary data. Would it be possible to combine the non-main text information into one alternative location rather than two?

Response: For Nature Microbiology, supplementary figures and tables that are essential to the manuscript's conclusions are reclassified as Extended Data. Because there is a

limit on the number of Extended Data items permitted, we retained some materials as Supplementary Data, as recommended by the Journal

Minor, line-by-line comments are as follows:

5. L67: As mentioned above, I appreciate the authors' attempt to tone down the language on the persistent infection hypothesis; however, some of that language is still maintained in the Significance statement here, e.g.: "surveillance should be prioritized during periods of ecological stress when shedding from persistent infections is most likely". I suggest to back off that hypothesis here as well

Response: We have corrected per the reviewer's comments. See general comment #1 above.

6. L83: clarify that these 68 spillovers correspond to spillovers to horse

Response: the reviewer is correct that this involved horses. We have made this now clearer.

In the 31 years since it emerged during a disease outbreak in the Brisbane suburb of Hendra, Australia ¹², 68 spillover events into horses have been documented ^{13,14}.

7. L132-134: I agree it appears that the bat sequences are more heterogeneous than human or horse, and this edited extra figure is exciting and helpful. However, it would be stronger, given variable numbers of sequences in each dataset, if you could say this with some statistical certainty? Confidence intervals on the mean identity in the figure and/or some statistics to pair with this statement would help.

Response: We have now computed the statistical significance for the identity across the genome among the hosts. In addition, we have made a figure that shows percent identity across host species. We have modified the results and the legend of the figure respectively to:

Results

"Between host, there is significant differences in Hendra virus genome percent identity ($p < 2.2 \times 10^{-16}$). Notably, bat-derived genomes exhibited significantly lower identity values compared to both horse and human isolates (adjusted $p < 0.0001$), while horse and human genomes did not differ significantly ($p = 0.882$)."

Figure legend

*“Extended Data Fig. 2: **Genetic variation and base composition in HeV-g1 from different hosts.** (a) sequences from all hosts, (b) sequences from human, (c) sequences from horse, and (d) sequences from bat. The genome organization of HeV-g1 and the relative position of each gene are shown at the top of right and left panels. The genetic variation in the HeV-g1 genome was determined by plotting the average nucleotide identity of 100 bases using a sliding window in steps of 5 bases (left y axis, colored brown, horizontal dashed line represents average nucleotide identity) across the genome (x axis). Genome base composition also was plotted using a sliding window that calculated the average GC content over 100 bases in steps of 5 bases (right y axis, colored blue, horizontal dashed line represents average GC content). (e) Distribution of HeV-g1 genome percent identity across host species. Boxplots show the distribution of genome-wide percent identity values calculated in sliding 100-nt windows for HeV sequences derived from humans, horses and bats. Each box represents the interquartile range (IQR) with the horizontal line indicating the median. Whiskers extend to $1.5 \times$ IQR, and individual data points (jittered for visibility) represent individual sliding-window values. Colors indicate host species. Percent identity significance among hosts was determined using a non-parametric Kruskal–Wallis test followed by pairwise Wilcoxon rank-sum tests (Mann–Whitney U tests) to identify which pairs differed significantly.”*

8. L144: Is Supp Table 3 the correct reference here? I am not sure I understand the connection. Maybe this should be Extended Data Table 2?

Response: The reviewer is correct. We have corrected this to Extended Data Table 2

9. L182-183: It is not very clear from the current wording that this refers to physical movement of HeV across the landscape. Please reword this sentence to make it clear what is being tested and why.

Response: We have reworded the sentence and made clear what we are testing:

“To formally assess the extent of physical movement of HeV among geographic regions, we applied a Bayesian stochastic search variable selection (BSSVS) framework to the genome (codon) dataset. This analysis tests whether particular directional transitions between regions are statistically supported, thereby identifying potential migration pathways of the virus across bat populations.”

10: L254: As mentioned above, I am confused by the sudden focus on two new HeV strains? I thought there were supposed to be three.

Response: We have provided the explanation to comment #2, above.

11. L306: typo – “HeV-positives samples”

Response: This is now corrected.

12. L312-314: As mentioned above, this is not a correct interpretation of Grenfell et al, as it is certainly possible for immunizing viruses (e.g. measles) to maintain many clades simultaneously. Sure, this is not a ladder-like arms race but it does not guarantee a persistent infection either. Also, this sentence appears to have some extra words – “due to” should be removed

Response: We have now rephrased this, see response to reviewer’s comment #1

13. L314: typo – should read “replacing one another”

Response: Corrected

14. L319: typo – extra comma

Response: Corrected

15. L332-334: As noted above, please remember that acute infection is not the same as immune escape – e.g. both flu and measles are acute but have different phylogenies

Response: We acknowledge the reviewer’s point. We have incorporated this in our discussion. See comment #1 above.

16. L337-339: Yes, thank you – the authors offer the alternative, measles-like hypothesis here but it is largely ignored for the rest of the discussion.

Response: We have now introduced this measles-like hypothesis earlier in the discussion: Line XX

17. L359-361: I would caution against discounting the hypothesis of species-specific structuring, as HeV does infect multiple species of Australian Pteropus, and sequences of only one are analysed here. Isn’t it possible the HeV phylogeny is also structured by species and that you just lack the data needed to evaluate this here?

Response: We acknowledge the reviewer's point and have made this clear in the discussion:

“By contrast, for bat-borne rabies virus, the phylogeny is structured by host species, with obvious temporal signal reflecting an evolutionary history of host shifts followed by predominantly within-species transmission⁴²⁻⁴⁵. The absence of comparable structuring in our data set might reflect limited host sampling, because HeV infects multiple flying-fox species across Australia, and most bat sequences here are from Pteropus species.”

18. L410: typo “specie”

Response: Corrected

19. L469: typo – missing closing parenthesis

Response: Corrected

20. L514-516: why were these host read visualizations not also included as supplementary materials in the paper? It would be helpful to have some actual methods here rather than the citation that is provided. It seems that reads were pulled from sequencing, BLASTed against a cyt b reference sequence set, and the highest frequency host match was assigned to the sampling session in question. However, no information is given for thresholds in frequency of host match that were used to make species calls.

Response: We have provided the read visualization as a supplementary figure 5. In addition, we have provided details of the methods. See response to comment #3 above.

21. Extended Data Fig 3: Some of the sequences list a sample number (e.g. HORSE003) rather than an accession number. While searching GenBank with the sample number does seem to pull up the right sequences, for consistency, I suggest that the phylogeny should identify them all by accession number.

Response: We thank the reviewer for this insightful comment. The reviewer is correct that both sample identifiers and accession numbers reliably retrieve the corresponding sequences. In the phylogeny, we decided to display sample identifiers (which have a lot more information) to preserve visual clarity and avoid overcrowding of tip labels. As all accession numbers are provided in Supplementary Table 2, this allows unambiguous cross-referencing while maintaining figure readability.

22. Extended Data Fig 5 – Apologies if I misunderstood the permutation analysis but

can you explain why the proportions of permuted samples $>$ and $<$ the observed value do not add up to 1?

Response: The proportions of permuted samples greater than or less than the observed value do not sum to 1 because both include the observed value itself (hence the use of the \leq and \geq symbols rather than $<$ and $>$). These proportions are intended to be interpreted independently rather than in combination; they would only sum to 1 if one excluded the observed value, which is not the case here.

23. Extended Data Fig 6 and 7 – The order of these should be switched to reflect the order they are presented in the main text of the paper.

Response: This is done.

24. Extended Data Fig. 6 – Please indicate the cell type somewhere in the plot, either on the left or right side of rows a/b/c. Additionally, it would be more intuitive to also reverse this plot, with cell type in the columns and gene expression in the rows, since this is how it is oriented in the complementary Fig 2 f the main text.

Response: We thank the reviewer for this helpful suggestion. We have now indicated the cell type on the left side of rows a–c, as recommended. However, we have chosen to retain the current plot orientation, as this layout aligns more intuitively with the structure and flow of the accompanying panels and better supports direct comparison across cell types within each gene expression profile.

25. Fig 6: In the main text, the authors refer to the ‘prototype Hendra’ strain, but in the figure here (and in Extended Fig. 6) they refer to it as ‘Hendra horse’. The same language should be used in both the main text and the figure to avoid confusion.

Response: We have now change both legends to “Hendra virus prototype (Hendra virus/Australia/horse/1994)”

Supplement

26. L44: references supplementary Fig. 8 which does not appear to exist.

Response: There is no reference to Supplementary Fig. 8 in line 44. We have carefully reviewed the manuscript to ensure that all figures, tables, Extended Data, and supplementary materials are appropriately and consistently referenced throughout.

Reviewer #3:

Remarks to the Author:

I thank the authors for their thorough address of my commentary and find the manuscript much improved. There are a few small remaining errors that should be corrected prior to publication:

L123: typos in newly added material—should be changed to “Between hosts, there were significant differences...”

Response: This has been corrected to “Between hosts, there were significant differences...”

L180: run-on sentence---should be made into two sentences or comma should be substituted for a semicolon: “Only a small number of transitions were statistically supported originating in South-East Queensland (Suppl. Tables 5, Extended Data Table 3); these involved very limited data (n=2 genomes) and likely reflect artefacts of sparse sampling.”

Response: We have replaced the comma with a semicolon.

L357: Please change “most bat sequences are from Pteropus species” to “Pteropus alecto” – it is possible that HeV clades segregate by host species within the Pteropus genus, but since your samples are almost-exclusively derived from P. alecto, you cannot identify that signal with the current dataset. Sampling and sequencing of HeV from the other 3 Pteropus species in Australia would be a valuable contribution to the field.

Response: During the editing process to meet the manuscript’s word count requirements, this sentence was removed.

L44 of the Supporting Information still references “Supp Fig. 8 show the permuted and observed proportions of each clade out of the four clades within each 6-month period” but no Supp Fig 8 is included in the manuscript. The rebuttal suggests this is an erroneous observation, but the most recent collection of materials presented for this review still includes this error.

Response: We have now rectified this — it is Extended Data Fig. 5.